# Change detection of bare-ice albedo in the Swiss Alps

Kathrin Naegeli[1,2], Matthias Huss[3,4], and Martin Hoelzle[3]

[1]Institut of Geography and Oeschger Center for Climate Change Research, University of Bern, Bern, Switzerland
[2]Centre for Glaciology, Department of Geography and Earth Sciences, Aberystwyth University, Wales, UK
[3]Department of Geosciences, University of Fribourg, Fribourg, Switzerland
[4]Laboratory of Hydraulics, Hydrology and Glaciology (VAW), ETH Zurich, Zurich

*Correspondence to:* Kathrin Naegeli (kathrin.naegeli@giub.unibe.ch)

**Abstract.** The albedo feedback is an important driver of glacier melt over bare-ice surfaces. Light-absorbing impurities strongly enhance glacier melt rates but their abundance, composition and variations in space and time are subject to considerable uncertainties and on-going scientific debates. In this study, we assess the temporal evolution of shortwave broadband albedo derived from 15 end-of-summer Landsat scenes for the bare-ice areas of 39 large glaciers in the western and southern Swiss Alps. Trends in bare-ice albedo crucially depend on the spatial scale considered. No significant negative temporal trend in bare-ice albedo was found on a regional to glacier-wide scale. However, at higher spatial scales, certain areas of bare-ice including the lowermost elevations and margins of the ablation zones revealed significant darkening over the study period 1999 to 2016. A total glacier area of $13.5 \, \text{km}^2$ (equivalent to about 12% of the average end-of-summer bare-ice area in the study area) exhibited albedo trends significant at the 95% confidence level or higher. Most of this area was affected by a negative albedo trend of about $-0.05$ per decade. Generally, bare-ice albedo exhibits a strong interannual variability, caused by a complex interplay of meteorological conditions prior to the acquisition of the data, local glacier characteristics and the date of the investigated satellite imagery. Although, a darkening of glacier ice was found to be present over only a limited region, we emphasise that due to the recent and projected growth of bare-ice areas and prolongation of the ablation season in the region, the albedo feedback will considerably enhance the rate of glacier mass loss in the Swiss Alps in the near future.

## 1 Introduction

Glaciers are known to be excellent indicators of climate change (IPCC, 2013). Increasing air temperatures and changing precipitation patterns provoke snowlines to rise to higher altitudes and thus a spatially greater exposure of bare-ice surfaces. In connection with a general prolongation of the ablation season, the increased climatic forcing causes an amplification of glacier melt (Kuhn, 1993; Oerlemans, 2001). However, these changing glacier characteristics trigger feedback mechanisms, in particular the positive albedo feedback which enhances bare-ice melting (Tedesco et al., 2011; Box et al., 2012). Hence, the strongly negative mass balances of many glaciers are not solely a direct signal of atmospheric warming but result from a complex interplay of changes in climate forcing and related surface-atmosphere feedback mechanisms.

Currently, there is an on-going debate about the occurrence and rate of glacier and ice sheet darkening worldwide. While studies like those of Oerlemans et al. (2009), Wang et al. (2014), Takeuchi (2001) or Mernild et al. (2015) observed a dark-

ening for one or several glaciers, respectively in the European Alps, the Chinese and Nepalese Himalaya or in Greenland's peripheral glacierised areas over varying time-scales, evidence of darkening from sectors of the Greenland Ice Sheet is less pronounced, leading to controversial discussions (cf. Box et al., 2012; Alexander et al., 2014; Dumont et al., 2014; Polashenski et al., 2015; Tedesco et al., 2016). The recalibration of the MODIS sensors lead to a reduction in spatial extent and statistical

strength of albedo trends over the Greenland Ice Sheet (Casey et al., 2017). Moreover, the emergence of legacy contaminants or radionuclides and heavy metals contained in cryoconite holes at lower elevations on Alpine glaciers (Bogdal et al., 2009; Pavlova et al., 2014; Steinlin et al., 2014, 2016; Baccolo et al., 2017) or outcropping ice in the ablation zone of the Greenland Ice Sheet that contain high dust concentrations potentially associated with paleo-climatic conditions (Wientjes and Oerlemans, 2010; Wientjes et al., 2012; Goelles and Boggild, 2017) and the recognition of the potential role of biological impurities (Cook

et al., 2017; Stibal et al., 2017; Tedstone et al., 2017) may emphasize and amplify the impact of light-absorbing impurities on ice melt.

To date, most long-term studies either used point data from automatic weather stations located in the ablation area of a glacier (e.g. Oerlemans et al., 2009), coarsely-spaced satellite data from the Moderate Resolution Imaging Spectroradiometer (MODIS) (e.g. Stroeve et al., 2013; Mernild et al., 2015), downscaled MODIS data (Dumont et al., 2012; Sirguey et al., 2016;

Davaze et al., 2018) or other remote sensing datasets (e.g. Wang et al., 2014) to infer trends in ice albedo. Mostly, studies validated the satellite-derived albedo values with in-situ data measured at one to several locations on the ground, which is not always ideal (Ryan et al., 2017). However, for the limited size of Alpine glaciers and the complex surrounding topography, the spatial resolution of MODIS data (500 m) is not suitable, and no appropriate high-resolution albedo product is readily available. Thus, studies focusing on alpine glaciers often base their analysis on higher resolution datasets to obtain information about

glacier surface albedo and related changes and processes (Dumont et al., 2011; Fugazza et al., 2016; Di Mauro et al., 2017) or point-measurements from an automatic weather stations, such as the long-term monitoring site on Vadret da Morteratsch, which revealed a point-based mean summer albedo decrease between 1996 and 2006 of 0.17 (Oerlemans et al., 2009). However, when debating the darkening of glaciers, a clear distinction between glacier-wide versus point-based investigations is necessary to be able to clearly separate a darkening effect due to a changing ratio of snow-covered to snow-free areas of a glacier from other

processes affecting the reflectivity of glacier surfaces. Moreover, a separation between albedo changes of bare ice or snow is required to correctly distinguish between differing processes and dependencies impacting snow and ice in particular ways.

In this study, we use Landsat data to obtain spatially distributed bare-ice albedo for 39 glaciers with a total area of 480 km$^2$ (corresponding to about a quarter of the present glaciation of the European Alps) located in the western and southern Switzerland over the 17-year period 1999 to 2016. We focus on the bare-ice areas, defined as glacier surfaces neither covered by snow

nor by thick debris, only. Snow- or debris-covered glacier surfaces affect glacier mass balance by different processes and are thus not of interest to our analysis. We examine trends and their significance to better quantify and investigate a possible darkening of glacier ice in the western and southern Swiss Alps from the point to the regional scale. Causes and external factors that might impact bare-ice albedo and explain its spatial and temporal evolution are discussed.

## 2 Study sites and data

Our study focuses on 39 glaciers located in the western and southern Swiss Alps (Figure 1). All of them are characterised by a surface area of roughly $5\,km^2$ and larger, and thus offer a large-enough spatial extent to study the evolution of bare-ice surfaces and related albedo changes. The investigated glaciers vary considerably in size, ranging from about $5\,km^2$ (Giétro, Schwarzberg) to almost $80\,km^2$ (Aletsch), and span an elevation range from about 1850 m above sea level (m a.s.l.) to over 4500 m a.s.l. (Table 1). According to the most recent Swiss Glacier Inventory, the 39 glaciers covered a total area of $483\,km^2$ in 2010 (Fischer et al., 2014). We used a Sentinel-2 scene (20 m spatial resolution) acquired on the 23$^{rd}$ of August 2016 to manually adjust the glacier outlines and to obtain up-to-date glacier extents totalling $442\,km^2$ (Paul et al., 2016). For our analysis, we excluded heavily debris-covered parts, such as medial moraines or debris-covered glacier tongues, as we focus on the albedo of bare ice only. Hence, the area difference of $41\,km^2$ between 2010 and 2016 does not solely stem from glacier retreat, but is also due to our exclusion of all glacier areas with thick debris cover for this study, i.e. debris-covered glacier tongues (e.g. Zmutt, Unteraar, Zinal, Oberaletsch) or medial moraines (e.g. Aletsch). The obtained glacier outlines for the year 2016 are used in all consecutive analysis, thus the glacier outline is kept constant over the study period and does not evolve with time.

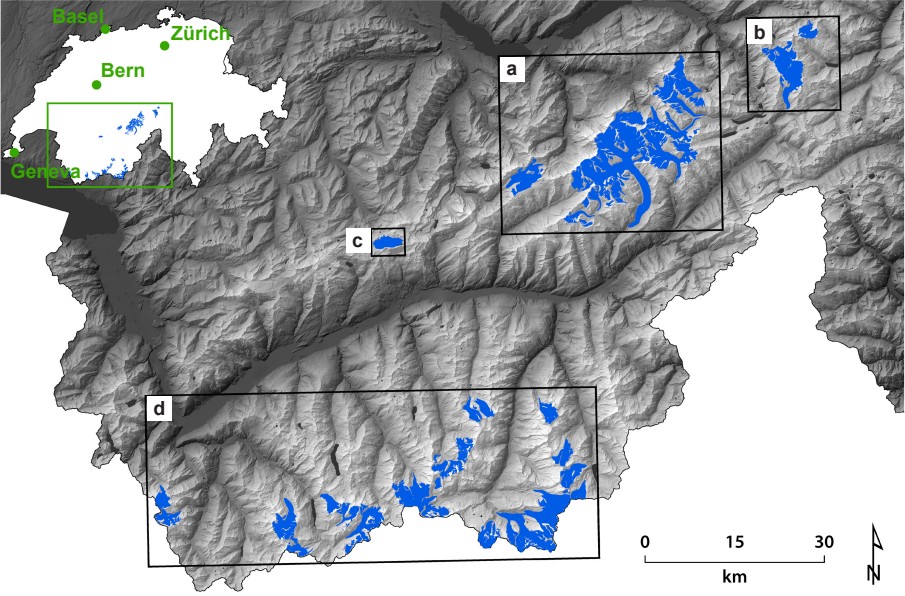

**Figure 1.** Overview of studied glaciers located in the western and southern Swiss Alps. The black frames mark the extent of detailed figures. The inset indicates the position of the study site in Switzerland.

We used the Landsat Surface Reflectance Level-2 science products of the USGS for Landsat 5 and 7 (TM/ETM+) and 8 (OLI) as a basis to obtain broadband shortwave albedo (see Section 3.2). For Landsat TM and ETM+, the product is generated from the specialized software Landsat Ecosystem Disturbance Adaptive Processing System (LEDAPS), whereas the Landsat

OLI product is based on the Landsat 8 Surface Reflectance Code (LaSRC). These data products consist of six (TM/ETM+) or seven (OLI) individual spectral bands in the wavelength range of around 440 nm to 2300 nm, with slight deviations of the individual band widths for the specific sensors. Detailed information about these products can be found in Masek et al. (2006) for Landsat TM/ETM+, and in Vermote et al. (2016) for Landsat 8, as well as in the product guides provided by the USGS. In the context of this study, it is important to mention that both products are neither corrected for topography nor shadow effects. (Claverie et al., 2015) investigated the accuracy of retrieved surface reflectance values based on the LEDAPS algorithm by inter-comparing the product with data from the Aerosol Robotic Network (AERONET) and MODIS data obtained on the same day. This comparison showed good results overall with the poorest performance in the blue band, which is known to have the greatest atmospheric sensitivity (Vermote and Kotchenova, 2008). Most importantly, they found no trend or significant year-to-year variability, suggesting this data product to be highly valuable for temporal analysis. Similarly, (Vermote et al., 2016) analysed the performance of the Landsat 8 surface reflectance product, concluding with high correlations between the MODIS and OLI surface reflectance values, with worst results found again for the blue band, and a general improvement of Landsat OLI surface reflectance product over the ad-hoc Landsat TM/ETM+ LEDAPS product. All 39 glaciers are comprised in one scene (path 195, row 28) and we examined a total of 16 scenes between the years 1999 and 2016, whereupon in 2013 one scene was chosen per glacier individually (Table 2). To obtain maximum information about the bare-ice area of the glaciers, only scenes acquired at the end-of-summer (from months August or September) were chosen. Unfortunately, no good scenes are available for the years 2001, 2007 and 2010. The restriction to choose end-of-summer scenes only hampers the investigation of seasonal changes, but favours an intercomparison over multiple years. On average, a scene comprised $119 \, \text{km}^2$ of bare ice (Table 2).

Our surface type retrieval approach based on the obtained broadband shortwave albedo (see Section 3.2) requires a digital elevation model. We used the DHM25 with an original spatial resolution of 25 m provided by Swisstopo (of Topography, 2005) and resampled it to 30 m spatial resolution to match the broadband shortwave albedo datasets derived from Landsat.

To contextualise our results, the lithology surrounding the individual glaciers based on the lithological-petrographic map of Switzerland (GK500) provided by Swiss Geotechnical Commission (SGTK) was used; the map is at 1:500000 scale, and shows the subsurface strata subdivided into 25 groups according to their formation, their mineralogical composition, their particle size and their crystallinity. Based on Käsling and Thuro (2010) (see their Table 2) these groups were divided into less abrasive rocks (calcareous phyllites, limestones and marly shales, CERCHAR Abrasivity Index (CAI) 0–2) and very to extremely abrasive rocks (amphibolites, basic rocks, gneiss, granites, mica shists and syenites, CAI 2–6). Thereupon each individual glacier was assigned to one of the sub-groups (CAI 0-2 or CAI 2-6).

## 3 Methods

### 3.1 Pre-processing

All reflectance data were downloaded through earthexplorer.usgs.gov (Table 2). The final selection of all scenes is based on a visual check. As cloud masks provided with the science products are known to have certain limitations, in particular for

**Table 1.** Overview of all 39 study glaciers, their area (2010 according to Fischer et al. (2014), 2016 excluding thick debris coverage), elevation range and the lithology of rocks surrounding the glacier. Glaciers are ordered according to their surface area.

| Glacier Name | Area 2010 (km$^2$) | Area 2016 (km$^2$) | Elev. range (m a.s.l.) | Lithology |
|---|---|---|---|---|
| Grosser Aletsch | 78.4 | 74.3 | 1872–4120 | mica shists, gneiss |
| Grenzg | 40.2 | 37.3 | 2319–4536 | mica shists, gneiss |
| Fiescher (VS) | 29.5 | 28.2 | 2102–4082 | mica shists, gneiss |
| Unteraar | 22.5 | 15.9 | 2278–3925 | granites, syenites |
| Rhone | 15.3 | 15.0 | 2300–3621 | granites, syenites |
| Trift | 14.9 | 14.6 | 2191–3383 | mica shists, gneiss |
| Corbassière | 15.2 | 14.2 | 2496–4306 | calc. phyllites, marly shales |
| Findelen | 14.2 | 13.8 | 2661–3929 | mica shists, gneiss |
| Oberaletsch | 17.5 | 12.7 | 2456–3831 | granites, syenites |
| Otemma | 12.6 | 11.0 | 2607–3779 | basic rocks |
| Kanderfirn | 12.2 | 11.4 | 2408–3203 | granites, syenites |
| Zinal | 13.4 | 10.9 | 2466–4035 | granites, syenites |
| Gauli | 11.4 | 11.1 | 2335–3597 | gneiss |
| Zmutt | 13.7 | 10.1 | 2650–4030 | amphibolites |
| Unterer Theodul | 9.4 | 9.1 | 2611–4155 | basic rocks |
| Mont Miné | 9.9 | 9.3 | 2403–3719 | granites, syenites |
| Allalin | 9.2 | 8.9 | 2686–4167 | gneiss |
| Fiescher (BE) | 9.4 | 8.4 | 1999–4086 | limestones |
| Ferpècle | 9.0 | 8.6 | 2289–3659 | granites, syenites |
| Oberer Grindelwald | 8.4 | 8.1 | 1931—3715 | granites, syenites |
| Plaine Morte | 7.3 | 7.5 | 2514–2874 | calc. phyllites, marly shales |
| Lang | 8.3 | 7.2 | 2461–3894 | mica shists, gneiss |
| Fee | 7.3 | 7.1 | 2590–4014 | basic rocks |
| Ried | 7.3 | 7.0 | 2400–4247 | mica shists, gneiss |
| Obers Ischmeer | 7.3 | 6.8 | 2107–3880 | gneiss |
| Saleina | 6.5 | 6.2 | 2320–3863 | granites, syenites |
| Brenay | 7.1 | 6.0 | 2728–3824 | granites, syenites |
| Trient | 5.8 | 5.8 | 2160–3477 | granites, syenites |
| Mittelaletsch | 6.9 | 5.7 | 2599–4059 | granites, syenites |
| Stein | 5.7 | 5.5 | 2190–3462 | amphibolites |
| Mont Durand | 6.1 | 5.4 | 2769–4102 | mica shists, gneiss |
| Rosenlaui | 5.4 | 5.3 | 2102–3623 | granites, syenites |
| Giètro | 5.2 | 5.1 | 2792–3815 | calc. phyllites, marly shales |
| Turtmann | 5.2 | 5.1 | 2409–4141 | granites, syenites |
| Mont Collon | 5.4 | 5.0 | 2532–3670 | granites, syenites |
| Brunegg | 5.5 | 5.0 | 2715–3796 | granites, syenites |
| Schwarzberg | 5.2 | 4.7 | 2728–3549 | mica shists, gneiss |
| Moming | 5.3 | 4.6 | 2692–4036 | basic rocks |
| Tschingelfirn | 5.2 | 4.4 | 2395–3318 | limestones |

bright targets such as snow and ice, but also misclassified medial and lateral moraines, we used a semi-automatic classification approach based on the Spectral Angle Mapper (SAM, Kruse et al. (1993)) implemented in ENVI to detect and delineate clouds obscuring the glacier surfaces. For each sensor (TM, ETM+, OLI) a spectral library of cloud signatures was manually compiled, which served as reference library for the respective sensor. Hence, for each scene we obtained a cloud mask that was used to exclude cloud-affected pixels from all consecutive analyses. Likewise, SAM was used to obtain shadow masks for each individual scene to exclude grid cells that are affected by cloud or topographic shadow effects. Except for the scene taken on the 09[th] of September 2013, when about 14% of the study area was cloud-covered (north-east part of the study area), the cloud coverage was generally smaller than 5% (Table 2). Cloud and topographic shadows were identified up to about 8% of the study area at maximum (28[th] of September 2014, Table 2).

**Table 2.** Overview of Landsat scenes used. The bare-ice area is given in $km^2$ and relative to the total study area of $442\,km^2$. Cloud and shadow coverage are also given relative to the total study area.

| Landsat mission (sensor) | Date (dd.mm.yyyy) | Bare-ice area ($km^2$) | (%) | Clouds (%) | Shadows (%) |
|---|---|---|---|---|---|
| Landsat 7 (ETM+) | 11.09.1999 | 119.7 | 27.1 | 0.1 | 4.5 |
| Landsat 7 (ETM+) | 12.08.2000 | 89.7 | 20.3 | 0.5 | 1.4 |
| Landsat 7 (ETM+) | 18.08.2002 | 61.0 | 13.8 | 1.0 | 3.5 |
| Landsat 5 (TM) | 13.08.2003 | 182.5 | 41.3 | 0.0 | 0.4 |
| Landsat 7 (ETM+) | 08.09.2004 | 132.0 | 29.9 | 0.0 | 2.5 |
| Landsat 7 (ETM+) | 10.08.2005 | 72.6 | 16.4 | 5.9 | 1.0 |
| Landsat 5 (TM) | 22.09.2006 | 95.8 | 21.7 | 0.0 | 6.0 |
| Landsat 7 (ETM+) | 18.08.2008 | 24.6 | 5.6 | 0.1 | 0.8 |
| Landsat 5 (TM) | 30.09.2009 | 185.2 | 41.9 | 0.0 | 7.2 |
| Landsat 7 (ETM+) | 12.09.2011 | 146.8 | 33.2 | 0.4 | 2.8 |
| Landsat 7 (ETM+) | 14.09.2012 | 22.1 | 5.0 | 0.8 | 3.7 |
| Landsat 8 (OLI) | 09.09.2013* | 91.1 | 20.6 | 14.0 | 4.5 |
| Landsat 8 (OLI) | 25.09.2013* | | | 0.5 | 6.7 |
| Landsat 8 (OLI) | 28.09.2014 | 153.5 | 34.7 | 1.0 | 8.3 |
| Landsat 8 (OLI) | 30.08.2015 | 214.9 | 48.6 | 1.4 | 2.1 |
| Landsat 8 (OLI) | 01.09.2016 | 189.4 | 42.8 | 2.3 | 3.1 |

*For each individual glacier only one of these two scenes in 2013 is taken (based on minimal cloud and/or snow coverage).

## 3.2 Albedo retrieval

We applied the narrow-to-broadband conversion by Liang (2001) to obtain shortwave broadband albedo $\alpha_{short}$ from the surface reflectance data. The conversion is based on five of the seven individual bands, and is formulated as follows:

$$\alpha_{short} = 0.356\alpha_1 + 0.130\alpha_3 + 0.373\alpha_4 + 0.085\alpha_5 + 0.072\alpha_7 - 0.0018 \tag{1}$$

where $\alpha_i$ represents the narrowband ground reflectance of TM/ETM+ in band i. For Landsat OLI, the band numbers were adjusted accordingly. This conversion was developed based on a large empirical data set and the band configurations of Landsat TM/ETM+. As shown by Naegeli et al. (2017) this albedo retrieval approach can be applied also to the most recent mission Landsat 8 and is suitable for mountain glaciers. It provides albedo products that have a high accuracy and only deviate marginally (< 0.01) from a more sophisticated albedo retrieval approach if using the same baseline dataset. Uncertainties in
the albedo product not stemming from the retrieval approach but caused by the input data or the general data processing, such as saturation problems over snow covered areas or missing topographic correction on the radiometry, are elaborated in Section 3.5. Unrealistic albedo values, i.e. over 1 or below 0.05, are set to no data.

## 3.3 Surface type evaluation

The delineation of bare-ice area versus snow-covered surfaces is based on a multi-step classification scheme of the surface
albedo values (Figure 2). The classification is thus based on a physical parameter specific for both snow and ice. In a first step, two threshold values for *certainly snow* ($\alpha > 0.55$) and *certainly ice* ($\alpha < 0.25$) are defined (*primary surface type evaluation*, Figure 2) based on recommendations in the literature (Cuffey and Paterson, 2010). This results in a critical albedo range ($0.25 < \alpha < 0.55$), where an unambiguous assignment of the surface type, i.e. snow or ice, is not possible without considering other parameters. Within this range of albedo values, outliers are suppressed by adjusting all albedo values ($\alpha_{corr}$) by multi-
plying with a constant value ($SLA_{const}$). We, therefore, take advantage of a digital elevation model available for all glaciers to evaluate the average albedo in elevation bands of 20 m within this critical albedo range. The transition between ice and snow is typically characterized by a distinct change in albedo (e.g. Hall et al., 1987; Zeng et al., 1983; Winther, 1993). We thus derive an estimate of the mean snowline altitude (SLA) for each glacier and scene based on the greatest slope of the albedo-elevation profile. The albedo for this altitude is considered to be the site- and scene-specific albedo threshold discriminating snow and
ice and is henceforth termed $\alpha_{crit}$.

     In a second step, we use the SLA and $\alpha_{crit}$ as reference to evaluate the surface type within the range of critical albedo values, where there is ambiguity between snow and ice (*secondary surface type evaluation*, Figure 2). Finally, all grid cells are evaluated regarding their relative position compared to the SLA within a critical radius $r_{crit}$ (*probability test to eliminate extreme outliers*, Figure 2). Grid cells located clearly above the SLA are more likely to be snow than ice, and vice versa.
An increasing positive/negative vertical distance from the SLA thus results in penalties for the likelihood of the cell within the critical albedo range of being either snow or ice. As an example, a grid cell near the glacier terminus with an albedo of 0.42, i.e. a rather high albedo for Alpine glacier ice, will be classified as ice. An albedo of e.g. 0.35 observed for the highest

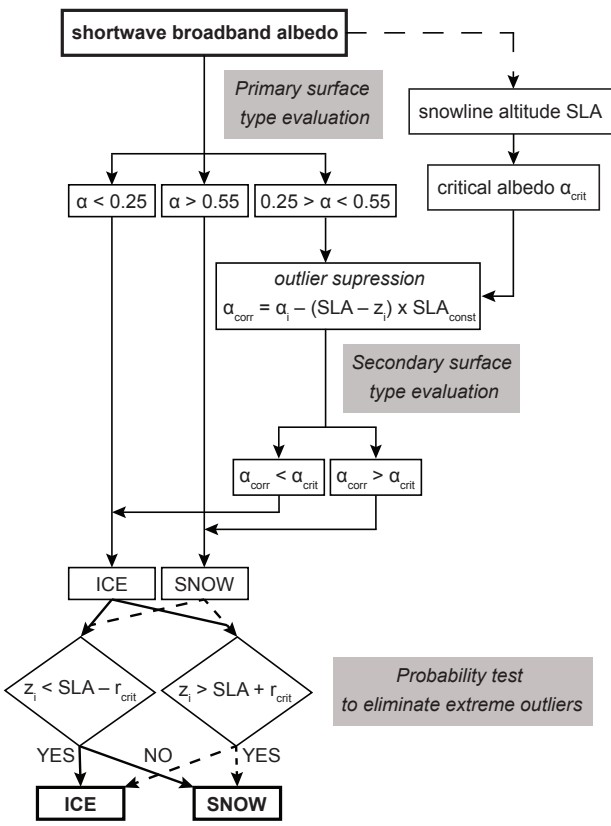

**Figure 2.** Flow chart of the methodology applied to evaluate the surface type (snow/ice) of every glacier grid cell based on the derived shortwave broadband albedo. For further explanations please see text.

regions of the glacier, in contrast, will be classified as snow, as the low albedo is more likely to be explained by an erroneous albedo determination (e.g. shadows) than by actually snow-free conditions. In summary, our procedure to distinguish between snow and bare-ice surfaces relies on remotely-determined surface albedo and merges this information with surface elevation with a probability-based approach to detect outliers and to automatically adapt the classification to the site- and scene-specific conditions.

### 3.4 Trend analysis

Over the study period 1999 to 2016, one end-of-summer Landsat snapshot was available for 15 years (cf. Table 2). Unfortunately, in three years no end-of-summer scene is available due to obscureness of clouds,. Thus, at most the albedo trend of an individual grid cell is characterized by 15 end-of-summer albedo values. However, due to cloud coverage, differing amounts of snow-covered areas in the scenes and/or sensor artefacts, less scenes were usually available to evaluate a temporal bare-ice albedo trend for single grid cells. We arbitrarily set the necessary number of scenes to 50%, thus at least eight albedo values

are required for calculating the albedo trend of one individual grid cell. We used the non-parametric Mann-Kendall (MK) test (Mann, 1945; Kendall, 1975) to evaluate the confidence level of the trends (significant at the 95% / 90% / 80% level, or not significant). For grid cells with significant trends, the magnitude of the trend was determined based on linear regression through all available data points. Trends are given as albedo change per decade.

## 3.5 Uncertainty assessment

Our results are subject to uncertainties arising from errors in the input data, the general data processing, the albedo retrieval approach and the availability of data as well as environmental factors. In general, the used input data, the Landsat Surface Reflectance Level-2 science products for Landsat 5 and 7 (TM/ETM+) and 8 (OLI) (see Section 2), are Tier 1 products offered by and suggested to be used for time series analysis at pixel level by the USGS. These data are geo-referenced with $\leq 12\,\text{m}$ radial root-mean-square error and intercalibrated across the different Landsat sensors (Young et al., 2017). Major drawbacks of these data are the missing topographic correction on the radiometry (Young et al., 2017), the saturation problem over snow-covered areas in the TM and ETM+ data and the SLC failure in the ETM+ data post May 2003 resulting in missing data. While the latter is negligible due to the rather small areas studied (see also Section 4.1), the former are of minor impact as only bare-ice areas situated in rather flat terrain on glacier tongues are considered for the analysis of temporal albedo evolution in this study. The retrieval of albedo values from the reflectance products is limited by the availability of spectral information of the input data. The application of a narrow-to-broadband equation (Equation 1) is known to perform reliably, in general and over glacierised areas in particular as outlined by different studies (Knap et al., 1999; Liang, 2001; Greuell et al., 2002; Naegeli et al., 2017). Moreover, the impact of a missing Bidirectional Reflectance Distribution Function (BRDF) correction scheme is negligible, but generally results in a slight underestimation of albedo values (Naegeli et al., 2017). Overall, the uncertainties stemming from the input data, the general data processing and the albedo retrieval approach are hard to quantify and, hence, no exact number is given here. However, as this study focuses on relative changes of albedo rather than absolute values, the conducted analyses based on the given input data can be considered as reliable and robust.

The general data availability is limited, and only end-of-summer albedo evolution could be analysed. For investigating sub-seasonal variations, the frequency of cloud and/or snow-free and high-quality Landsat scenes was to sparse. This lack of data throughout the entire ablation season of the glaciers is mainly caused by the occurrence of clouds, but also other environmental factors, such as fresh snow falls, hinder the investigation of bare-ice albedo. Subsequently, no data was available for three years of the study period (see Section 2). The occurrence of fresh snow on the glacier surfaces is manifested in elevated albedo values and/or strongly reduced bare-ice surfaces. We checked the scenes used in this study to minimise the impact of environmental factors on our retrieved albedo values. For example, for the year 2013, two different scenes are considered and for each individual glacier the more valuable (less snow and/or cloud/shadow coverage) was selected.

The evaluation of bare-ice versus snow-covered grid cells might result in some misclassified cells. Clouds and shadows that were not detected by the removal algorithms may influence/falsify calculated bare-ice albedos of individual grid cells. However, manual checks revealed a low frequency of such cases. Uncertainty due to mixed pixels, specifically pixels along the margins of a glacier, can influence the temporal albedo trend observed in these areas. We minimized this effect by using

glacier outlines updated to 2016 in order to exclude grid cells from the analysis that become ice-free towards the end of the study period.

To account for the uncertainty introduced by the use of one end-of-summer scene only and thus the exclusion of sub-seasonal variability in albedo, the snap-shot uncertainty, we performed a comprehensive uncertainty analysis based on ten
end-of-summer Landsat 8 scenes acquired between 2013 and 2016 (Table 3). The analysis was performed for one glacier, Findelen, as more scenes were available for this glacier due to the overlapping coverage by two different Landsat scenes (path/row 194/28 and 195/28) of this glacier. For the same grid cell and multiple satellite scenes acquired during the same year (1–5 weeks apart at maximum) we found an average variability in inferred albedo of 0.026 over all four investigated years (2013–2016) (Table 3). Assuming that bare-ice albedo remains constant over this short time period in reality, this value
provides a direct uncertainty estimate for local satellite-retrieved albedo that is assumed to be representative for all investigated glaciers in this study.

**Table 3.** Overview of scenes used in the snap-shot uncertainty analysis. Px refers to the number of pixels that were used to derive uncertainty. Mean ($\alpha_{mean}$), minimum ($\alpha_{min}$) and maximum ($\alpha_{max}$) albedo, as well as the mean ($\sigma_{mean}$) standard deviation of point-based bare-ice albedo for each individual scene pair or triple per year are given.

| Year | Day | Px | $\alpha_{mean}$ | $\alpha_{min}$ | $\alpha_{max}$ | $\sigma_{mean}$ |
|------|-----|-----|-----|-----|-----|-----|
| 2013 | 09 Sept. | 1190 | 0.204 | 0.052 | 0.370 | 0.040 |
|      | 25 Sept. |      | 0.233 | 0.051 | 0.361 |       |
| 2014 | 27 Aug. | 3869 | 0.213 | 0.051 | 0.382 | 0.024 |
|      | 12 Sept. |     | 0.224 | 0.052 | 0.383 |       |
|      | 28 Sept. |     | 0.255 | 0.054 | 0.396 |       |
| 2015 | 07 Aug. | 3446 | 0.174 | 0.052 | 0.403 | 0.031 |
|      | 30 Aug. |      | 0.178 | 0.051 | 0.356 |       |
|      | 08 Sept. |     | 0.236 | 0.053 | 0.358 |       |
| 2016 | 25 Aug. | 5495 | 0.152 | 0.051 | 0.315 | 0.008 |
|      | 01 Sept. |     | 0.156 | 0.051 | 0.295 |       |
| mean for 2013–2016 | | 3500 | | | | 0.026 |

To assess the impact of local albedo uncertainty on the determination and the robustness of potential temporal trends, we randomly perturbed the distributed bare-ice albedo values of every grid cell and scene, and for all 39 individual glaciers with the computed average uncertainty of local albedo of 0.026 (average pixel number of 3500). The re-evaluation of the long-
term albedo trends significant at the 80% level according to the MK test revealed that they were not affected by the random perturbation of the albedo values. Both a very similar area of the glaciers' bare-ice surfaces and distribution of trend magnitude

was found in the perturbed datasets. However, for trends significant at the 95% confidence level or higher a slightly smaller area (11 km$^2$) was detected (c.f. Table 3). Within this area, the majority (77%) of all pixels is affected by negative trends, which is highly similar as obtained by the original albedo datasets (cf. Table 4). Moreover, trends in local bare-ice albedo remained robust even if assumed uncertainties were chosen substantially higher than just the value for snap-shot uncertainty.

## 4 Results

### 4.1 Spatially distributed shortwave broadband albedo

Figure 3 shows the spatio-temporal evolution of glacier-wide shortwave broadband albedo for Findelengletscher. The retrieval of meaningful albedo values is restricted by the quality of the surface reflectance data and, thus, the availability of realistic values in the individual bands needed for the narrow-to-broadband conversion. For Landsat TM/ETM+, a saturation problem over snow-covered areas exists, resulting in missing values for these regions (years 1999–2012 in Figure 3). This problem is not present in the Landsat 8 data (years 2013–2016 in Figure 3). Missing data in some of the Landsat ETM+ data, generated due to the scan line corrector (SLC) failure post May 2003, also occurs in our albedo retrievals (e.g. 08.09.2004 in Figure

Generally, the average albedo values for the bare-ice surfaces are rather low, ranging from 0.18 to 0.31 for individual glaciers as a mean over the entire study period. For all 39 glaciers and over the entire study period we obtained a mean bare-ice surface albedo of 0.22. Extreme years with generally very high snowline altitude (2003, 2011, 2015) or very low snowline altitude (2013, 2014) are linked to summers with exceptionally long and warm or rather cold and humid weather situations, and thus strong or weak ablation, respectively (Glaciological Reports, 2017).

### 4.2 Regional and ablation-area trend in bare-ice albedo

We averaged mean albedo over the entire bare-ice area for each year and glacier to obtain 39 individual time-series for the study period 1999 to 2016. As the outlines from year 2016 are consistently used over time, constant, minimal extents per glacier are evaluated. In addition, overall, yearly averages were determined based on the individual time-series of the 39 glaciers (Figure 4a).

Individual glaciers show considerable variations (up to 0.45 difference between minimum/maximum values) of mean bare-ice albedo between years. However, some glaciers show only minor interannual variability of about 0.06 such as for Grosser Aletsch and Unteraar. On average, the glaciers exhibit a range of 0.22 in minimum and maximum values in-between individual years. Due to these large interannual variations, no significant trends in average glacier-wide bare-ice albedo between 1999 and 2016 for 37 out of the 39 glaciers were found. Only two (Brenay, Ferpècle) show slightly positive trends that are significant at the 95% confidence level according to the MK test.

The yearly values of the 39-glacier average albedo time series range from 0.18 to 0.29, with a mean of 0.22. As for the individual glaciers, no significant trend was found for the averaged time series over the period 1999 to 2016.

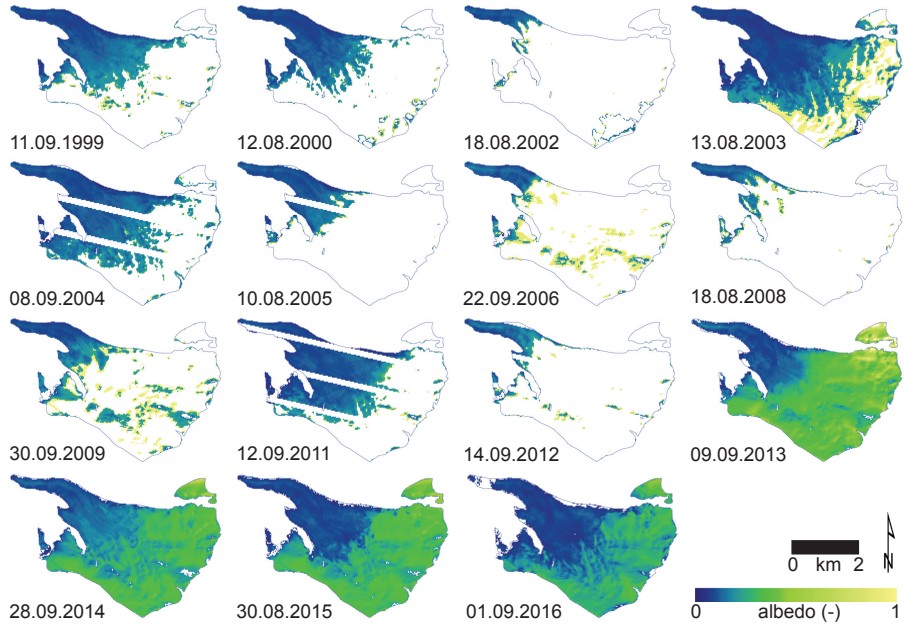

**Figure 3.** Spatio-temporal evolution of shortwave broadband albedo between 1999 and 2016 for Findelengletscher.

## 4.3 Local trend in bare-ice albedo

As trends in bare-ice albedo for the entire ablation area of glaciers might be diluted by averaging over larger areas, or be affected by data uncertainty, we also evaluated the trend in albedo for all grid cells individually. For 114.5 km$^2$ (26%) of the entire surface area of all glaciers, trends were significant at the 80% level according to the MK test (Table 4). Thereof, 13.5 km$^2$

(12%) showed trends significant at the 95% confidence level or higher. Trends were classified according to their magnitude for interpretation. Our classification is shown in Table 3. Classes with clear negative trends (class 1–3) are more abundant compared to classes with no clear or positive trends (class 4–7) at very high confidence level (95% or higher) (Table 3). Thus, significant albedo trends at a confidence level of 95% or higher in the bare-ice areas of the studied glaciers were only detected for grid cells with a rather strong reduction of albedo over the 17 years. Surprisingly, more than 80% of all grid cells with

significant albedo changes at the 95% or higher confidence level showed negative trends: 25% exhibited changes of around $-0.02$ per decade, but almost 60% of cells showed trends more negative than $-0.03$ per decade. For some grid cells, about 15% or 2 km$^2$, also positive albedo trends significant at the 95% confidence level were detected however.

For most of the bare-ice area, the derived trends in albedo were only significant at low levels. Compared to the glaciers' overall ablation area only relatively few grid cells with trends significant at the 95% confidence level or higher (dark blue areas

in Figure 5) are present. The cells with significant trends at high confidence levels are usually situated at the termini or along the lower margins of the glaciers and trends are mostly negative (cf. Table 4, Figures 5–7). The darkening can be attributed to different causes. At the glacier termini, an accumulation of fine debris due to the deposition of allochthonous material

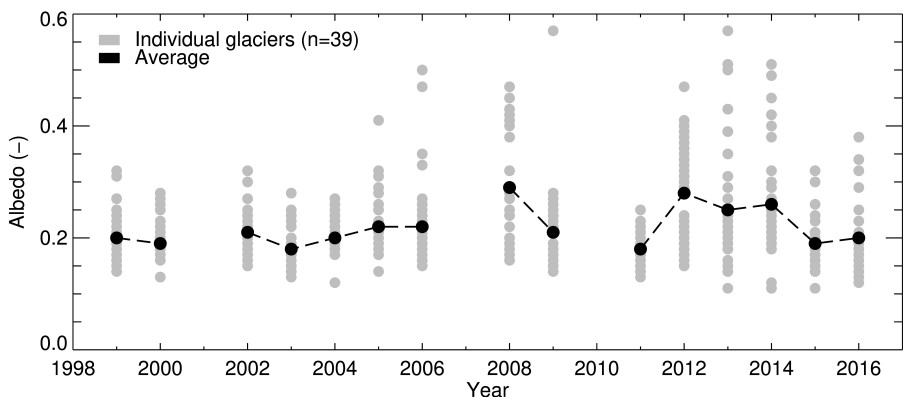

**Figure 4.** (a) Time series of mean bare-ice albedo of all 39 glaciers (grey dots) and their overall average (black dots with dashed line).

**Table 4.** Overview of classes of bare-ice albedo trends for individual grid cells between 1999 and 2016 corresponding to the confidence levels of 80% and 95% according to the MK test. Numbers refer to the sum of all bare-ice grid cells of all 39 study glaciers.

| Class (#) | Albedo trend (albedo change decade$^{-1}$) | Confidence level 80% (%) | Confidence level 80% (km$^2$) | Confidence level 95% (%) | Confidence level 95% (km$^2$) |
|---|---|---|---|---|---|
| 1 | $< -0.05$ | 8.8 | 10.1 | 28.9 | 3.9 |
| 2 | $-0.05$ to $-0.03$ | 11.2 | 12.9 | 28.2 | 3.8 |
| 3 | $-0.03$ to $-0.01$ | 21.5 | 24.6 | 25.6 | 3.5 |
| 4 | $-0.01$ to $0.01$ | 26.0 | 29.8 | 1.8 | 0.2 |
| 5 | 0.01 to 0.03 | 17.9 | 20.5 | 2.7 | 0.4 |
| 6 | 0.03 to 0.05 | 8.4 | 9.7 | 5.8 | 0.8 |
| 7 | $> 0.05$ | 6.1 | 7.0 | 6.9 | 0.9 |
| Total | | 100 | 114.5 | 100 | 13.5 |

and/or melt-out of englacial debris is most likely. These materials, together with the presence of organic material, usually dark and humic substances, decrease local albedo values considerably and foster the growth of algae and bacteria (Hodson et al., 2010; Yallop et al., 2012; Takeuchi, 2013; Stibal et al., 2017). However, many of these effects and interactions are still unclear. Along the glacier margins an increase in debris cover due to small collapses or input of morainic material and, hence, a deposition of rather thick debris on the bare-ice is possible. Moreover, the appearance of debris-rich basal ice alongside the lower glacier margins due to the general glacier recession poses a further cause of local darkening (Hubbard and Sharp, 1995; Hubbard et al., 2009). Along the central area of the glacier tongue, particularly in the vicinity of medial moraines (e.g. in the case of Gornergletscher, Figure 7), a strongly negative albedo trend indicates an expanding medial moraine, changing the local area from clean to (partly) debris-covered ice. In contrast, we also find significant positive albedo trends for some

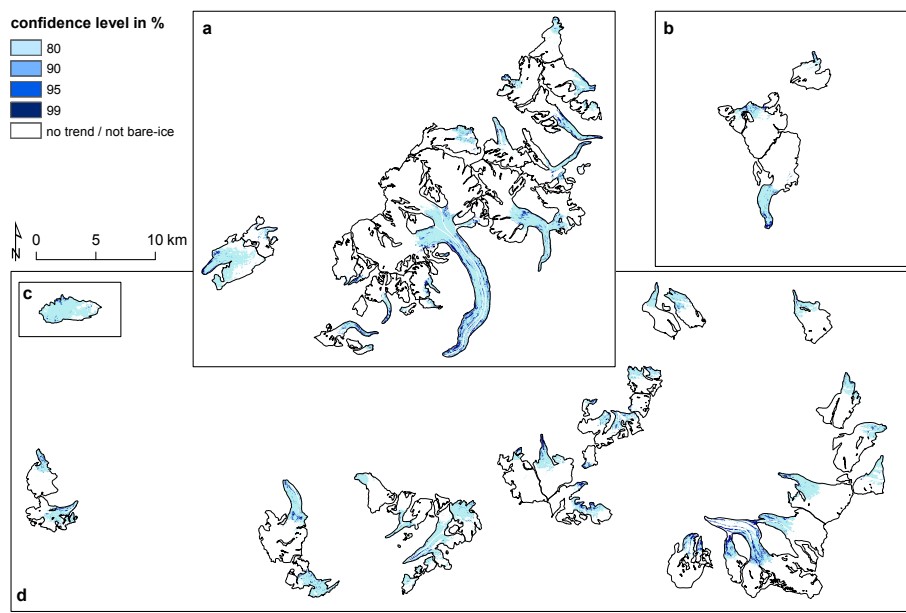

**Figure 5.** Confidence levels of bare-ice albedo trends over the study period 1999 to 2016 according to the MK test. See Figure 1 for the location of the different panels in the Swiss Alps.

locations on the glacier tongues (see Figure 7). These might be explained by the effect of glacier flow changing the position of the medial moraine, hence leading to a transition from debris-covered to clean ice with a higher albedo for certain grid cells. Lateral shifts of the position of medial moraines are possible for retreating glaciers (Anderson, 2000). The investigation of the lithology surrounding the 39 individual glaciers and their overall albedo trend observed for the study period (Table

1) revealed that glaciers predominantly surrounded by less abrasive rocks (calcareous phyllites, limestones and marly shales, CERCHAR Abrasivity Index (CAI) 0–2 after Käsling and Thuro (2010)) exhibited a stronger negative albedo change of $-0.05$ per decade compared to glaciers that are located in an area of very to extremely abrasive rocks ($-0.03$ albedo change per decade; amphibolites, basic rocks, gneiss, granites, mica shists and syenites, CAI 2–6 after (Käsling and Thuro, 2010)).

## 5 Discussion

### 5.1 Temporal evolution of shortwave broadband albedo

Throughout the study period of 17 years, the spatial pattern of bare-ice albedo remained relatively stable for the 39 glaciers. However, the extent of the bare-ice area exhibits a strong interannual variability (Table 2). This is mainly determined by local, temporary meteorological conditions varying strongly from year to year. The meteorological conditions prior to the acquisition dates are crucial as they considerably alter the surface characteristics and thus the observed broadband shortwave albedo.

Moreover, a prolonged ablation period has a strong impact on surface properties such as surface roughness and, hence, also

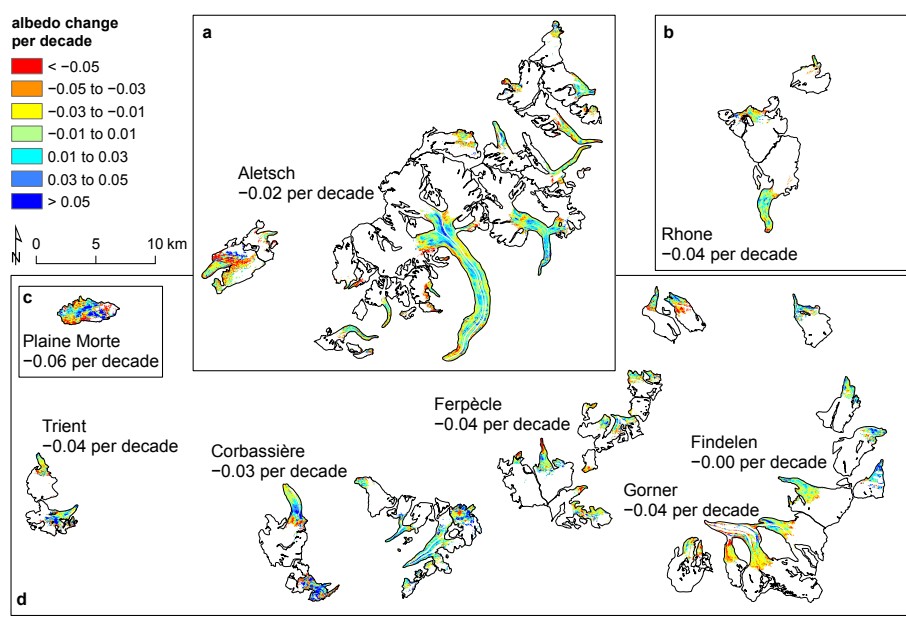

**Figure 6.** Classified albedo trends per decade for all grid cells with trends significant at the 80% confidence level or higher according to the MK test. Averages for areas with bare-ice albedo trends significant at the 95% confidence level are given for selected large glaciers. See Figure 1 for the location of the different panels in the Swiss Alps.

impacts on glacier surface albedo (Cathles et al., 2011; Rippin et al., 2015; Rossini et al., 2018). On smaller spatial and temporal scales, variations in glacier surface albedo are further evoked by meltwater redistribution of impurities (Hodson et al., 2007; Irvine-Fynn et al., 2012).

However, these complex surface-atmosphere interactions are still rather poorly constrained, in particular the temporal dimen-
sion, and further research in this area is needed. Nevertheless, as this study focused on end-of-summer (August and September) scenes only, the relative variations between the individual years is comparable and robust.

In general, relatively low bare-ice albedo values were detected for all glaciers and over the entire study period. It is therefore conceivable that a darkening process occurred before the beginning of our observation period in 1999. However, unfortunately there is no data to investigate this hypothesis. The general conclusions of this study are thus valid for the investigated period,
but do not exclude a possible darkening over a longer time span.

### 5.2 Spatial scales of trends

A clear distinction between regional/glacier-wide bare-ice and local albedo changes is necessary if temporal trends are investigated. A negative trend in glacier-wide albedo (i.e. including both the ablation and the accumulation area) does not necessarily indicate a darkening of the glacier surface but rather a shift in snowline, or in other words, an enlargement of the bare-ice area
relative to the total glacier surface. This effect is particularly pronounced in times of rising air temperatures and prolonged

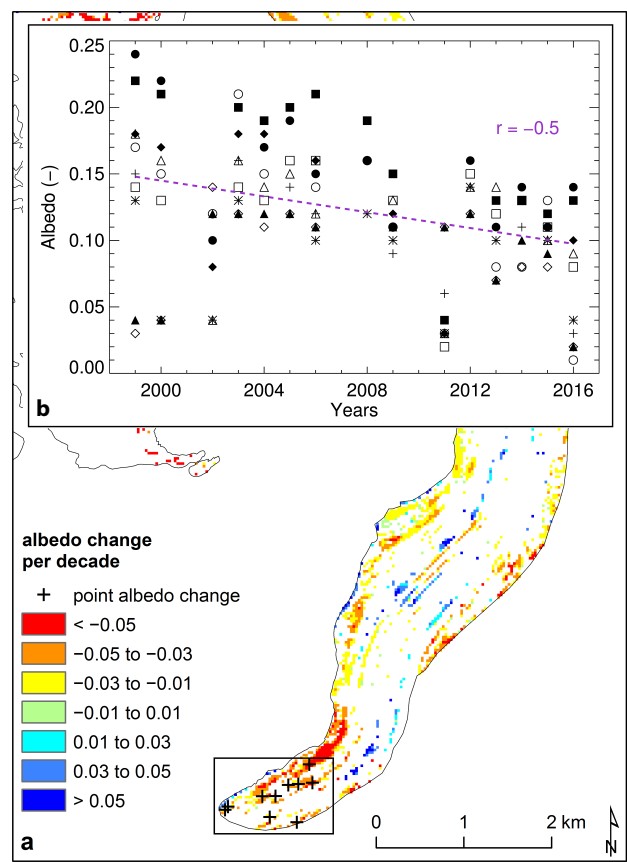

**Figure 7.** (a) Close-up of bare-ice albedo trends per decade significant at the 95% confidence level or higher for the tongue of Aletsch, and (b) time-series of bare-ice albedo between 1999 and 2016 for ten randomly selected points on the terminus (crosses in (a)) including a linear fit (dashed purple, r = −0.5).

ablation periods. In contrast, a negative trend in bare-ice albedo can be an indicator of a darkening phenomenon due to an increased abundance of light-absorbing impurities (mineral dust, organic matter, algae, soot, etc.). Similarly, a lack of trends in bare-ice albedo change at the regional scale does not necessarily exclude the presence of significant trends at the local scale for individual grid cells.

5    In the frame of this study, we were unable to detect a spatially wide-spread, regional trend in bare-ice glacier albedo at a significant confidence level. However, for certain regions of the glaciers, such as the lowermost glacier tongues or along the lower margins, significant negative trends were found. Hence, a clear darkening was observed at the local scale for a limited number of grid cells rather than for entire ablation areas. These findings are in agreement with published literature, but also show that findings of previous studies conducted at the local scale cannot be generalized for an entire glacier or the regional

10   scale. For example, Oerlemans et al. (2009) observed a strongly negative albedo trend at a fixed location close to the terminus of Vadret da Morteratsch, Switzerland, based on weather station data. They found an albedo reduction of 0.17, from 0.32 to

0.15, between 1996 and 2006. This trend is substantially higher than those detected in the present study over the period 1999 to 2016 (see also Table 4). Other studies investigated glacier-wide albedo trends and found negative albedo trends of around 0.1 over the period 2000 to 2013 for Mittivakkat Gletscher, Greenland (Mernild et al., 2015) or up to 0.06 during the period 2000 to 2011 for nine glaciers in western China (Wang et al., 2014). However, the differing study periods, the varying observation scales and the impact of local characteristics on albedo changes make a direct comparison of albedo trends susceptible to misinterpretations.

## 5.3 Possible causes and dependencies of bare-ice darkening

In contrast to the quasi-continuous measurement setup of an automatic weather station, which is however only representative for a limited spatial extent (Ryan et al., 2017), airborne and spaceborne remote sensing datasets only represent a snap-shot in time. Hence, the temporal variability is only included to a certain degree and thus provokes a snap-shot uncertainty in surface albedo for evolution analyses. The meteorological conditions prior to the acquisition of the remote sensing imagery are highly important for the snap-shot uncertainty (Fugazza et al., 2016). Naegeli et al. (2017) highlighted this fact by cross-comparing albedo products from three different sensors with acquisition times within one week. If glacier-wide albedo is compared, a dataset acquired later in the ablation season is expected to show a larger bare-ice area characterised by low albedo values compared to a dataset acquired at the beginning of the melting period. However, this is only true, if meteorological characteristics between the individual acquisition dates are relatively constant. Snowfall or heavy rainfall events might significantly alter the ice surface conditions and the associated albedo values. While fresh snow increases the albedo strongly (e.g. Brock, 2004) and decreases the extent of the bare-ice area (Naegeli et al., 2017), rain can have a two-sided effect. A heavy precipitation event can lead to a short-term (between 1 to 4 days (Azzoni et al., 2016)) increase in albedo due to decreasing surface roughness and/or wash-out of fine debris present on the ice surface (between 5 to 20% according to Brock (2004) and Azzoni et al. (2016)), whereas light rainfall can cause the presence of a thin waterfilm on the glacier ice surface that absorbs radiation much stronger than the underlying ice and thus result in a decreased albedo. Similarly, a long-lasting phase with high air temperatures or intense shortwave radiation input during mid-day can lead to a permanent or temporary waterfilm on the ice surface that reduces reflectivity and thus shortwave broadband albedo considerably (Cutler and Munro, 1996; Jonsell et al., 2003; Paul et al., 2005). Moreover, a remaining thin snow cover might cause slightly increased albedo values in the ablation area (still being in the typical range of glacier ice) that is difficult to be recognized with remote sensing data sets only (Naegeli et al., 2017). Besides these more direct linkages between meteorological conditions and the presence of impurities on the glacier surface, there are many indirect and still rather poorly studied relations. The evolution of the uppermost ice layer, often referred to as weathering crust, is strongly modulated by the local meteorological conditions throughout the ablation period. Surface properties such as microtopography or grain/crystal size are thus changing strongly over time and with them the basic conditions of the bare-ice surface to hold light absorbing impurities and/or facilitate an environment for organisms living in and on the ice surface in cryoconite holes (Irvine-Fynn et al., 2011; Cook et al., 2016; Vincent et al., 2018). Again, the available data sets are thus only representing a snap-shot of the ice surfaces and all its components (Hodson et al., 2007).

Apart from the meteorological conditions that strongly influence bare-ice surfaces, the surrounding lithology of a glacier determines (at least partially) the availability of fine debris material that can be transported by wind and water, and be deposited on the glacier ice, reducing its albedo considerably (Di Mauro et al., 2015, 2017; Azzoni et al., 2016). Thus, easily erodible rock-types provide more loose material that might be transported by wind and water on to the glacier surface and, hence,

impact the bare-ice albedo. This is supported by our analysis of the surrounding lithology and the albedo change of each individual glacier. However, no relation between the albedo of the surrounding geology and the magnitude of the ice albedo change was evident. These findings indicate the importance of the surrounding rocks as possible debris input source on a glacier, in particular as lateral moraines tend to become steeper and more instable due to general glacier recession in times of global atmospheric warming (Fischer et al., 2013), as well as their influence on the energy balance of the nearby glacier ice

and snow surfaces. While some glaciers are surrounded by large lateral moraines that provide a great source of debris that can be transported on to the glacier, others are partly covered by wide medial moraines. The dynamics of these medial moraines due to the general glacier dynamics are poorly studied. However, in the context of this study it is important to note that lateral shifts and growth and/or loss in volume of medial moraines might strongly impact the albedo evolution of some parts of the glaciers. Areas covered by thick debris were excluded from all analyses, but some mixed grid cells alongside medial moraines

might still impact the results locally. Thus, the occurrence of grid cells with positive albedo changes is not surprising, but hard to explicitly link to one specific cause such as the dynamics of medial moraines. The latter might favour local positive albedo changes over time. Localized microtopographic effects, i.e. changes in slope and aspect or modulations in the surface crust (e.g. growth of larger, brighter ice crystals) and the development of cryoconite holes (in contrast to a thin dispersed debris layer) can also strongly impact the evolution of bare-ice albedo.

The discussion of these uncertainties and dependencies, highlights only parts of the complex spatio-temporal evolution of glacier surface albedo. While some influential factors mediating bare-ice albedo are obvious but challenging to quantify (e.g. meteorological conditions prior to the acquisition of data, micro-topography of the surface, etc.) others despite being quantifiable, are more ambiguous (glacier geometry, surface slope and aspect, surrounding lithology, etc.). Based on the presented results we therefore emphasize the need for further investigations of temporal and spatial dependencies of bare-ice albedo

changes regarding various meteorological or geomorphological conditions and their interactions.

## 6 Conclusions

Based on 15 Landsat scenes over a 17-year study period, we assessed the spatio-temporal evolution of bare-ice glacier surface albedo for 39 glaciers in the western and southern Swiss Alps. Our results indicate that the considered spatial scale (local versus regional) is crucial for the investigation of albedo trends and the detection of a potential darkening effect that is often

referred to in recent literature (Takeuchi, 2001; Oerlemans et al., 2009; Dumont et al., 2014; Wang et al., 2014; Mernild et al., 2015; Tedesco et al., 2016). While we did not find a darkening of bare-ice glacier areas at the regional scale or averaged for the ablation areas of individual glaciers, significant albedo trends (95% confidence level or higher) were, however, revealed at the local scale. These individual grid cells or small areas were mainly located at the glacier termini or along the lower glacier

margins in case of negative albedo trends (84% of all significant trends), and along the central flowline further up-glacier in case of positive albedo trends (16%).

The presented study is subject to various uncertainties stemming from the input data itself, its processing and availability, the albedo retrieval approach or environmental factors. However, unfortunately most of them are hard to numeralise. Nevertheless, our uncertainty assessment revealed highly similar trend patterns, thus indicating the robustness of the inferred albedo trends. We would like to emphasize the importance of the snap-shot uncertainty — limited availability of end-of-summer scenes demand recognition. Specifically, the meteorological conditions preceding the acquisition of the satellite data can influence bare-ice albedo, e.g. summer snow fall events, and so should be taken into account.

Although, only snap-shots of glacier surface albedo are available, the almost two-decade long time-series indicate significant trends for about 13.5 km$^2$ (corresponding to about 12% of the average end-of-summer bare-ice surface in the study area) at the local scale. Thereof almost 8 km$^2$ exhibit clear negative trends of $-0.03$ per decade. In contrast, only about 2 km$^2$ of all grid cells with significant albedo trends show positive ( $+0.03$ per decade) and about 4 km$^2$ show weak changes in bare-ice albedo ($> -0.03$ and $< +0.03$ per decade). For the areas with negative albedo trends over the last two decades, the ice-albedo feedback enhanced melt rates which are expected to be enforced in the near future. Even though the darkening of glacier ice has been found to occur over only a limited area of the investigated glaciers, the projected enlargement of bare-ice areas characterised by low albedo coupled with the predicted prolongation of the melt season will most likely strongly impact on the glacier surface energy balance and substantially enhance glacier mass loss.

*Acknowledgements.*  This study is funded by a grant of the Swiss University Conference and ETH board in frame of the KIP-5 project Swiss Earth Observatory Network (SEON). KN was further supported by an Early Postdoc.Mobility fellowship of the Swiss National Science Foundation (SNSF, grant P2FRP2_174888). Landsat Surface Reflectance products were provided by the courtesy of the U.S. Geological Survey Earth Resources Observation and Science Center. The digital elevation model was obtained from the Federal Office of Topography swisstopo. We thank T. Irvine-Fynn for his constructive comments on the manuscript. Two anonymous reviewers are acknowledged for their constructive comments.

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
