# Peer review of "Darkening Swiss glacier ice?"

_The Cryosphere, 2018_

## Short Comment (SC1) · 13 Feb 2018

Page 19: Please cite the final version of the paper, and not the discussion paper: GOELLES, T., & BØGGILD, C. (2017). Albedo reduction of ice caused by dust and black carbon accumulation: A model applied to the K-transect, West Greenland. Journal of Glaciology, 63(242), 1063-1076. doi:10.1017/jog.2017.74

---

## Referee Comment (RC1) · Anonymous Referee #1 · 26 Feb 2018

Dear Editor,

the paper by Naegeli et al. addresses the topic of glacier darkening in the Swiss Alps using Landsat derived surface albedo. The often-referenced "darkening" is still a debated topic in glaciology both regarding mountain glaciers and ice sheets. The paper is very interesting and fits the aims and scope of The Cryosphere journal. I have a couple of major concerns (and some minor issues) that have to be addressed before final publication in TC.

Major comments:

1- The authors use late-summer Landsat scenes for retrieving the seasonal minimum of glacier albedo of Swiss glaciers. Then they apply a trend analysis to bare ice albedo in order to detect positive or negative trends in the albedo series. My first concern regards the choice to study trends only on bare ice. Actually, important contribution

to the radiative balance of glaciers comes also from the accumulation areas: in fact this part of a glacier plays an important role in determining its mass balance. Albedo decrease in the accumulation area is very important, and I don't understand why the authors did not include this part of the glaciers in their analysis. In the area across the ablation and accumulation zone the impact of light-absorbing impurities is important. Excluding this area from the analysis is not justified in my opinion.

For example, Gabbi et al. 2015 showed that black carbon and dust have an impact glacier mass balance. They used data from ice cores collected in the accumulation basin of two Swiss glaciers. I think that the comparison with mass balance should be done with averaged albedo over the entire glacier, and not only from bare ice. In fact, it is not straightforward that the mass balance is determined only by bare ice albedo. I'm not surprised that the authors did not find any correlation between those two variables (pg9 ln3-4).

Furthermore, in the whole paper I did not find any reference to 'cryoconite' (organic and inorganic sediment found on ice). I think that a discussion about "what" could be the cause of the darkening is necessary in this work to give a broader perspective to the remote sensing analysis. Further discussion should regard also the competing role of grain growth (due to ice melting) in potential ice darkening.

2- My second concern regards the choice of Swiss glaciers for the analysis, and the validation of the trends. I was a little surprised that Morteratsch glacier was not included in the analysis. As far as I know, it is the only Swiss glacier with a long series of albedo measured with an Automatic Weather Station in the time windows used in this paper (1999-2016). Oerlemans et al. (2009), was one of the first papers dealing with ice darkening in the Alps. In that paper, a decreasing trend of summer albedo was detected (from 2003 to 2006), and associated with dust deposition from lateral moraines. This series could have been a perfect validation for the methodology developed in Naegeli's paper. I don't understand why they excluded this glacier from their analysis. Furthermore, they also reference to "Swiss glacier ice" in the title. In my

opinion, some validation is needed for the albedo series derived from Landsat. This trend analysis is strongly dependent on the availability of Landsat data during late summer. From table 2, it is evident that differences of more than one month in the dates of the images can create inconsistencies in the albedo retrieval. For examples, early snowfalls in September can generate strong overestimation of albedo.

Specific comments:

Title: it is correct? I think it should be something like "Are Swiss glaciers getting darker?" or "Is Swiss glacier ice getting darker?"

pg2 ln2: here a brief review of these "controversial discussions" should be reported, in particular for Greenland trends. Please, consider also a reference to Casey et al. (2017).

pg2 ln3: recently Baccolo et al. (2017) showed that also radionuclides and heavy metals are contained in cryoconite holes, and will be likely released with current and future melting.

pg2 ln6: I personally don't like references to Discussion papers that was not accepted for publication. I suggest to reference a successive paper by the same authors: Goelles & Boggild (2017).

pg2 ln11: here I suggest a reference to Dumont et al. (2012) that used downscaled MODIS data to estimate surface albedo of a glacier in the French Alps. Also a recent paper by Davaze et al. (2018) used a similar dataset to compare mass balance with albedo.

pg4 ln1-4: here there is no description on how you used this lithological-petrographic map. Please be more specific on the aim of this analysis. I don't see the added values of this analysis. Objectives must be clearly stated.

Section 4: In section 4 (Results), also interpretations are found. I suggest to merge this section with the Discussion, or to move all the interpretations and discussion to

Section 5.

pg8 ln6: "for some analysis". Please, be more specific here.

pg8 ln29: replace ";" with "."

pg10 Fig.4: In 1999, averaged albedo is already very low (about 0.2 from Figure 4). So it is possible that the darkening trend of Swiss glacier ice already occurred. I think that this discussion should be added to the paper.

pg10 ln3: "for some grid cells", please add the percentage.

pg12 ln2: Results from the lithological analysis should stay here

pg15 ln27: here a reference to organic material should be made. Probably the effect of the organic fraction of cryoconite may overwhealm the mineralogic signature of surrounding rocks.

pg16 ln 7: a recent paper by Rossini et al. (2018) also explored the relation between ice darkening, roughness and melting in a Swiss glacier

References:

Baccolo, G., Di Mauro, B., Massabò, D., Clemenza, M., Nastasi, M., Delmonte, B., Prata, M., Prati, P., Previtali, E., and Maggi, V. (2017). Cryoconite as a temporary sink for anthropogenic species stored in glaciers, Scientific Reports, 7, 9623, https://doi.org/10.1038/s41598-017-10220-5.

Casey, K. A., Polashenski, C. M., Chen, J., & Tedesco, M. (2017). Impact of MODIS sensor calibration updates on Greenland Ice Sheet surface reflectance and albedo trends. The Cryosphere, 11(4), 1781.

Davaze, L., Rabatel, A., Arnaud, Y., Sirguey, P., Six, D., Letreguilly, A., & Dumont, M. (2018). Monitoring glacier albedo as a proxy to derive summer and annual surface mass balances from optical remote-sensing data. The Cryosphere, 12(1), 271.

Dumont, M., Gardelle, J., Sirguey, P., Guillot, A., Six, D., Rabatel, A., & Arnaud, Y. (2012). Linking glacier annual mass balance and glacier albedo retrieved from MODIS data. The Cryosphere, 6, 1527-1539.

Goelles, T., & Boggild, C. (2017). Albedo reduction of ice caused by dust and black carbon accumulation: A model applied to the K-transect, West Greenland. Journal of Glaciology, 63(242), 1063-1076. doi:10.1017/jog.2017.74

Rossini, M., Di Mauro, B., Garzonio, R., Baccolo, G., Cavallini, G., Mattavelli, M., De Amicis, M. and Colombo, R. (2018). Rapid melting dynamics of an alpine glacier with repeated UAV photogrammetry. Geomorphology, 304, 159–172. https://doi.org/10.1016/J.GEOMORPH.2017.12.039

---

## Referee Comment (RC2) · Anonymous Referee #2 · 1 Mar 2018

Dear Editor,

The authors use 17 years of repeated end-of-summer Landsat images to investigate the change in bare-ice albedo over the ablation area of 39 large glaciers in Switzerland. In doing so, this study has the merit to address an important research question currently debated in glaciology regarding the potential lowering of the albedo of ice surfaces. This is an important topic in the context of the significant demise affecting mountain glaciers around the world. Capturing whether the spatial distribution of surface albedo of glaciers exhibits trends in the current context is important to characterize processes and longer-term effects associated with the widespread retreat.

To do so, the authors rely on near-yearly Landsat (TM/ETM+/OLI) Level-2 data corrected for atmospheric effect. An empirical formulation by Liang (2001) is used to estimate broadband albedo. This is aimed at being applied to bare-ice surfaces only,

thus relying on an algorithm to determine the snowline elevation and segment the area over which "bare-ice" albedo is computed. The evolution of albedo is then retrieved per pixel and trends assessed via statistical testing and regression analysis.

When all glaciers are considered globally, no trend in mean albedo seem to emerge. Nonetheless, the authors find that statistical testing of trends per-pixel reveal that most of the ablation areas exhibit trends significant at the 80% level, in which case albedo change equally increases or decreases, and about 10% of the ablation area exceeding the 95% confidence level, and where a decrease in albedo is observed to prevail. The authors suggest a number of processes at work to explain this trend, namely accumulation of debris, presence of organic material, and the role of lithology in the area producing loose light-absorbing material potentially accumulating on bare ice.

In my view, and despite the merit of addressing this important question, I find the proposed manuscript requires substantial methodological improvements and significantly better support for the conclusion being drawn before this work be acceptable for publication in The Cryosphere.

General comments

My main concern is that the method claims to assess the changes of albedo on bare-ice. However, most of the ablation areas of the glaciers under consideration exhibit large medial moraines and changing debris covers on sides and terminus, capable to affect strongly the albedo signal. It comes that much of the changes found to be significant enough appear obviously related to changes in the spatial distribution of debris rather than a "darkening of bare-ice" surfaces as suggested by the authors. I am however concerned that the study gives only little acknowledgement to the fact that the target has changed in most instances, but rather insist on the fact that significant areas of bare-ice are perceived to darken. I find this insufficiently supported, if not misleading in view of the data and results provided. Removing from the analysis all areas where the significant change in albedo may be associated with the surface not being

bare-ice at some point of the chronology has the potential to change substantially the message of this study, and I believe can compromise the significance and robustness of its current conclusion.

Specific comments

P1L17: Is it "to" or "at higher altitude"? I suppose the authors mean "to" but since the meaning would be different with either preposition, it is important to correct this.

P2L13: remove "necessarily"

P2L15-18: Although it is true that the use of MODIS data to retrieve surface albedo on mountain glaciers is complicated by a relatively coarse resolution, it is not "unsuitable" as the authors claim. Since Dumont et al. (2011) the use of MODIS data to measure temporal variations in glacier surface albedo has proven to be successful to inform about changes occurring on alpine glaciers, see for example:

Dumont, M., Gardelle, J., Sirguey, P., Guillot, A., Six, D., Rabatel, A. & Arnaud, Y. (2012). Linking glacier annual mass balance and glacier albedo retrieved from MODIS data. The Cryosphere 6 (6), 1527–1539. (Doi: 10.5194/tc-6-1527-2012.) Sirguey, P., Still, H., Cullen, N. J., Dumont, M., Arnaud, Y. & Conway, J. P. (2016). Reconstructing the mass balance of Brewster Glacier, New Zealand, using MODIS-derived glacier-wide albedo. The Cryosphere 10, 2465–2484. (Doi: 10.5194/tc-10-2465-2016.) Davaze, L., Rabatel, A., Arnaud, Y., Sirguey, P., Six, D., Letreguilly, A. & Dumont, M. (2018). Monitoring glacier albedo as a proxy to derive summer and annual surface mass balances from optical remote-sensing data. The Cryosphere 12, 271–286. (Doi: 10.5194/tc-12-271-2018.)

P2L28: The study focuses on glaciers which exceeds 5km2, yet Table 1 reports three glaciers smaller than that.

P2L33: I don't think "of high accuracy" is meaningful or well used in this sentence. I suggest removing it.

P3L1: I believe there is a contradiction between the emphasize put on the fact that the study seeks to characterize changes in surface albedo of "bare ice only" (mentioned 8 times in the abstract and 6 times in the introduction) to discover now that the medial moraines have been kept in the analysis. A number of glaciers such as Aletsch exhibit relatively large medial moraines that are expected to affect strongly albedo estimates. Furthermore, over the time period of the study, it is reasonable to think that the location of the moraines may vary within or beyond the 30-m pixel resolution of the Landsat images (not mentioning the modulation associated with co-registration variability between images), thus convoluting their spectral response with that of bare ice, and potentially creating perceived changes in albedo value.

P3L5: Landsat 7 sensor should be named ETM+ throughout the text, tables and figures.

P3L5: Although it leaves little doubts that the authors are referring to the Level 2 surface reflectance product LEDAPS for Landsat 5-7 for and LaSRC for Landsat 8 OLI, I would suggest these products are named accordingly to their rightful designation for clarity. Since those surface reflectance are not correcting for topography nor shadow effects and that this can compromise the accuracy of albedo estimates, I believe the product being used deserve a more comprehensive description in relation to the context of the study. In particular, it would be important to review what is corrected for in those products, the expected accuracy and limitations of those corrections, and how suitable it is for the study at hand in the present context.

P3L5: As the authors must know and have experienced, all ETM+ data post May 2003 are affected by the SLC failure leaving significant areas of the glaciers missing observations. I must admit I am somewhat surprised that this is not mentioned once, despite a number of images being affected as illustrated in Figure 3, and despite this possibly having noticeable consequences on the computation of mean bare-ice albedo in Figure 4. To me, the inclusion of those data in the analysis would need a specific test to quantify how much the mean albedo may be affected by SLC-off data. This could be

done relatively easily with a SLC-on dataset from which a SLC-off would be simulated using the mask of a SLC-off epoch.

P4L8: "All reflectance data were downloaded"

P4L10: The authors relied on their own cloud classification approach. Since cloud masks are provided as part of the LEDAPS and LaSRC products albeit having known issues, the authors could explain and discuss the reason why they favored a custom algorithm, and how this was assessed to deliver more useful images. I did download a number of the LaSRC images used in this study and left wondering why a custom cloud detection algorithm was needed here, especially given the relatively limited number of images and the visual assessment being made to select those less of not affected by clouds at all.

P4L16: the citation should be Liang (2001), not 2000, same in P19

P4L18: It would be desirable to use the same notation as Liang (2001) with \alpha_i used to represent narrowband ground reflectance of TM/ETM+ in band i.

P4L19: the symbol b_n is not the spectral band number but should be the narrowband ground reflectance of TM/ETM+ in band i. As commented above, using \alpha would make it consistent with Liang (2001) and general understanding of the quantity used.

P4L22: The narrow to broadband conversion assumes ground reflectance on horizontal surfaces. The LEDAPS and LaSRC products account for topography only in terms of the control of elevation on atmospheric effects, yet the control of topography on the modulation of irradiance on varying slope and aspects, hence on the measured satellite radiance in mountainous regions, is ignored. In other words, the surface reflectance products assume a flat surface and the relative geometry between irradiance, the target, and observation direction is not accounted for, let alone the higher orders of effects related to the topography such as terrain reflected irradiance and modulation of the observed reflectance by the BRDF of the target under the said geometry.

[Figure]

Although this effect may be limited on relatively flat glacier tongues and/or when considering only variations rather than absolute values of albedo through time, this and the disconnect between the variable topographic setting under study and the relevance of the empirical formulation of the albedo cannot just be ignored by the authors.

P4L22: I find the claim that the albedo products are of "very high accuracy" and "deviate by less than <0.001 on average from more sophisticated approach" not well informed, if not misleading. It suggests that albedo values retrieved by the empirical method, on the basis of atmospherically but not topographically corrected data may be 100 times more accurate than reference albedo estimated from an albedometer. I can find the sentence in Naegeli et al. (2017) claiming such "accuracy". I however understand this is the mean difference between Liang albedo applied to synthetic APEX data compared to a rigorous albedo derived from the full spectrum. At least in this case the source of data is all APEX and thus topographically corrected. All in Naegeli et al. (2017) suggest that albedo derived from L8 can hardly meet this target even on average. If to comment on the accuracy of this formula (and provided the limitations associated with the use of non-topographically corrected data and other sources of uncertainty is justified), a more useful number would rather be the standard error of this comparison between APEX and APEXLiang, namely 0.11. This is far more realistic with what can reasonably be expected from albedo retrieval from satellite.

P4L25: the definition of such hard threshold on albedo derived from non-topographically corrected data is disputable. Again and related to my previous comment, this should be reviewed and/or critically discussed in view of variability of albedo associated with the topographic control on irradiance.

P4L9: "$(0.25 < \alpha < 0.55)$" not "$(0.25 > \alpha < 0.55)$"

P7L9: snapshot is singular

P7L10: The use of "ideal case" is value-laden. This sentence should be rephrased more objectively.

P8L2: The authors seem to consider equally epochs one (sometimes two) year(s) apart and sometimes just a few days apart to compute trends. I don't think this is an acceptable methodology. When several images are available for the same summer on a grid cell, I think only one estimate should prevail (or an average maybe) for this year and trends only derived from a "yearly" record. This also applies to the linear regression through all available data points. Note also that it is recommended that the Mann-Kendall test be conducted with only one data point per time period (see Chapter 12 in Helsel and Hirsch, Statistical Methods in Water Resources, U.S. Geological Survey, Techniques of Water-Resources Investigations Book 4, Chapter A3).

P8L14: the areas of nodata in ETM+ are not to be referred as striping (which is usually associated with radiometric calibration) but the Side-line-correction failure that occurred in May 2003.

P8L15: the 2004 image seems to be 8/9/2004, not 9/9/2004. All dates in Figure 3 must also be double-checked for erroneous date reported in Table 2 as stressed in a later comment.

P8L16: The authors acknowledge cloud shadow exemplified by one image in the upper accumulation area. They then claim that "bare-ice area is almost always well represented and inferred albedo is realistic thus allowing monitoring through time". I find this claim particularly vague and not supported. What do the authors mean by "realistic"? In what sense would realistic "allow" monitoring? Just looking at both 1999 ETM+ images on Aletsch glacier, the 10/08 is unusable due to cloud cover on most glaciers of interest and in particular Aletsch, while the 11/09 image shows Aletsch tongue severely impacted by cloud shadow, not even to mention Fiescher glacier. Right from this first date, the quality of the image begs the question about how much consideration was given to the "realistic" retrieval of albedo. Beyond the inherent accuracy of the albedo retrieval method which I think is misrepresented here, there is also no consideration of factors such as the different radiometric quality of TM/ETM vs OLI (whiskbroom vs pushbroom, 8b vs 12b radiometric resolution) or imperfect coregistration between image, and the potential of these factors on the quality of the albedo signal.

To me the accuracy of the albedo retrieval method once all source of uncertainties are considered is the main factor that must be given consideration before seeking to make inferences on changes. So far and despite what the authors may suggest earlier, the albedo retrieval for a single pixel can hardly be proven to perform at better than 0.1 accuracy if not worst. A discussion of all environmental factors that may even degrade this further would be welcome. For example how cloud shadow as well as topographic shading whether in the upper reach or on the tongue of glaciers is handled by the methodology remains far too obscure. Yet the relative share of any of such phenomena on some dates could potentially result in variation of albedo. Furthermore, obviously the average bare-ice-albedo are computed on varying areas depending on the classification of the snow/ice limit. Naegeli et al. (2017) themselves reported on substantial variation in albedo up the tongue of Findelengletscher. This should be put into perspective with a mean albedo obtained from a variable number of "bare-ice" pixels at different years.

P10L5: In the context of the relatively large uncertainties being involved in the albedo estimates and interpretation of Table 3 and Figure 5, I find the phrasing that "trends were significant yet at low level" rather misleading. When focusing on those pixels with trends significant at 80% confidence (most of the bare ice), Table 3 reveals a symmetrical distribution (arguably Gaussian) of occurrences exhibiting either positive or equally negative trends. The trends themselves are estimated within the .05/decades or .1 overall magnitude over the approximate 20 years of the study; in other word, barely what we could hope as the uncertainty of the albedo retrieval method. To me this is rather showing that the level of detection (or signal to noise ratio) is simply not enough to be conclusive. Presenting this results and Figure 5 suggesting there are trends significant at least at 80% level everywhere is to me contrary to an alternate interpretation being that this level of detection is not suitable to evidence any obvious trend at all. I think it would be fairer that the interpretation of this result and its significance stress

the limitations of the methods rather than suggesting that there are indeed trends, potentially driven by some physical cause. At this stage, my interpretation of the results given the proposed methodology is simply that it can only be inconclusive.

Turning now to the small portions of pixels exhibiting high confidence (95%) of a trend. In this case, the negative trend apparently prevails. Simply looking at the spatial patterns of occurrences of those pixels in Figure 5 and 6 exemplifies what could be expected of the redistribution and spatial variation of debris and medial moraine on most glaciers. A clear example of this is Gorner glacier. Comparing the 11/9/1999 image to recent 25/9/2013 or 30/8/2015 immediately reveals that all those areas of "highly" significant darkening actually don't qualify as "bare-ice" but rather occur mostly due to spatial variation in the distribution of debris. Visually, there is an obvious widening of the medial moraine on the main trunk, retreat of the glacier front and that of tributaries that are exposing rocks, and obvious down wasting with lateral moraines material falling on the glacier. The fact this drives some pixels to appear as exhibiting a strong decrease in albedo is in fact not so much reliant on the analysis presented in the paper to be revealed. More importantly, I am concerned that the paper may suggest a darkening of "bare ice" as if this was a subtle trend associated with increasing concentration of LAI on glaciers when the areas where the changes occur appear mostly to be those exhibiting a step-change in surface type altogether. In view of this, the way the authors elaborate in P10L10 on probable causes associated with such a progressive darkening such as the growth of algae and bacteria is to me far too speculative at this stage, and finally not supported by any new data in the present work.

P10L12: Further to my comments above, I find that the authors far underestimate or seem to lower the role of the debris and changes in the distribution of moraine material in their observations of albedo changes. Based on my visual interpretation of the images used by the authors, it is obvious that there is more than a mere "possibility" than the areas of significant changes are associated to increase in debris cover and changes in medial moraines. I find the authors suggesting that this may only affect

"certain grid cells" not supported by observations.

P12L5: The authors claim that assessing the uncertainties associated with the albedo estimate is beyond the scope of this work. I don't think this is acceptable and as stressed several times in my comments, I believe this and would require a far more thorough consideration of the uncertainties than presently offered, for any inferences being made about processes at work to be deemed robust.

P12L8: What do the authors mean by "better result". This is unspecific, value-laden, and should be unsupported by a stronger argument.

P12L9: As demonstrated above, this study does NOT "focus on bare-ice areas" only. Furthermore, the data quality issue (e.g., SLC off, cloud shadows, saturation) can severely affect that albedo retrieval.

P12L13: The authors claim that "manual checks" revealed low frequency of misclassified pixels compromising the albedo retrieval. My own check alone on the first two dates revealed immediately that the biggest glacier in the study (Aletsch) is severely affected by clouds and cloud shadow to the extent that I don't believe a realistic estimate of albedo could have been obtained across many parts of the glacier tongue. In view of this, one cannot be satisfied by the unsupported claim of the authors.

P13L1: While it is true that using 2016 outlines of glaciers in the context of the current glacier demise would have reduced the dominance of ground becoming exposed dominating the change in albedo at the terminus and lateral moraines, Figure 5 and 6 still reveal that the significant changes in albedo remain associated to areas of probably thick debris deposition. It begs the question about what would have been the signal and conclusions of this work if the analysis had focus on a (conservative) mask of bare-ice, meaning a mask where pixels throughout the study period can be observed as free of thick debris. I think this is the main methodological issue that the authors should address to revise this work.

[Figure]

P13L5: what does "most scenes" means in this context? This sentence is ambiguous and should be clarified.

P13L8: This and later paragraphs relate to a specific methodology and analysis that should be introduced earlier. The steps taken to assess uncertainty should be integral of the research design and reporting of results.

P13L11: The authors stated in the previous paragraph that the uncertainty analysis was performed on Findelen glacier, but now indicate that the 39 glaciers were considered. Please clarify.

P14L1: The repeatability of mean albedo determination on (supposedly) bare-ice areas of Findelen glacier is reported in Table 4, leading to an assessment of uncertainty claimed by the authors on the albedo (pixel-wise) being 0.026. It should be clearly indicated that this reports on the precision (repeatability) of the albedo retrieval only, not its accuracy. I also note that 0.026 is obtained by simply averaging the four estimates corresponding to each year. I am not convinced this simple averaging is providing a fair assessment of the repeatability that can be obtained from this approach. As highlighted in my comment on Table 4, I have concerns about the very small value reported for 2016 and it would contribute to lower the perceived precision. I could not find images in table 4 for 2014 so it leave only two other instances in 2013 and 2015 together suggesting quite a substantially larger precision than 0.026.

P14L1: The fact that the assessment of trends does not appear to change in view of the perturbations is not truly surprising. As discussed above, I don't think the methodology and level of detection is conclusive enough to elaborate on areas exhibiting trends at the 80% CI in the context of the current methodology. It would be more informative to test how this level of significance changes when increasing the perturbation on albedo to more realistic uncertainty levels given all other environmental factors. The area exhibiting most change, generally a strong albedo decline are apparently greatly controlled by a redistribution of debris and moraines, hence does not qualify as bareice throughout the study period. It is expected that those areas remain confidently detected as areas of strong change in albedo. In conclusion, I cannot agree with the author's statement that the "inferred trends in local bare-ice albedo are considered to be robust despite the uncertainty in the albedo retrieval".

P14L11: the expression "snap-shot uncertainty" is vague and unspecific. I recommend that this unfamiliar and uncommon wording is revisited to express more plainly what uncertainty the authors are referring to.

P15L17-24: This far-reaching interpretation is not supported by any tangible results presented in present study. Presented as it is now and provided some of the weaknesses of the methodology, this equates more to a relatively general hypothesis rather than one directly informed and supported by the data and results presented here.

P17L9: I believe the authors mean "snap-shot" and not "snap-short" yet I maintain that the use of this "new" terminology is not specific enough to make it informative of a clear source of uncertainty, and hence would advise against the use of it.

P17L11: The authors insist in their conclusion that "meteorological conditions preceding the acquisition (. . .) need to be considered". I however saw no such consideration in the manuscript despite most images exhibiting various stages of snowline retreat, some of them with obvious short-lived snowfall.

P17L12: I am actually quite concerned that one of the main conclusion point is that "highly significant darkening" affect about 10% of the ablation areas. Based on my own interpretation of many of the images used, much of this darkening can be attributed to change in surface type from bare-ice to debris, rather than a "darkening" of bare-ice via the accumulation of LAI for example. To me the point raised by this study rather stresses the potential accumulation of debris on the lower reaches of mountain glaciers in the context of retreat.

Table 2: Landsat 7 sensor should be named ETM+

Table 2: Although it is understandable that rounding may bring variations in numbers at the decimal level, it would be preferable that in 2008 when 0 km2 cloud is reported, this correspond to 0% as well. Table 2: Some of the information reported in the table 2 appears to be incorrect. I could not find images for quite a few dates: I suggest 8/9/2004 (not 9/9/2004); 22/9/2006 (not 20/09/2006); 27/08/2014 (not 26/08/2014); 12/09/2014 or 28/09/2014 (not 1/9/2014 and 27/09/2014). The sensor in 9/9/2013 appears to be L8 and not L7. This also affects Table 4.

Table 2: I am perplex about the reporting of clouds in 25/9/2013 and the reason for reliance on a secondary date 9/9/2013. I downloaded both dates and it appears that clouds in 25/9/2013 indeed marginally affect the east of area (d), however barely above the terminus of Bachi and Minsti glaciers, both outside the scope of this study. I could not see anywhere else where clouds may have caused an issue and the need to rely on an alternate date. This begs the question about the performance of the cloud classification algorithm used by the authors. It is even more confusing since the 9/9/2013 image (L8 and not L7 as reported by the authors) is obviously far worse with many clouds and the fact that the authors specifically mention this image in P4L13. Looking at both dates, it is obvious that the snowline retreated over the period, thus exposing more of the bare ice, at least for those glaciers not obscured by clouds and as reported in Figure 3. It suggests the use of two dates in this instance may be done to retrieve albedo over larger areas of bare ice, at least on some glaciers, but the reader can only wonder. Even in this case, it would beg the question about the consistency of the data being used and the potential effect on mean albedo. Table 4 shows that several scenes are used for assessing uncertainty, but also shows an inconsistency with table 2 as multiple acquisitions in 2013 and 2014 are consistent, while those in 2015 and 2016 are not reported in Table 2.

Figure 2: "snowline altitude" not "snowline altitdue"

Figure 2: what are SLAconst and rcrit?

Table 4: I could not find any image on 1/09/2014 nor 27/09/2014.

Table 4: In 2016, the number of pixels used for the assessment is 5495, thus representing ∼5km2 of supposedly bare-ice surface. I obtained both images and could only map ∼3km2 of ice at the most, the rest being mostly the accumulation area still obviously covered by snow. Note also that the lower part of the glacier tongue is severely affected by cloud shadow on 1/9/2016. Beside an issue of size being considered as bare-ice which I can't reconciliate with what the images depicts and shedding doubt on the performance of the snow/ice classification, the albedo variability appears surprisingly small (0.008) when any other years yield about five times larger albedo precision. I believe some clarification is required here.

---

## Author Comment (AC3) · 17 Apr 2018

*Page 19: Please cite the final version of the paper, and not the discussion paper: GOELLES, T., BØGGILD, C. (2017). Albedo reduction of ice caused by dust and black carbon accumulation: A model applied to the K-transect, West Greenland. Journal of Glaciology, 63(242), 1063-1076. doi:10.1017/jog.2017.74*

Thanks for this short comment. We changed the reference accordingly.

---

## Author Response (AR1)

Dear Editor

We hereby submit the manuscript entitled "**Darkening Swiss glacier ice?**" by Kathrin Naegeli, Matthias Huss and Martin Hoelzle to be considered for publication as an article in The Cryosphere. We have considerably revised the paper in response to the constructive comments of two reviewers and have tried to respond to all concerns raised.

In response to the two anonymous reviews, we

- clarified the main aim of the paper and deleted the link to glacier mass balance to improve the consistency of the manuscript,
- elaborated the role of supraglacial debris in more detail by: (1) manually delineating medial moraines and areas where tributaries separated from the main glacier trunk and debris has become exposed to obtain a complete supraglacial debris mask based on the Sentinel-2 image acquired in August 2016, and (2) applying this debris mask to all data and, thus, to exclude areas with debris coverage from all consecutive analyses,
- developed the uncertainty assessment of the retrieved albedo values by providing more information about the datasets used as well as their specific constraints and uncertainties that may result thereof in a separate sub-section in the methods section,
- expanded the discussion of possible causes and dependencies of the detected albedo changes in the discussion section. Finally, several valuable references to link our study to existing research were added.

We declare that this manuscript is original, has not been published before and is not currently being considered for publication elsewhere.

Below we respond to all comments by the two anonymous referees. The **responses** (bold font style) are following the referees' comments (normal font style) directly. The corresponding revised sentences in the manuscript are given in quotation marks and particular changes are marked in red.

We hope you find our manuscript suitable for publication and look forward to hearing from you.

Sincerely,

Kathrin Naegeli

**Comments by anonymous referee #1**

The authors use late-summer Landsat scenes for retrieving the seasonal minimum of glacier albedo of Swiss glaciers. Then they apply a trend analysis to bare ice albedo in order to detect positive or negative trends in the albedo series. My first concern regards the choice to study trends only on bare ice. Actually, important contribution to the radiative balance of glaciers comes also from the accumulation areas: in fact this part of a glacier plays an important role in determining its mass balance. Albedo decrease in the accumulation area is very important, and I don't understand why the authors did not include this part of the glaciers in their analysis. In the area across the ablation and accumulation zone the impact of light-absorbing impurities is important. Excluding this area from the analysis is not justified in my opinion.

**We agree that the accumulation areas of glaciers are important for the radiative budget and consequently mass balance of the entire glacier as mentioned by the referee. We also see the importance of albedo changes in these snow-covered areas of a glacier, especially in connection with glacier mass balance. However, changes in ice albedo compared to snow albedo are not linked to the same processes and thus require a separate investigation. Whereas changes in snow albedo are of rather short-lasting impact, due to snow falls that occur more often and year-round, and thus reset the albedo to higher values again, ice albedo changes are usually more durable. Our main research question is, thus, not linked to the mass balance or changes in the energy budget of the entire glaciers due to changing albedo but addresses the general question of the darkening of bare glacier ice. Moreover, the strong saturation problems in the TM and ETM+ surface reflectance data for snow-covered surfaces would substantially restrict an analysis of snow-covered glacier surfaces. We clarified the motivation in the introduction section.**

*"Furthermore, when debating the darkening of glaciers, a clear distinction between glacier-wide versus point-based investigations is necessary to be able to clearly separate a darkening effect due to a changing ratio of snow-covered to snow-free areas of a glacier from other processes affecting the reflectivity of glacier surfaces. Moreover, a separation between albedo changes of bare ice or snow is required to correctly distinguish between differing processes and dependencies impacting snow and ice in particular ways."*
* * *
For example, Gabbi et al. 2015 showed that black carbon and dust have an impact glacier mass balance. They used data from ice cores collected in the accumulation basin of two Swiss glaciers. I think that the comparison with mass balance should be done with averaged albedo over the entire glacier, and not only from bare ice. In fact, it is not straightforward that the mass balance is determined only by bare ice albedo. I'm not surprised that the authors did not find any correlation between those two variables (pg9 ln3-4).

**As mentioned in our answer above, the main research question addresses the general darkening of bare ice and not albedo changes of the entire glacier. This is also already made clear in the title of the article. Thus, our study does not aim at explaining mass balance fluctuations with albedo changes. To increase the consistency of the paper, we deleted all parts of the manuscript that link albedo changes to mass balance, i.e. the statement on pg9 ln3-4 and part (b) of figure 4.**
* * *
Furthermore, in the whole paper I did not find any reference to 'cryoconite' (organic and inorganic sediment found on ice). I think that a discussion about "what" could be the cause of the darkening is necessary in this work to give a broader perspective to the remote sensing analysis. Further discussion should regard also the competing role of grain growth (due to ice melting) in potential ice darkening.

**We are aware of not having mentioned the term "cryoconite" in the manuscript, and have no corrected this omission. Possible causes of the darkening are mentioned at several positions in the paper. We expanded on possible causes and dependencies with a respective sub-section in the discussion section, addressing important points like cryoconite, grain growth, input of debris, surface roughness, etc.**

"Besides these more direct linkages between meteorological conditions and the presence of impurities on the glacier surface, there are many indirect and still rather poorly studied relations. The evolution of the uppermost ice layer, often referred to as weathering crust, is strongly modulated by the local meteorological conditions

throughout the ablation period. Surface properties such as microtopography or grain/crystal size are thus changing strongly over time and with them the basic conditions of the bare-ice surface to hold light absorbing impurities and/or facilitate an environment for organisms living in and on the ice surface in cryoconite holes [*Irvine-Fynn et al.*, 2011; *Cook et al.*, 2016; *Vincent et al.*, 2018]. Again, the available data sets are thus only representing a snap-shot of the ice surfaces and all its components [*Hodson et al.*, 2007]."
* * *
My second concern regards the choice of Swiss glaciers for the analysis, and the validation of the trends. I was a little surprised that Morteratsch glacier was not included in the analysis. As far as I know, it is the only Swiss glacier with a long series of albedo measured with an Automatic Weather Station in the time windows used in this paper (1999-2016). Oerlemans et al. (2009), was one of the first papers dealing with ice darkening in the Alps. In that paper, a decreasing trend of summer albedo was detected (from 2003 to 2006), and associated with dust deposition from lateral moraines. This series could have been a perfect validation for the methodology developed in Naegeli's paper. I don't understand why they excluded this glacier from their analysis.

**The main aim of the study was to try detecting bare-ice albedo changes based on readily available Landsat science products. We thus decided to focus on only one Landsat scene that comprises most of the larger glaciers in the Swiss Alps. Unfortunately, Vadret da Morteratsch is thus not included. To account for the comment made, we included some more information about the observed albedo change on Vadret da Morteratsch published in *Oerlemans et al.* [2009] in the introduction.**

"Thus, studies focusing on alpine glaciers often base their analysis on higher resolution datasets to obtain information about glacier surface albedo and related changes and processes [*Dumont et al.*, 2011; *Fugazza et al.*, 2016; *Di Mauro et al.*, 2017] or point-measurements from an automatic weather stations, such as the long-term monitoring site on Vadret da Morteratsch, which revealed a point-based mean summer albedo decrease between 1996 and 2006 of 0.17 [*Oerlemans et al.*, 2009]. However, when debating the darkening of glaciers, a clear distinction between glacier-wide versus point-based investigations is necessary to be able to clearly separate a darkening effect due to a changing ratio of snow-covered to snow-free areas of a glacier from other processes affecting the reflectivity of glacier surfaces.
* * *
Furthermore, they also reference to "Swiss glacier ice" in the title. In my opinion, some validation is needed for the albedo series derived from Landsat. This trend analysis is strongly dependent on the availability of Landsat data during late summer. From table 2, it is evident that differences of more than one month in the dates of the images can create inconsistencies in the albedo retrieval. For examples, early snowfalls in September can generate strong overestimation of albedo.

**We agree that the use of only one end-of summer scene demands recognition. In particular, the meteorological conditions prior to the acquisition of the scene are of importance. However, the meteorological conditions prior to the acquisition of the scenes used do not indicate any fresh summer snow fall events that would affect the retrieved albedo values. Moreover, our snap-shot uncertainty analysis revealed that the use of end-of summer scenes within a two-month period (August and September) indicates robust albedo results that can be used to study temporal changes. To clarify the uncertainty resulting from such environmental factors, we extended the statements in a new sub-section "uncertainty assessment" at the end of the methods section.**

"The general data availability is limited, and only end-of summer albedo evolution could be analysed. For investigating sub-seasonal variations, the frequency of cloud and/or snow-free and high-quality Landsat scenes was to sparse. This lack of data throughout the entire ablation season of the glaciers is mainly caused by the occurrence of clouds, but also other environmental factors, such as fresh snow falls, hinder the investigation of bare-ice albedo. Subsequently, no data was available for three years of the study period (see Section 2). The occurrence of fresh snow on the glacier surfaces is manifested in elevated albedo values and/or strongly reduced bare-ice surfaces. We checked the scenes used in this study to minimise the impact of environmental factors on our retrieved albedo values. For example, for the year 2013, two different scenes are considered and for each individual glacier the more valuable (less snow and/or cloud/shadow coverage) was selected."
* * *
Specific comments:
* * *
Title: it is correct? I think it should be something like "Are Swiss glaciers getting darker?" or "Is Swiss glacier ice getting darker?"

**The suggested changes to the existing title are not changing its meaning. We thus would like to keep the title as it is.**
* * *
pg2 ln2: here a brief review of these "controversial discussions" should be reported, in particular for Greenland trends. Please, consider also a reference to Casey et al. (2017).

**Thanks for reminding us of this reference. We added a clarifying sentence and the suggested reference.**

"The recalibration of the MODIS sensors lead to a reduction in spatial extent and statistical strength of albedo trends over the Greenland Ice Sheet [*Casey et al.*, 2017]."
* * *
pg2 ln3: recently Baccolo et al. (2017) showed that also radionuclides and heavy metals are contained in cryoconite holes, and will be likely released with current and future melting.

**Thanks for the mentioning of this reference. We adjusted these sentences and added the suggested reference.**

"Moreover, the emergence of legacy contaminants or radionuclides and heavy metals contained in cryoconite holes at lower elevations on Alpine glaciers [*Bogdal et al.*, 2009; *Pavlova et al.*, 2014; *Steinlin et al.*, 2014, 2016; *Baccolo et al.*, 2017] (…)."
* * *
pg2 ln6: I personally don't like references to Discussion papers that was not accepted for publication. I suggest to reference a successive paper by the same authors: Goelles & Boggild (2017).

**We changed the reference accordingly.**
* * *
pg2 ln11: here I suggest a reference to Dumont et al. (2012) that used downscaled MODIS data to estimate surface albedo of a glacier in the French Alps. Also a recent paper by Davaze et al. (2018) used a similar dataset to compare mass balance with albedo.

**We added the studies that used downscaled MODIS data to infer glacier surface albedo.**

"To date, most long-term studies either used point data from automatic weather stations located in the ablation area of a glacier [*Oerlemans et al.*, 2009], coarsely-spaced satellite data from the Moderate Resolution Imaging Spectroradiometer (MODIS) [e.g. *Stroeve et al.*, 2013; *Mernild et al.*, 2015], downscaled MODIS data [*Dumont et al.*, 2012; *Sirguey et al.*, 2016; *Davaze et al.*, 2018] or other remote sensing datasets [e.g. *Wang et al.*, 2014] to infer trends in ice albedo."
* * *
pg4 ln1-4: here there is no description on how you used this lithological-petrographic map. Please be more specific on the aim of this analysis. I don't see the added values of this analysis. Objectives must be clearly stated.

**We agree that the added value of this analysis was not clearly stated. We thus added some statements in the introduction to strengthen our motivation to include this analysis.**

"We examine trends and their significance to better quantify and investigate a possible darkening of glacier ice in the western and southern Swiss Alps from the point to the regional scale. Causes and external factors that might impact bare-ice albedo and explain its spatial and temporal evolution are discussed."
* * *
Section 4: In section 4 (Results), also interpretations are found. I suggest to merge this section with the Discussion, or to move all the interpretations and discussion to Section 5.

**We agree that some interpretations are already presented in the results section. However, in our opinion they are necessary to present the results to the reader in the right context. We would thus like to keep the separation of results and discussion as is.**
* * *
pg8 ln6: "for some analysis". Please, be more specific here.

**We realized that this sentence is not of importance anymore and thus deleted it completely.**
* * *
pg8 ln29: replace ";" with "."

**Changed.**
* * *
pg10 Fig.4: In 1999, averaged albedo is already very low (about 0.2 from Figure 4). So it is possible that the darkening trend of Swiss glacier ice already occurred. I think that this discussion should be added to the paper.

**We agree that the albedo values in 1999 are already rather low, which could be due to local conditions during the Landsat overpass. Speculating about possibly higher albedo values of bare ice in earlier years (70's and 80's) is delicate as we do not have a dataset that covers this time period. However, we added a respective comment in the discussion section to indicate this fact and possible indications for our analysis.**

"In general, relatively low bare-ice albedo values were detected for all glaciers and over the entire study period. It is therefore conceivable that a darkening process occurred before the beginning of our observation period in 1999. However, unfortunately there is no data to investigate this hypothesis. The general conclusions of this study are thus valid for the investigated period, but do not exclude a possible darkening over a longer time span."
* * *
pg10 ln3: "for some grid cells", please add the percentage.

**The percentage and km$^2$ were mentioned in the parentheses at the end of the sentence. We moved the number to the beginning of the sentence.**

"For some grid cells, about 15% or 2 km$^2$, also positive albedo trends significant at the 95% confidence level were detected however."
* * *
pg12 ln2: Results from the lithological analysis should stay here

**We moved the respective section from the discussion to the end of this paragraph (Local trend in bare-ice albedo).**
* * *
pg15 ln27: here a reference to organic material should be made. Probably the effect of the organic fraction of cryoconite may overwhealm the mineralogic signature of surrounding rocks.

**To account for this comment (and others on the same line), we rewrote the paragraph about dependencies of the discussion section. A more detailed discussion of the role of organic fraction present on a glacier surface was added. Please see answer to comment above.**
* * *
pg16 ln 7: a recent paper by Rossini et al. (2018) also explored the relation between ice darkening, roughness and melting in a Swiss glacier

**We added the respective reference.**

**Comments by anonymous referee #2**

My main concern is that the method claims to assess the changes of albedo on bare ice. However, most of the ablation areas of the glaciers under consideration exhibit large medial moraines and changing debris covers on sides and terminus, capable to affect strongly the albedo signal. It comes that much of the changes found to be significant enough appear obviously related to changes in the spatial distribution of debris rather than a "darkening of bare-ice" surfaces as suggested by the authors. I am however concerned that the study gives only little acknowledgement to the fact that the target has changed in most instances, but rather insist on the fact that significant areas of bare-ice are perceived to darken. I find this insufficiently supported, if not misleading in view of the data and results provided. Removing from the analysis all areas where the significant change in albedo may be associated with the surface not being bare-ice at some point of the chronology has the potential to change substantially the message of this study, and I believe can compromise the significance and robustness of its current conclusion.

We agree that many areas with strong albedo changes are somehow linked to supraglacial debris cover (e.g. medial moraines). However, a clear definition of bare-ice or debris-covered ice is not existent in glaciology to our knowledge as the transition is smooth and strongly site-specific with many intermediate stages of dirty ice. Thus, with the remote sensing data used in this study a clear separation of bare- and debris-covered ice is difficult. However, to acknowledge the referee's concern, we manually delineated medial moraines as well as areas where tributaries separated from the main glacier trunk and debris has become exposed. These glacier areas that are obviously covered by thick debris represent 8.5 km$^2$ for this study and were consecutively excluded from all further analyses. Other clearly debris-covered areas of the glacier (e.g. the tongue of Unteraar or Zmutt Glacier) had already previously been excluded from the analysis.

"For our analysis, we excluded heavily debris-covered parts, such as medial moraines or debris-covered glacier tongues, as we focus on the albedo of bare ice only. Hence, the area difference of 41 km$^2$ between 2010 and 2016 does not solely stem from glacier retreat, but is also due to our exclusion of all glacier areas with thick debris cover, i.e. debris-covered glacier tongues (e.g. Zmutt, Unteraar, Zinal, Oberaletsch) or medial moraines (e.g. Aletsch)."
* * *
**Specific comments:**
* * *
P1L17: Is it "to" or "at higher altitude"? I suppose the authors mean "to" but since the meaning would be different with either preposition, it is important to correct this.

**Changed.**

"Increasing air temperatures and changing precipitation patterns provoke snowlines to rise to higher altitudes and thus a spatially greater exposure of bare-ice surfaces."
* * *
P2L13: remove "necessarily"

**Deleted.**
* * *
P2L15-18: Although it is true that the use of MODIS data to retrieve surface albedo on mountain glaciers is complicated by a relatively coarse resolution, it is not "unsuitable" as the authors claim. Since Dumont et al. (2011) the use of MODIS data to measure temporal variations in glacier surface albedo has proven to be successful to inform about changes occurring on alpine glaciers.

**We added the use of downscaled MODIS data in long-term albedo studies and respective references.**

"To date, most long-term studies either used point data from automatic weather stations located in the ablation area of a glacier [*Oerlemans et al.*, 2009], coarsely-spaced satellite data from the Moderate Resolution Imaging Spectroradiometer (MODIS) [e.g. *Stroeve et al.*, 2013; *Mernild et al.*, 2015], downscaled MODIS data [*Dumont et al.*, 2012; *Sirguey et al.*, 2016; *Davaze et al.*, 2018] or other remote sensing datasets [e.g. *Wang et al.*, 2014] to infer trends in ice albedo."
* * *
P2L28: The study focuses on glaciers which exceeds 5km2, yet Table 1 reports three glaciers smaller than that.

**We changed the respective wording.**

"All of them are characterised by a surface area of roughly 5 km$^2$ and larger, (…).""
* * *
P2L33: I don't think "of high accuracy" is meaningful or well used in this sentence. I suggest removing it.

**Deleted.**
* * *
P3L1: I believe there is a contradiction between the emphasize put on the fact that the study seeks to characterize changes in surface albedo of "bare ice only" (mentioned 8 times in the abstract and 6 times in the introduction) to discover now that the medial moraines have been kept in the analysis. A number of glaciers such as Aletsch exhibit relatively large medial moraines that are expected to affect strongly albedo estimates. Furthermore, over the time period of the study, it is reasonable to think that the location of the moraines may vary within or beyond the 30-m pixel resolution of the Landsat images (not mentioning the modulation associated with co-registration variability between images), thus convoluting their spectral response with that of bare ice, and potentially creating perceived changes in albedo value.

**As mentioned in our response to the referee's general concern at the beginning, we delineated all medial moraines and areas where tributaries separated from the main glacier trunk and debris has become exposed based on the Sentinel-2 image from August 2016. These debris-covered areas, 8.5 km$^2$, are consecutively excluded from the investigated study area. Furthermore, we agree that the location and extent of medial moraines over the study period might change. Thus, we strengthened our statement about this process affecting albedo changes in both directions (decreasing and increasing albedo values along medial moraines) in of the discussion section in the revised version of the paper.**

"While some glaciers are surrounded by large lateral moraines that provide a great source of debris that can be transported on to the glacier, others are partly covered by wide medial moraines. The dynamics of these medial moraines due to the general glacier dynamics are poorly studied. However, in the context of this study it is important to note that lateral shifts and growth and/or loss in volume of medial moraines might strongly impact the albedo evolution of some parts of the glaciers. Areas covered by thick debris were excluded from all analyses, but some mixed grid cells alongside medial moraines might still impact the results locally."
* * *
P3L5: Landsat 7 sensor should be named ETM+ throughout the text, tables and figures.

**Changed.**
* * *
P3L5: Although it leaves little doubts that the authors are referring to the Level 2 surface reflectance product LEDAPS for Landsat 5-7 for and LaSRC for Landsat 8 OLI, I would suggest these products are named accordingly to their rightful designation for clarity. Since those surface reflectance are not correcting for topography nor

shadow effects and that this can compromise the accuracy of albedo estimates, I believe the product being used deserve a more comprehensive description in relation to the context of the study. In particular, it would be important to review what is corrected for in those products, the expected accuracy and limitations of those corrections, and how suitable it is for the study at hand in the present context.

**We added more information about the used Landsat products, in particular on the level of correction and the expected accuracy of retrieved surface reflectance values.**

"We used the Landsat Surface Reflectance Level-2 science products of the USGS for Landsat 5 and 7 (TM/ETM+) and 8 (OLI) as a basis to obtain broadband shortwave albedo (see Section 3.2). For Landsat TM and ETM+, the product is generated from the specialized software Landsat Ecosystem Disturbance Adaptive Processing System (LEDAPS), whereas the Landsat OLI product is based on the Landsat 8 Surface Reflectance Code (LaSRC). These data products consist of six (TM/ETM+) or seven (OLI) individual spectral bands in the wavelength range of around 440 nm to 2300 nm, with slight deviations of the individual band widths for the specific sensors. Detailed information about these products can be found in [*Masek et al.*, 2006] for Landsat TM/ETM+, and in [*Vermote et al.*, 2016] for Landsat 8, as well as in the product guides provided by the USGS. In the context of this study, it is important to mention that both products are neither corrected for topography nor shadow effects. *Claverie et al.* [2015] investigated the accuracy of retrieved surface reflectance values based on the LEDAPS algorithm by inter-comparing the product with data from the Aerosol Robotic Network (AERONET) and MODIS data obtained on the same day. This comparison showed good results overall with the poorest performance in the blue band, which is known to have the greatest atmospheric sensitivity [*Vermote and Kotchenova*, 2008]. Most importantly, they found no trend or significant year-to-year variability, suggesting this data product to be highly valuable for temporal analysis. Similarly, *Vermote et al.* [2016] analysed the performance of the Landsat 8 surface reflectance product, concluding with high correlations between the MODIS and OLI surface reflectance values, with worst results found again for the blue band, and a general improvement of Landsat OLI surface reflectance product over the ad-hoc Landsat TM/ETM+ LEDAPS product."
* * *
P3L5: As the authors must know and have experienced, all ETM+ data post May 2003 are affected by the SLC failure leaving significant areas of the glaciers missing observations. I must admit I am somewhat surprised that this is not mentioned once, despite a number of images being affected as illustrated in Figure 3, and despite this possibly having noticeable consequences on the computation of mean bare-ice albedo in Figure 4. To me, the inclusion of those data in the analysis would need a specific test to quantify how much the mean albedo may be affected by SLC-off data. This could be done relatively easily with a SLC-on dataset from which a SLC-off would be simulated using the mask of a SLC-off epoch.

**Thanks for this friendly reminder to add some details about the SLC failure in the ETM+ data post May 2003. We added respective information in the data section of the manuscript.**

**Furthermore, we evaluated the referee's concern about the impact of the SCL failure on mean bare-ice albedo by using the ETM+ SLC-off data for Findelengletscher from 12.09.2011 as a mask and the three scenes from 12.08.2000 (ETM+, SCL-on), 13.08.2003 (TM, SCL-on) and 30.08.2015 (OLI, SCL-on) as test data to obtain sensitivity values. This analysis revealed that the impact of the SLC failure on mean bare-ice albedo is negligible with a difference of 1.2 to 2.2% (e.g. 12.08.2000 SLC-on mean bare-ice albedo 0.204 versus SLC-off mean bare-ice albedo 0.209 indicating a difference of 2.2%).**

"Missing data in some of the Landsat ETM+ data, generated due to the scan line corrector (SLC) failure post May 2003, also occurs in our albedo retrievals (e.g. 09.09.2004 in Figure 3). We tested the impact of the SLC failure by simulating missing data for three scenes with an intact SLC for Findelengletscher. SLC failure resulted in slightly higher mean bare-ice albedo values (1.2 to 2.2%, e.g. 12.08.2000 SLC-on mean bare-ice albedo 0.204 versus SLC-off mean bare-ice albedo 0.209 indicating a difference of 2.2%), which is a negligible impact."
* * *
P4L8: "All reflectance data were downloaded"

**Changed.**
* * *
P4L10: The authors relied on their own cloud classification approach. Since cloud masks are provided as part of the LEDAPS and LaSRC products albeit having known issues, the authors could explain and discuss the reason why they favored a custom algorithm, and how this was assessed to deliver more useful images. I did download a number of the LaSRC images used in this study and left wondering why a custom cloud detection algorithm was needed here, especially given the relatively limited number of images and the visual assessment being made to select those less of not affected by clouds at all.

**As mentioned above, the provided cloud masks are known to have several limitations, especially concerning bright surfaces such as snow and ice. Furthermore, based on our assessment the provided masks usually strongly misclassify medial moraines and lateral debris along glaciers as clouds too. We thus used Spectral Angle Mapper as an independent classification algorithm for cloud classification. To justify our reasoning, we added some clarifying statement in the manuscript.**

"As cloud masks provided with the science products are known to have certain limitations, in particular for bright targets such as snow and ice, but also misclassified medial and lateral moraines, we used a semi-automatic classification approach based on the Spectral Angle Mapper (SAM, [*Kruse et al.*, 1993]) implemented in ENVI to detect and delineate clouds obscuring the glacier surfaces."
* * *
P4L16: the citation should be Liang (2001), not 2000, same in P19

**Changed in all places.**
* * *
P4L18: It would be desirable to use the same notation as Liang (2001) with alpha i used to represent narrowband ground reflectance of TM/ETM+ in band i.

**Changed.**
* * *
P4L19: the symbol b n is not the spectral band number but should be the narrowband ground reflectance of TM/ETM+ in band i. As commented above, using alpha would make it consistent with Liang (2001) and general understanding of the quantity used.

**Changed.**
* * *
P4L22: The narrow to broadband conversion assumes ground reflectance on horizontal surfaces. The LEDAPS and LaSRC products account for topography only in terms of the control of elevation on atmospheric effects, yet the control of topography on the modulation of irradiance on varying slope and aspects, hence on the measured satellite radiance in mountainous regions, is ignored. In other words, the surface reflectance products assume a flat surface and the relative geometry between irradiance, the target, and observation direction is not accounted for, let alone the higher orders of effects related to the topography such as terrain reflected irradiance and modulation of the observed reflectance by the BRDF of the target under the said geometry. Although this effect may be limited on relatively flat glacier tongues and/or when considering only variations rather than absolute values of albedo through time, this and the disconnect between the variable topographic setting under study and the relevance of the empirical formulation of the albedo cannot just be ignored by the authors.

**We agree that the fact of the missing topographic correction in the Landsat science products LEDAPS and LaSRC must be mentioned. This is now done (see answer to comment above). Moreover, we extended our statements**

about the uncertainties stemming from the input data itself and about the impact of neglecting BRDF based on the investigations from *Naegeli et al.* [2017b] in the uncertainty assessment.

"Our results are subject to uncertainties arising from errors in the input data, the general data processing, the albedo retrieval approach and the availability of data as well as environmental factors. In general, the used input data, the Landsat Surface Reflectance Level-2 science products for Landsat 5 and 7 (TM/ETM+) and 8 (OLI) (see Section 2), are Tier 1 products offered by and suggested to be used for time series analysis at pixel level by the USGS. These data are geo-referenced with ≤12 m radial root-mean-square error and intercalibrated across the different Landsat sensors [*Young et al.*, 2017]. Major drawbacks of these data are the missing topographic correction on the radiometry [*Young et al.*, 2017], the saturation problem over snow-covered areas in the TM and ETM+ data and the SLC failure in the ETM+ data post May 2003 resulting in missing data. While the latter is negligible due to the rather small areas studied (see also Section 4.1), the former are of minor impact as only bare-ice areas situated in rather flat terrain on glacier tongues are considered for the analysis of temporal albedo evolution in this study. The retrieval of albedo values from the reflectance products is limited by the availability of spectral information of the input data. The application of a narrow-to-broadband equation (Equation 1) is known to perform reliably in general and over glacierised areas in particular as outlined by different studies [*Knap et al.*, 1999; *Liang*, 2001; *Greuell et al.*, 2002]. Moreover, the impact of a missing Bidirectional Reflectance Distribution Function (BRDF) correction scheme is negligible, but generally results in a slight underestimation of albedo values [*Naegeli et al.*, 2017]. Overall, the uncertainties stemming from the input data, the general data processing and the albedo retrieval approach are hard to quantify and, hence, no exact number is given here. However, as this study focuses on relative changes of albedo rather than absolute values, the conducted analyses based on the given input data can be considered as reliable and robust."
* * *
P4L22: I find the claim that the albedo products are of "very high accuracy" and "deviate by less than <0.001 on average from more sophisticated approach" not well informed, if not misleading. It suggests that albedo values retrieved by the empirical method, on the basis of atmospherically but not topographically corrected data may be 100 times more accurate than reference albedo estimated from an albedometer. I can find the sentence in Naegeli et al. (2017) claiming such "accuracy". I however understand this is the mean difference between Liang albedo applied to synthetic APEX data compared to a rigorous albedo derived from the full spectrum. At least in this case the source of data is all APEX and thus topographically corrected. All in Naegeli et al. (2017) suggest that albedo derived from L8 can hardly meet this target even on average. If to comment on the accuracy of this formula (and provided the limitations associated with the use of non-topographically corrected data and other sources of uncertainty is justified), a more useful number would rather be the standard error of this comparison between APEX and APEXLiang, namely 0.11. This is far more realistic with what can reasonably be expected from albedo retrieval from satellite.

We agree that the statements made so far are not sufficient enough to clarify the performance of the narrow-to-broadband (NTB) formula by *Liang* [2001] to the reader. We thus extended the explanations about the NTB formula performance, based on the detailed assessments made by *Naegeli et al.* [2017], and incorporated them into the new written sub-section "3.5 Uncertainty assessment" at the end of the methods section. Please see also answer given to the comment just above.

However, concerning the referee's suggestion to use "the standard error of this comparison between APEX and APEXLiang, namely 0.11", we note that this seems to have been a misunderstanding. The number 0.11 refers to the standard deviation within this albedo product and, thus, not a comparison between APEX and APEX$_{Liang}$. Experiment 3 in *Naegeli et al.* [2017] investigates the comparison between APEX and APEX$_{Liang}$, and reports a mean glacier-wide albedo of 0.41 ± 0.17 for L8 and 0.41 ± 0.18 for L8$_{Liang}$ for Findelengletscher and 0.15 ± 0.03 for L8 and 0.17 ± 0.09 for L8$_{Liang}$ for Glacier de la Plaine Morte. Moreover, we refer to the discussion and conclusion made in *Naegeli et al.*, [2017] about the performance of the NTB formula by Liang applied to Landsat 8 data: "From Experiment 3 (Table 3) it becomes evident that the *Liang* [2001] formula provides good estimates for glacier-wide mean albedo for both glaciers and all datasets, whereas the *Knap et al.* [1999] formula is subject to stronger deviations."
* * *
P4L25: the definition of such hard threshold on albedo derived from nontopographically corrected data is disputable. Again and related to my previous comment, this should be reviewed and/or critically discussed in view of variability of albedo associated with the topographic control on irradiance.

**We agree that a hard threshold is questionable. However, as our surface type evaluation is a multi-step classification, the initially set threshold has a limited effect on the final result as it only provides a zero-order classification that is afterwards adjusted for grid cells for which the surface type is unclear.**
* * *
P4L9: "(0.25<alpha<0.55)" not "(0.25>alpha<0.55)"

**Changed.**
* * *
P7L9: snapshot is singular

**Changed.**
* * *
P7L10: The use of "ideal case" is value-laden. This sentence should be rephrased more objectively.

**Replaced "in an ideal case" with "at most".**
* * *
P8L2: The authors seem to consider equally epochs one (sometimes two) year(s) apart and sometimes just a few days apart to compute trends. I don't think this is an acceptable methodology. When several images are available for the same summer on a grid cell, I think only one estimate should prevail (or an average maybe) for this year and trends only derived from a "yearly" record. This also applies to the linear regression through all available data points. Note also that it is recommended that the Mann-Kendall test be conducted with only one data point per time period (see Chapter 12 in Helsel and Hirsch, Statistical Methods in Water Resources, U.S. Geological Survey, Techniques of Water-Resources Investigations Book 4, Chapter A3).

**Thank you for this valuable comment. We admit having used multiple scenes within the same year for three instances. To account for this statistical mistake, we've now selected one scene per year only based on optimal quality (largest bare-ice area exposed) and derived trends from "yearly" records as suggested by the reviewer.**

"Over the study period 1999 to 2016, one end-of-summer Landsat snapshot was available for 15 years (cf. Table 2). Unfortunately, in three years no end-of-summer scene is available due to obscureness of clouds. Thus, at most the albedo trend of an individual grid cell is characterized by 15 end-of-summer albedo values."
* * *
P8L14: the areas of nodata in ETM+ are not to be referred as striping (which is usually associated with radiometric calibration) but the Side-line-correction failure that occurred in May 2003.

**Rephrased.**

"Missing data in some of the Landsat ETM+ data, generated due to the scan line corrector (SLC) failure post May 2003, also occurs in our albedo retrievals (e.g. 08.09.2004 in Figure 3)."
* * *
P8L15: the 2004 image seems to be 8/9/2004, not 9/9/2004. All dates in Figure 3 must also be double-checked for erroneous date reported in Table 2 as stressed in a later comment.

Thanks for the careful review of this information. We apologize for the incorrect reporting of the respective scene dates. We double-checked, and changed where necessary, all dates in Table 2 and 4, Figure 3 and the entire manuscript.
* * *
P8L16: The authors acknowledge cloud shadow exemplified by one image in the upper accumulation area. They then claim that "bare-ice area is almost always well represented and inferred albedo is realistic thus allowing monitoring through time". I find this claim particularly vague and not supported. What do the authors mean by "realistic"? In what sense would realistic "allow" monitoring? Just looking at both 1999 ETM+ images on Aletsch glacier, the 10/08 is unusable due to cloud cover on most glaciers of interest and in particular Aletsch, while the 11/09 image shows Aletsch tongue severely impacted by cloud shadow, not even to mention Fiescher glacier. Right from this first date, the quality of the Image begs the question about how much consideration was given to the "realistic" retrieval of albedo. Beyond the inherent accuracy of the albedo retrieval method which I think is misrepresented here, there is also no consideration of factors such as the different radiometric quality of TM/ETM vs OLI (whiskbroom vs pushbroom, 8b vs 12b radiometric resolution) or imperfect coregistration between image, and the potential of these factors on the quality of the albedo signal.
To me the accuracy of the albedo retrieval method once all source of uncertainties are considered is the main factor that must be given consideration before seeking to make inferences on changes. So far and despite what the authors may suggest earlier, the albedo retrieval for a single pixel can hardly be proven to perform at better than 0.1 accuracy if not worst. A discussion of all environmental factors that may even degrade this further would be welcome. For example how cloud shadow as well as topographic shading whether in the upper reach or on the tongue of glaciers is handled by the methodology remains far too obscure. Yet the relative share of any of such phenomena on some dates could potentially result in variation of albedo. Furthermore, obviously the average bare-ice-albedo are computed on varying areas depending on the classification of the snow/ice limit. Naegeli et al. (2017) themselves reported on substantial variation in albedo up the tongue of Findelengletscher. This should be put into perspective with a mean albedo obtained from a variable number of "bare-ice" pixels at different years.

We agree that the treatment of cloud/topographic shadows in our analyses has not been explained clearly enough. To account for very low signals in the input data and thus unrealistic albedos, we generally excluded albedo values < 0.05. However, we now also applied a Spectral Angle Mapper (SAM) classification for cloud and topographic shadows and calculated, similarly as the cloud coverage column in Table 2, a shadow percentage per scene. This helped to eliminate albedo values that are affected by shadows and, thus, should not be included in the analyses.

"Likewise, SAM was used to obtain shadow masks for each individual scene to exclude grid cells that are affected by cloud or topographic shadow effects. Except for the scene taken on the 09th of September 2013, when about 14% of the study area was cloud-covered (north-east part of the study area), the cloud coverage was generally smaller than 5% (Table 2). Cloud and topographic shadows were identified up to about 8% of the study area at maximum (28th of September 2014, Table 2)."

Regarding the specific sensor differences, we note that, in particular the differences between Landsat TM/ETM+ and OLI, are incorporated in the respective surface reflectance retrieval algorithms LEDAPS and LaSRC. Furthermore, the used science products are Tier 1 products offered by the USGS. They are, according to the USGS, suitable for time-series analysis with well-characterized radiometry. Moreover, Tier 1 data are geo-referenced with ≤12 m root-mean-square error and inter-calibrated across the different Landsat sensors. We added this information in the revised manuscript within the revised section "uncertainty assessment". Please see the respective paragraph in an answer further above.

The analysis of mean bare-ice albedo over time is based on varying spatial extents per glacier indeed. We performed a similar analysis based on a constant spatial extent, the lowermost third, per glacier. This analysis revealed highly similar results as the data reported so far, i.e. no significant trends for neither the mean bare-ice albedo time series per glacier nor averaged over all 39 glaciers.
* * *
P10L5: In the context of the relatively large uncertainties being involved in the albedo estimates and interpretation of Table 3 and Figure 5, I find the phrasing that "trends were significant yet at low level" rather misleading. When focusing on those pixels with trends significant at 80% confidence (most of the bare ice), Table 3 reveals a symmetrical distribution (arguably Gaussian) of occurrences exhibiting either positive or equally negative trends. The trends themselves are estimated within the .05/decades or .1 overall magnitude over the approximate 20 years of the study; in other word, barely what we could hope as the uncertainty of the albedo retrieval method. To me this is rather showing that the level of detection (or signal to noise ratio) is simply not enough to be conclusive. Presenting this results and Figure 5 suggesting there are trends significant at least at 80% level everywhere is to me contrary to an alternate interpretation being that this level of detection is not suitable to evidence any obvious trend at all. I think it would be fairer that the interpretation of this result and its significance stress the limitations of the methods rather than suggesting that there are indeed trends, potentially driven by some physical cause. At this stage, my interpretation of the results given the proposed methodology is simply that it can only be inconclusive.

We agree with the reviewer's comment that some of the uncertainties were not fully stated and quantified. We revised the section "uncertainty assessment" and listed uncertainties in more detail (please see paragraph above). However, the calculated significant trends, even though over large areas of the glacier only at a low significance level, are based on a robust methodology and cannot be judged, neither as conclusive nor inconclusive. Moreover, except for the snap-shot uncertainty, most of the uncertainties can not be directly quantified and, thus, no exact numbers are given.

To test the robustness of our trends particularly concerning the use of one end-of-summer scene only, we randomly perturbed all grid-cell albedo values with estimated snap-shot uncertainties (σ = 0.026). The analysis revealed that all trends persist (see last paragraph of Section 3.5). To account for the reviewer's comment, we repeated this analysis with arbitrarily chosen higher uncertainty levels (σ = 0.05 and σ = 0.1), accounting for unquantified uncertainties, and re-calculated point-based albedo trends. This new analysis revealed that the distribution pattern of long-term point-based albedo trends persist and at the 95% confidence level or higher still 60-70% of all grid-cells show negative albedo trends (compared to 77% and 83% if perturbed with σ = 0.026 or no perturbation). We thus keep the presentation of trends and their significance level in the results section. The additional analysis with arbitrarily increased uncertainties is not included as such in the manuscript as we are lacking a basis for actually quantifying them. Nevertheless, we mention that inferred trends remain robust even if assumed uncertainties are substantially higher than just the value for the snap-shot uncertainty.

"Moreover, trends in local bare-ice albedo remained robust even if assumed uncertainties were chosen substantially higher than just the value for snap-shot uncertainty."
* * *
Turning now to the small portions of pixels exhibiting high confidence (95%) of a trend. In this case, the negative trend apparently prevails. Simply looking at the spatial patterns of occurrences of those pixels in Figure 5 and 6 exemplifies what could be expected of the redistribution and spatial variation of debris and medial moraine on most glaciers. A clear example of this is Gorner glacier. Comparing the 11/9/1999 image to recent 25/9/2013 or 30/8/2015 immediately reveals that all those areas of "highly" significant darkening actually don't qualify as "bare-ice" but rather occur mostly due to spatial variation in the distribution of debris. Visually, there is an obvious widening of the medial moraine on the main trunk, retreat of the glacier front and that of tributaries that are exposing rocks, and obvious down wasting with lateral moraines material falling on the glacier. The fact this drives some pixels to appear as exhibiting a strong decrease in albedo is in fact not so much reliant on the analysis presented in the paper to be revealed. More importantly, I am concerned that the paper may suggest a darkening of "bare ice" as if this was a subtle trend associated with increasing concentration of LAI on glaciers when the areas where the changes occur appear mostly to be those exhibiting a step-change in surface type altogether. In view of this, the way the authors elaborate in P10L10 on probable causes associated with such a progressive darkening such as the growth of algae and bacteria is to me far too speculative at this stage, and finally not supported by any new data in the present work.

As outlined above, we delineated and excluded medial moraines and areas where tributaries separated from the main glacier trunk and debris has become exposed.
Moreover, as mentioned in several other answers, the importance of debris redistribution and thus possible causes of bare-ice darkening  is discussed more critically in a new sub-section of the discussion section.

\-\-\-\-\-\-\-\-\-\-\-\-\-\-\-\-\-\-\-\-\-\-\-\-\-\-\-\-\-\-\-\-\-\-\-\-\-\-\-\-\-\-\-\-\-\-\-\-

P10L12: Further to my comments above, I find that the authors far underestimate or seem to lower the role of the debris and changes in the distribution of moraine material in their observations of albedo changes. Based on my visual interpretation of the images used by the authors, it is obvious that there is more than a mere "possibility" than the areas of significant changes are associated to increase in debris cover and changes in medial moraines. I find the authors suggesting that this may only affect "certain grid cells" not supported by observations.

**Please see our answers about this topic further above.**

\-\-\-\-\-\-\-\-\-\-\-\-\-\-\-\-\-\-\-\-\-\-\-\-\-\-\-\-\-\-\-\-\-\-\-\-\-\-\-\-\-\-\-\-\-\-\-\-

P12L5: The authors claim that assessing the uncertainties associated with the albedo estimate is beyond the scope of this work. I don't think this is acceptable and as stressed several times in my comments, I believe this and would require a far more thorough consideration of the uncertainties than presently offered, for any inferences being made about processes at work to be deemed robust.

**To account for uncertainties in the retrieved albedo values in a clearer way, we moved the section "uncertainty analysis" from the discussion section to the methods section. Moreover, we added detailed information about the datasets used in the data section (see answers above).**

\-\-\-\-\-\-\-\-\-\-\-\-\-\-\-\-\-\-\-\-\-\-\-\-\-\-\-\-\-\-\-\-\-\-\-\-\-\-\-\-\-\-\-\-\-\-\-\-

P12L8: What do the authors mean by "better result". This is unspecific, value-laden, and should be unsupported by a stronger argument.

**Rephrased.**

"Moreover, it became clear, that in contrast to Landsat TM and ETM+ data, for which a saturation problem over snow-covered areas exists, the most recent Landsat OLI data has a higher quality and albedo values can also be retrieved for snow."

\-\-\-\-\-\-\-\-\-\-\-\-\-\-\-\-\-\-\-\-\-\-\-\-\-\-\-\-\-\-\-\-\-\-\-\-\-\-\-\-\-\-\-\-\-\-\-\-

P12L9: As demonstrated above, this study does NOT "focus on bare-ice areas" only. Furthermore, the data quality issue (e.g., SLC off, cloud shadows, saturation) can severely affect that albedo retrieval.

**We clarified our definition of bare ice in the introduction. Furthermore, as elaborated in several previous answers, we adjusted our debris exclusion, added information about the impact of SLC-off, cloud shadows and saturation issues at several places in the manuscript.**

"We focus on the bare-ice areas, defined as glacier surfaces neither covered by snow nor by thick debris, only. Snow- or debris-covered glacier surfaces affect glacier mass balance by different processes and are thus not of interest to our analysis."

\-\-\-\-\-\-\-\-\-\-\-\-\-\-\-\-\-\-\-\-\-\-\-\-\-\-\-\-\-\-\-\-\-\-\-\-\-\-\-\-\-\-\-\-\-\-\-\-

P12L13: The authors claim that "manual checks" revealed low frequency of misclassified pixels compromising the albedo retrieval. My own check alone on the first two dates revealed immediately that the biggest glacier in the study (Aletsch) is severely affected by clouds and cloud shadow to the extent that I don't believe a realistic estimate of albedo could have been obtained across many parts of the glacier tongue. In view of this, one cannot be satisfied by the unsupported claim of the authors.

**We agree that our statement was too vague. Actually, very low albedo values ($\alpha < 0.05$) were generally excluded from the analysis. This already minimized the incorporation of grid cells affected by cloud/topographic shadows.**

Furthermore, we applied a SAM classification for cloud/topographic shadows to eliminate grid cells affected by cloud/topographic shadows before the albedo retrieval. We updated Table 2 to show details about this analysis.
* * *
P13L1: While it is true that using 2016 outlines of glaciers in the context of the current glacier demise would have reduced the dominance of ground becoming exposed dominating the change in albedo at the terminus and lateral moraines, Figure 5 and 6 still reveal that the significant changes in albedo remain associated to areas of probably thick debris deposition. It begs the question about what would have been the signal and conclusions of this work if the analysis had focus on a (conservative) mask of bare-ice, meaning a mask where pixels throughout the study period can be observed as free of thick debris. I think this is the main methodological issue that the authors should address to revise this work.

As outlined in previous comments, we manually generated a mask of supraglacial debris with a relevant thickness that was used to exclude these grid cells from all consecutive analyses. However, we note that a clear distinction between debris-covered and dirty ice does not exist and that the transition between the two is smooth and often unclear.
* * *
P13L5: what does "most scenes" means in this context? This sentence is ambiguous and should be clarified.

Findelen is located within two different Landsat scenes per overpass, thus more data is available for Findelen than e.g. Aletsch. Clarified in the manuscript.

"The analysis was performed for one glacier, Findelen, as more scenes were available for this glacier due to the overlapping coverage of this glacier by two different Landsat scenes (path/row 194/28 and 195/28)."
* * *
P13L8: This and later paragraphs relate to a specific methodology and analysis that should be introduced earlier. The steps taken to assess uncertainty should be integral of the research design and reporting of results.

To account for this comment, we moved the entire section concerning the uncertainty analysis to the end of the methods section.
* * *
P13L11: The authors stated in the previous paragraph that the uncertainty analysis was performed on Findelen glacier, but now indicate that the 39 glaciers were considered. Please clarify.

The uncertainty analysis was in fact only performed for one glacier, Findelen, due to the greater data availability (Findelen is located in two Landsat footprints, see answer above). However, the uncertainty estimates were taken to be representative for all glaciers. We added a respective statement in the manuscript.

"Assuming that bare-ice albedo remains constant over this short time period in reality, this value provides a direct uncertainty estimate for local satellite-retrieved albedo that is assumed to be representative for all investigated glaciers in this study."
* * *
P14L1: The repeatability of mean albedo determination on (supposedly) bare-ice areas of Findelen glacier is reported in Table 4, leading to an assessment of uncertainty claimed by the authors on the albedo (pixel-wise) being 0.026. It should be clearly indicated that this reports on the precision (repeatability) of the albedo retrieval only, not its accuracy. I also note that 0.026 is obtained by simply averaging the four estimates corresponding to each year. I am not convinced this simple averaging is providing a fair assessment of the repeatability that can be obtained from this approach. As highlighted in my comment on Table 4, I have concerns about the very small value reported for 2016 and it would contribute to lower the perceived precision. I could not find images in table

4 for 2014 so it leave only two other instances in 2013 and 2015 together suggesting quite a substantially larger precision than 0.026.

Our uncertainty analysis indeed estimates the snap-shot uncertainty and, hence, the seasonal/temporal variability and robustness of using only one scene per year. It clearly shows that the closer to each other scenes are acquired, such as in 2016, the lower the snap-shot uncertainty. This emphasizes that there can be substantial temporal variations of bare-ice albedo within one ablation season. However, as we only make use of scenes acquired in August or September (end-of-summer scenes) this uncertainty remains relatively small for our approach of analysing temporal changes.

Regarding the referee's concern of the very small value for 2016 that might lower the perceived precision, we argue that if only taking years 2014 and 2015, when about a month lies between the acquisition dates and a medium number of pixels were snow-free, a similar overall snap-shot uncertainty of local albedo of 0.027 is found. The computed average uncertainty of local albedo of 0.026 is thus considered to be representative for end-of-summer albedo snap-shot uncertainty within the frame of this study.
* * *
P14L1: The fact that the assessment of trends does not appear to change in view of the perturbations is not truly surprising. As discussed above, I don't think the methodology and level of detection is conclusive enough to elaborate on areas exhibiting trends at the 80% CI in the context of the current methodology. It would be more informative to test how this level of significance changes when increasing the perturbation on albedo to more realistic uncertainty levels given all other environmental factors. The area exhibiting most change, generally a strong albedo decline are apparently greatly controlled by a redistribution of debris and moraines, hence does not qualify as bare-ice throughout the study period. It is expected that those areas remain confidently detected as areas of strong change in albedo. In conclusion, I cannot agree with the author's statement that the "inferred trends in local bare-ice albedo are considered to be robust despite the uncertainty in the albedo retrieval".

Following our answer above, the calculated trends, also with albedo values randomly perturbed with higher estimated uncertainties, are based on a robust methodology. Again, we agree with the referee that not all uncertainties were properly considered and discussed so far. In our revised manuscript, we expanded the section uncertainty assessment considerably. As mentioned above, we repeated the checks of the robustness of the obtained trends with higher uncertainty levels and can state that the trends are, even though not very strong, robust. We also added a respective statement in the conclusions section. Moreover, some areas with strongly negative albedo changes cleared away due to the delineation of medial moraines as well as areas where tributaries separated from the main glacier trunk and debris has become exposed.

"The presented study is subject to various uncertainties stemming from the input data itself, its processing and availability or the albedo retrieval approach. However, unfortunately most of them are hard to numeralise. Nevertheless, our uncertainty assessment revealed highly similar trend patterns and thus indicated the robustness of the obtained albedo trends. We like to emphasize the importance of the snap-short uncertainty --- limited numbers of end-of-summer scenes demand recognition. Specifically, the meteorological conditions preceding the acquisition of the satellite data can influence albedo, e.g. summer snow fall events, and so should be taken into account."
* * *
P14L11: the expression "snap-shot uncertainty" is vague and unspecific. I recommend that this unfamiliar and uncommon wording is revisited to express more plainly what uncertainty the authors are referring to.

We agree that there was no clear definition of snap-shot uncertainty given in the manuscript. We thus added a clarifying statement in the uncertainty analysis section.

"To account for the uncertainty introduced by the use of one end-of-summer scene only and thus the exclusion of sub-seasonal variability in albedo, the snap-shot uncertainty, we performed a comprehensive uncertainty analysis based on ten end-of summer Landsat 8 scenes acquired between 2013 and 2016 (Table 4)."
* * *
P15L17-24: This far-reaching interpretation is not supported by any tangible results presented in present study. Presented as it is now and provided some of the weaknesses of the methodology, this equates more to a relatively general hypothesis rather than one directly informed and supported by the data and results presented here.

**We agree that the added value of this analysis was not clearly stated, and the interpretation as presented maybe reached too far and was based on rather weak/loose statements. However, we note that the analysis is based on valuable data and methodology and adds an interesting and reasonable component to the discussion of possible causes and dependencies of the observed bare-ice albedo changes as a whole. We thus strengthened our motivation to include this analysis in the introduction and rephrased the discussion and interpretation of its results under causes and dependencies in the discussion section in the revised manuscript.**

"We examine trends and their significance to better quantify and investigate a possible darkening of glacier ice in the western and southern Swiss Alps from the point to the regional scale. Causes and external factors that might impact bare-ice albedo and explain its spatial and temporal evolution are discussed."
* * *
P17L9: I believe the authors mean "snap-shot" and not "snap-short" yet I maintain that the use of this "new" terminology is not specific enough to make it informative of a clear source of uncertainty, and hence would advise against the use of it.

**As explained in the answer above, we clarified the definition of snap-shot uncertainty. We corrected the typing error.**
* * *
P17L11: The authors insist in their conclusion that "meteorological conditions preceding the acquisition (. . .) need to be considered". I however saw no such consideration in the manuscript despite most images exhibiting various stages of snowline retreat, some of them with obvious short-lived snowfall.

**We admit that the statement wasn't properly formulated, and thus rephrased the respective section. However, in our study we examined the data, in particular concerning fresh summer snow on the glacier surfaces.**

"Specifically, the meteorological conditions preceding the acquisition of the satellite data can influence albedo, e.g. summer snow fall events, and so should be taken into account."
* * *
P17L12: I am actually quite concerned that one of the main conclusion point is that "highly significant darkening" affect about 10% of the ablation areas. Based on my own interpretation of many of the images used, much of this darkening can be attributed to change in surface type from bare-ice to debris, rather than a "darkening" of bare-ice via the accumulation of LAI for example. To me the point raised by this study rather stresses the potential accumulation of debris on the lower reaches of mountain glaciers in the context of retreat.

**Based on the exclusion of supraglacial debris, by manually delineating medial moraines and areas where tributaries separated from the main glacier trunk and debris has become exposed, from our analyses, our conclusions focus more clearly on bare-ice and are thus not misleading. However, we would like to mention again that is generally difficult to clearly disentangle thick debris from Light Absorbing Impurities (LAI) coverage, especially in relation to possible darkening phenomena.**
* * *
Table 2: Landsat 7 sensor should be named ETM+

**Changed.**
* * *
Table 2: Although it is understandable that rounding may bring variations in numbers at the decimal level, it would be preferable that in 2008 when 0 km2 cloud is reported, this correspond to 0% as well.

**We agree that the reported, rounded number is somewhat misleading. Changed.**
* * *
Table 2: Some of the information reported in the table 2 appears to be incorrect. I could not find images for quite a few dates: I suggest 8/9/2004 (not 9/9/2004); 22/9/2006 (not 20/09/2006); 27/08/2014 (not 26/08/2014); 12/09/2014 or 28/09/2014 (not 1/9/2014 and 27/09 /2014). The sensor in 9/9/2013 appears to be L8 and not L7. This also affects Table 4.

**Thanks for this careful review. We adjusted the information in Table 2 accordingly.**
* * *
Table 2: I am perplex about the reporting of clouds in 25/9/2013 and the reason for reliance on a secondary date 9/9/2013. I downloaded both dates and it appears that clouds in 25/9/2013 indeed marginally affect the east of area (d), however barely above the terminus of Bachi and Minsti glaciers, both outside the scope of this study. I could not see anywhere else where clouds may have caused an issue and the need to rely on an alternate date. This begs the question about the performance of the cloud classification algorithm used by the authors. It is even more confusing since the 9/9/2013 image (L8 and not L7 as reported by the authors) is obviously far worse with many clouds and the fact that the authors specifically mention this image in P4L13. Looking at both dates, it is obvious that the snowline retreated over the period, thus exposing more of the bare ice, at least for those glaciers not obscured by clouds and as reported in Figure 3. It suggests the use of two dates in this instance may be done to retrieve albedo over larger areas of bare ice, at least on some glaciers, but the reader can only wonder. Even in this case, it would beg the question about the consistency of the data being used and the potential effect on mean albedo. Table 4 shows that several scenes are used for assessing uncertainty, but also shows an inconsistency with table 2 as multiple acquisitions in 2013 and 2014 are consistent, while those in 2015 and2016 are not reported in Table 2.

**We agree that the there is a need to better clarify the choice of the scene used in our analysis for the year 2013. The later date (25. September) is considerably less affected by clouds, however, fresh snow seems to be present on many glaciers in the southern part of the study area, but also on Glacier de la Plaine Morte in the northern part for example. In contrast, the early date (09. September) is, as mentioned correctly by the reviewer, more affected by clouds, especially in the Aletschgletscher region. We thus decided for each glacier individually which end-of-summer scene was more appropriate for 2013. This is now explained and clarified in the manuscript.**

"All 39 glaciers are comprised in one scene (path 195, row 28) and we examined a total of 16 scenes between the years 1999 and 2016, whereupon in 2013 one scene was chosen per glacier individually (Table 2)."

"*For each individual glacier only one of these two scenes in 2013 is taken (based on minimal cloud and/or snow coverage)." (Footnote for Table 2)

**Concerning the application of our own cloud classification algorithm, please see the answer to a comment further up. Similarly, we clarified the inconsistency that you mentioned between Table 2 and 4 (see also above).**
* * *
Figure 2: "snowline altitude" not "snowline altitdue"

**Changed.**
* * *
Figure 2: what are SLAconst and rcrit?

We apologize to have missed the explanation of both parameters and thus added some explanation in the manuscript.

"Within this range of albedo values, outliers are suppressed by adjusting all albedo values ($\alpha_{corr}$) by multiplying with a constant value ($SLA_{const}$)."

"Finally, all grid cells are evaluated regarding their relative position compared to the SLA within a critical radius $r_{crit}$ (*probability test to eliminate extreme outliers*, Figure 2)."
* * *
Table 4: I could not find any image on 1/09/2014 nor 27/09/2014.

The given dates were incorrect – sorry! Changed.
* * *
Table 4: In 2016, the number of pixels used for the assessment is 5495, thus representing 5km2 of supposedly bare-ice surface. I obtained both images and could only map 3km2 of ice at the most, the rest being mostly the accumulation area still obviously covered by snow. Note also that the lower part of the glacier tongue is severely affected by cloud shadow on 1/9/2016. Beside an issue of size being considered as bare-ice which I can't reconciliate with what the images depicts and shedding doubt on the performance of the snow/ice classification, the albedo variability appears surprisingly small (0.008) when any other years yield about five times larger albedo precision. I believe some clarification is required here.

We double-checked the bare-ice classification based on our surface type evaluation for both scenes used in the uncertainty analysis for the year 2016. It revealed that for the earlier date (25.08.2016) about 5.4 km$^2$ were mapped as bare ice with a maximum albedo of 0.315. Similarly, for the later date (01.09.2016) also about 5.4 km$^2$ of bare ice were mapped with a maximum albedo of 0.295. We thus are not sure how the referee's estimate of only about 3 km$^2$ has been obtained.

The low albedo variability between the two scenes used in the snap-shot uncertainty analysis for the year 2016 results from the fact that the two scenes are only acquired seven days apart. In contrast, the scenes used in 2013, 2014 and 2015 were 16, 32 and 32 days apart, respectively. If the obtained standard deviations per year are normalized by the time period in between the acquisition of the individual scenes used per year (in days), highly similar (0.001 for the years 2014, 2015 and 2016) and slightly higher (0.003 for the year 2013) in-between scenes albedo variability are obtained. The slightly higher normalized standard deviation for the snap-shot uncertainty in year 2013 most likely results from the small bare-ice area available (1190 pixels) to perform the analysis.

\begin{abstract}

[revised manuscript text omitted]

\footnotesize

\centering

\renewcommand{\arraystretch}{0.92}

\begin{tabular}{lcccccc}%{\textwidth, column = lcccccc}

\tophline

Glacier Name & Area 2010 &    Area 2016 &    Elev. range    &        Lithology \\
&       (km\textsuperscript{2}) & (km\textsuperscript{2})        & (m a.s.l.)      & \\

```
& (km\textsuperscript{2}) & (m a.s.l.) &   \\

\middlehline

Grosser Aletsch &       78.4 &  74.3 &  1872--4120      &       mica shists, gneiss       \\

Grenzg &        40.2 &  37.3 &  2319--4536      &       mica shists, gneiss      \\

Fiescher (VS) &  29.5 &  28.2 &  2102--4082      &       mica shists, gneiss      \\

Unteraar &       22.5 &  15.9 &  2278--3925      &       granites, syenites      \\

Rhone &          15.3 &  15.0 &  2300--3621      &       granites, syenites      \\

Trift &  14.9 &  14.6 &  2191--3383      &       mica shists, gneiss      \\

Corbassière &   15.2 &  14.2 &  2496--4306      &       calc. phyllites, marly shales \\

Findelen &       14.2 &  13.8 &  2661--3929      &       mica shists, gneiss      \\

Oberaletsch &   17.5 &  12.7 &  2456--3831      &       granites, syenites      \\

Otemma &        12.6 &  11.0 &  2607--3779      &       basic rocks      \\

Kanderfirn &    12.2 &  11.4 &  2408--3203      &       granites, syenites      \\

Zinal &  13.4 &  10.9 &  2466--4035      &       granites, syenites      \\

Gauli & 11.4 &  11.1 &  2335--3597      &       gneiss      \\

Zmutt &          13.7 &  10.1 &  2650--4030      &       amphibolites      \\

Unterer Theodul &        9.4      &       9.1      &       2611--4155      &       basic rocks      \\

Mont Miné &    9.9      &       9.3      &       2403--3719      &       granites, syenites      \\

Allalin   &      9.2      &       8.9      &       2686--4167      &       gneiss  \\

Fiescher (BE) &  9.4      &       8.4      &       1999--4086      &       limestones      \\

Ferpècle &       9.0      &       8.6      &       2289--3659      &       granites, syenites      \\

Oberer Grindelwald       &       8.4      &       8.1      &       1931—3715 & granites, syenites \\

Plaine Morte &  7.3      &       7.5      &       2514--2874      & calc. phyllites, marly shales \\

Lang &  8.3      &       7.2      &       2461--3894      &       mica shists, gneiss      \\

Fee     &       7.3      &       7.1      &       2590--4014      &       basic rocks      \\

Ried &  7.3      &       7.0      &       2400--4247      &       mica shists, gneiss      \\

Obers Ischmeer       &       7.3      &       6.8      &       2107--3880      & gneiss      \\

Saleina &       6.5      &       6.2      &       2320--3863      &       granites, syenites      \\

Brenay  &       7.1      &       6.0      &       2728--3824      &       granites, syenites      \\

Trient &        5.8      &       5.8      &       2160--3477      &       granites, syenites      \\

Mittelaletsch & 6.9      &       5.7      &       2599--4059      &       granites, syenites      \\

Stein & 5.7      &       5.5      &       2190--3462      &       amphibolites      \\
```

Mont Durand & 6.1 & 5.4 & 2769--4102 & mica shists, gneiss \\

Rosenlaui & 5.4 & 5.3 & 2102--3623 & granites, syenites \\

Giètro & 5.2 & 5.1 & 2792--3815 & calc. phyllites, marly shales \\

Turtmann & 5.2 & 5.1 & 2409--4141 & granites, syenites \\

Mont Collon & 5.4 & 5.0 & 2532--3670 & granites, syenites \\

Brunegg & 5.5 & 5.0 & 2715--3796 & granites, syenites \\

Schwarzberg & 5.2 & 4.7 & 2728--3549 & mica shists, gneiss \\

Moming & 5.3 & 4.6 & 2692--4036 & basic rocks \\

Tschingelfirn & 5.2 & 4.4 & 2395--3318 & limestones \\

Ried & 7.1 & 2400--4247 & mica shists, gneiss \\

Obers Ischmeer & 6.9 & 2107--3880 & gneiss \\

Saleina & 6.3 & 2320--3863 & granites, syenites \\

Brenay & 6.2 & 2728--3824 & granites, syenites \\

Trient & 5.8 & 2160--3477 & granites, syenites \\

Mittelaletsch & 5.7 & 2599--4059 & granites, syenites \\

Stein & 5.5 & 2190--3462 & amphibolites \\

Mont Durand & 5.4 & 2769--4102 & mica shists, gneiss \\

Rosenlaui & 5.3 & 2102--3623 & granites, syenites \\

Giètro & 5.1 & 2792--3815 & calc. phyllites, marly shales \\

Turtmann & 5.1 & 2409--4141 & granites, syenites \\

Mont Collon & 5.0 & 2532--3670 & granites, syenites \\

Brunegg & 5.0 & 2715--3796 & granites, syenites \\

Schwarzberg & 4.7 & 2728--3549 & mica shists, gneiss \\

Moming & 4.5 & 2692--4036 & basic rocks \\

Tschingelfirn & 4.5 & 2395--3318 & limestones \\

\bottomhline

\end{tabular}

\label{tab:1}

\belowtable{} % Table Footnotes

\end{table}

\section{Methods}

\subsection{Pre-processing}

All reflectance data was were downloaded through earthexplorer.usgs.gov (Table 2). The final selection of all scenes is based on a visual check. As cloud masks provided with the science products are known to have certain limitations, in particular for bright targets such as snow and ice, but also misclassified medial and lateral moraines, we used a semi-automatic classification approach based on the Spectral Angle Mapper (SAM, \citet{Kruse1993}) implemented in ENVI tTo detect and delineate clouds obscuring the glacier surfaces, we used a semi-automatic classification approach based on the Spectral Angle Mapper (SAM, \citet{Kruse1993}) algorithm implemented in ENVI. For each sensor (TM, ETMETM+, OLI) a spectral library of cloud signatures was manually compiled, which served as reference library for the respective sensor. Hence, for each scene we obtained a cloud mask that was used to exclude cloud-affected pixels from all consecutive analyses. Likewise, SAM

was used to obtain shadow masks for each individual scene to exclude grid cells that are affected by cloud or topographic shadow effects. Except for the scene taken on the 09\textsuperscript{th} of September 2013, when about 2014\% of the bare icestudy area was cloud-covered (north-east part of the study area), the cloud coverage was generally smaller than 5\% over the bare-ice area (Table 2). Cloud and topographic shadows were identified up to about 8\% of the study area at maximum (28\textsuperscript{th} of September 2014, Table 2).

\begin{table*}[h!]

\caption{Overview of Landsat scenes used. The bare-ice area is given in km\textsuperscript{2} and relative to the total study area of 451442\,km\textsuperscript{2}. Cloud and shadow coverage areis also given for bare-ice areas only, and is also evaluated relative to the overall bare-ice area of the respective scene.relative to the total study area.}

\centering

\begin{tabular}{lccccc}%{\textwidth, column = c}

\tophline

Landsat mission &	Date &	\multicolumn{2}{c}{Bare-ice area} &	Clouds & Shadows \\

(sensor) &	(dd.mm.yyyy) &(km\textsuperscript{2}) &	(\%) &	(\%) & (\%) \\

\middlehline

Landsat 7 (ETM+)	&	11.09.1999	&	119.7 &	27.1	&	0.1	&	4.5	\\

Landsat 7 (ETM+)	&	12.08.2000	&	89.7	&	20.3	&	0.5	&	1.4	\\

Landsat 7 (ETM+)	&	18.08.2002	&	61.0	&	13.8	&	1.0	&	3.5	\\

Landsat 5 (TM) &	13.08.2003	&	182.5 &	41.3	&	0.0	&	0.4	\\

Landsat 7 (ETM+)	&	08.09.2004	&	132.0 &	29.9	&	0.0	&	2.5	\\

Landsat 7 (ETM+)	&	10.08.2005	&	72.6	&	16.4	&	5.9	&	1.0	\\

Landsat 5 (TM) &	22.09.2006	&	95.8	&	21.7	&	0.0	&	6.0	\\

Landsat 7 (ETM+)	&	18.08.2008	&	24.6	&	5.6	&	0.1	&	0.8	\\

Landsat 5 (TM) &	30.09.2009	&	185.2 &	41.9	&	0.0	&	7.2	\\

Landsat 7 (ETM+)	&	12.09.2011	&	146.8 &	33.2	&	0.4	&	2.8	\\

Landsat 7 (ETM+)          &          14.09.2012          &          22.1          &          5.0          &          0.8          &          3.7          \\

Landsat 8 (OLI)  &          09.09.2013*          &          \multirow{2}{*}{91.1}          &          \multirow{2}{*}{20.6}          &          14.0          &          4.5          \\

Landsat 8 (OLI)  &          25.09.2013*          &                    &                    &          0.5          &          6.7          \\

Landsat 8 (OLI)  &          28.09.2014          &          153.5          &          34.7          &          1.0          &          8.3          \\

Landsat 8 (OLI)  &          30.08.2015          &          214.9          &          48.6          &          1.4          &          2.1          \\

Landsat 8 (OLI)  &          01.09.2016          &          189.4          &          42.8          &          2.3          &          3.1          \\Landsat mission &          Date &          \multicolumn{2}{c}{Bare-ice area} &          \multicolumn{2}{c}{Cloud coverage} \\

(sensor) &          (dd.mm.yyyy) &(km\textsuperscript{2}) &          (\%) &          (km\textsuperscript{2}) &(\%) \\

\middlehline

Landsat 7 (ETM)          &          10.08.1999          &          51.6          &          11.4          &          2.3          &          4.4          \\

Landsat 7 (ETM)          &          11.09.1999          &          145.8          &          32.3          &          0.8          &          0.6          \\

Landsat 7 (ETM)          &          12.08.2000          &          101.0          &          22.4          &          0.2          &          0.2          \\

Landsat 7 (ETM)          &          18.08.2002          &          80.5          &          17.9          &          0.0          &          0.0          \\

Landsat 5 (TM) &          13.08.2003          &          193.3          &          42.9          &          0.0          &          0.0          \\

Landsat 7 (ETM)          &          09.09.2004          &          145.7          &          32.3          &          0.0          &          0.0          \\

Landsat 7 (ETM)          &          10.08.2005          &          80.8          &          17.9          &          0.4          &          0.5          \\

Landsat 5 (TM) &          20.09.2006          &          122.4          &          27.1          &          0.0          &          0.0          \\

Landsat 7 (ETM)          &          18.08.2008          &          29.4          &          6.5          &          0.0          &          0.1          \\

Landsat 5 (TM) &          30.09.2009          &          221.7          &          49.2          &          0.1          &          0.0          \\

Landsat 7 (ETM)          &          12.09.2011          &          168.0          &          37.2          &          1.0          &          0.6          \\

Landsat 7 (ETM) & 14.09.2012 & 33.1 & 7.3 & 0.0 & 0.0 \\

Landsat 7 (ETM) & 09.09.2013 & 107.7 & 23.9 & 19.5 & 18.1 \\

Landsat 8 (OLI) & 25.09.2013 & 129.3 & 28.7 & 2.1 & 1.7 \\

Landsat 8 (OLI) & 26.08.2014 & 142.5 & 31.6 & 4.6 & 3.2 \\

Landsat 8 (OLI) & 01.09.2014 & 139.1 & 30.8 & 6.3 & 4.6 \\

Landsat 8 (OLI) & 27.09.2014 & 188.6 & 41.8 & 4.5 & 2.4 \\

Landsat 8 (OLI) & 30.08.2015 & 246.8 & 54.7 & 11.2 & 4.5 \\

Landsat 8 (OLI) & 01.09.2016 & 222.9 & 49.4 & 1.3 & 0.6 \\

\bottomhline

\end{tabular}

\label{tab:2}

\belowtable{*For each individual glacier only one of these two scenes in 2013 is taken (based on minimal cloud and/or snow coverage).} % Table Footnotes

\end{table*}

\subsection{Albedo retrieval}

\label{subsec:albedoretrieval}

We applied the narrow-to-broadband conversion by \citet{Liang2000Liang2001} to obtain shortwave broadband albedo $\alpha$\textsubscript{short} from the surface reflectance data. The conversion is based on five of the seven individual bands, and is formulated as follows:

\begin{equation}

\alpha_{short} = 0.356b\alpha_{1} + 0.130\alpha b_{3} + 0.373\alpha b_{4} + 0.085\alpha b_{5} + 0.072\alpha b_{7} - 0.0018

\label{eq:1}

\end{equation}

where \alpha_ b\textsubscript{in} represents the narrowband ground reflectance of TM/ETM+ in band ispectral band number of Landsat TM/ETM. For Landsat OLI, the band numbers were adjusted accordingly. This conversion was developed based on a large empirical data set and the band configurations of Landsat TM/ETMETM+. As shown by \citet{Naegeli2017} this albedo retrieval approach can be applied also to the most recent mission Landsat 8 and is suitable for mountain glaciers. It provides albedo products that are of very high accuracy and deviate by less than < 0.001 on average from a more sophisticated albedo retrieval approach.

\subsection{Surface type evaluation}

The delineation of bare-ice area versus snow-covered surfaces is based on a multi-step classification scheme of the surface albedo values (Figure \ref{fig:2}). The classification is thus based on a physical parameter specific for both snow and ice. In a first step, two- threshold values for \textit{certainly snow} ($\alpha$ > 0.55) and \textit{certainly ice} ($\alpha$ < 0.25) are defined (\textit{primary surface type evaluation}, Figure \ref{fig:2}) based on recommendations in the literature \citep{Cuffey2010}. This results in a critical albedo range (0.25\,<>\,$\alpha$\,<\,0.55), where an unambiguous assignment of the surface type, i.e. snow or ice, is not possible without considering other parameters. Within this range of albedo values, outliers are suppressed by adjusting all albedo values ($\alpha$\textsubscript{corr}) by multiplying with a constant value (SLA\textsubscript{const}). We, therefore, take advantage of a digital elevation model available for all glaciers to evaluate the average albedo in elevation bands of 20 m within this critical albedo range. The transition between ice and snow is typically characterized by a distinct change in albedo \citep[e.g.][]{Hall1987, Winther1993, Zeng1983}. We thus derive an estimate of the mean snowline altitude (SLA) for each glacier and scene based on the greatest slope of the albedo-elevation profile. The albedo for this altitude is considered to be the site- and scene-specific albedo threshold discriminating snow and ice and is henceforth termed $\alpha$\textsubscript{crit}.

\begin{figure}[h!]

%\includegraphics[width=8.3cm]{figures/NEW/SurfaceTypeEvaluation_FlowChart_4-01}

\includegraphics[width=8.3cm]{figures/NEW/Figure_2}

\caption{Flow chart of the methodology applied to evaluate the surface type (snow/ice) of every glacier grid cell based on the derived shortwave broadband albedo. For further explanations please see text.}

\label{fig:2}

\end{figure}

In a second step, we use the SLA and $\alpha$\textsubscript{crit} as reference to evaluate the surface type within the range of critical albedo values, where there is ambiguity between snow and ice (\textit{secondary surface type evaluation}, Figure \ref{fig:2}). Finally, all grid cells are evaluated regarding their relative position compared to the SLA within a critical radius r\textsubscript{crit} (\textit{probability test to eliminate extreme outliers}, Figure \ref{fig:2}). Grid cells located clearly above the SLA are more likely to be snow than ice, and vice versa. An increasing positive/negative

vertical distance from the SLA thus results in penalties for the likelihood of the cell within the critical albedo range of being either snow or ice. As an example, a grid cell near the glacier terminus with an albedo of 0.42, i.e. a rather high albedo for Alpine glacier ice, will be classified as ice. An albedo of e.g. 0.35 observed for the highest regions of the glacier, in contrast, will be classified as snow, as the low albedo is more likely to be explained by an erroneous albedo determination (e.g. shadows) than by actually snow-free conditions. In summary, our procedure to distinguish between snow and bare-ice surfaces relies on remotely-determined surface albedo and merges this information with surface elevation with a probability-based approach to detect outliers and to automatically adapt the classification to the site- and scene-specific conditions.

\subsection{Trend analysis}

Over the study period 1999 to 2016, at least one good end-of-summer Landsat end-of-summer snapshots was available for 15 years (cf. Table \ref{tab:2}). Unfortunately, iIn three years no end-of-summer scene is available due to obscureness of clouds, two or three scenes at the end-of-summer were available. Thus, in an ideal caseat most the albedo trend of an individual grid cell is characterized by 1915 end-of-summer albedo values. However, due to cloud coverage, differing amounts of snow-covered areas in the scenes and/or sensor artefacts, less scenes were usually available to evaluate a temporal bare-ice albedo trend for single grid cells. We arbitrarily set the necessary number of scenes to 50\%, thus at least eightten albedo values are required for calculating the albedo trend of one individual grid cell. We used the non-parametric Mann-Kendall (MK) test \citep{Mann1945, Kendall1975} to evaluate the confidence level of the trends (significant at the 95\% / 90\% / 80\% level, or not significant). For grid cells with significant trends, the magnitude of the trend was determined based on linear regression through all available data points. Trends are given as albedo change per decade. For some analyses, we only use trends significant at the 95\% confidence level for further interpretation and exclude less significant trends.

\subsection{Uncertainty assessment}

[revised manuscript text omitted]

\begin{table*}[h]

\caption{Overview of scenes used in the snap-shot uncertainty analysis. Px refers to the number of pixels that were used to derive uncertainty. Mean ($\alpha$\textsubscript{mean}), minimum ($\alpha$\textsubscript{min}) and maximum ($\alpha$\textsubscript{max}) albedo, as well as the

mean ($\sigma$\textsubscript{mean}) standard deviation of point-based bare-ice albedo for each individual scene pair or triple per year are given.}

\centering

\begin{tabular}{lcccccc}%{\textwidth, column = c}

\tophline

Year     & Day &         Px       & $\alpha$\textsubscript{mean} & $\alpha$\textsubscript{min} & $\alpha$\textsubscript{max} & $\sigma$\textsubscript{mean} \\

\middlehline

\multirow{2}{*}{2013} & 09 Sept. & \multirow{2}{*}{1190} & 0.204 & 0.052 & 0.370 & \multirow{2}{*}{0.040} \\

& 25 Sept. & & 0.233 & 0.051 & 0.361 & \\

\middlehline

\multirow{3}{*}{2014} & 27 Aug.  & \multirow{3}{*}{3869} & 0.213 & 0.051 & 0.382 & \multirow{3}{*}{0.024} \\

& 12 Sept. & & 0.224 & 0.052 & 0.383 & \\

& 28 Sept. & & 0.255 & 0.054 & 0.396 & \\

\middlehline

\multirow{3}{*}{2015} & 07 Aug.  & \multirow{3}{*}{3446} & 0.174 & 0.052 & 0.403 & \multirow{3}{*}{0.031} \\

& 30 Aug.  & & 0.178 & 0.051 & 0.356 & \\

& 08 Sept. & & 0.236 & 0.053 & 0.358 & \\

\middlehline

\multirow{2}{*}{2016} & 25 Aug.  & \multirow{2}{*}{5495} & 0.152 & 0.051 & 0.315 & \multirow{2}{*}{0.008} \\

        & 01 Sept. &

        & 0.156 &

0.051 & 0.295 &

\\

\middlehline

\multicolumn{2}{c}{mean for 2013--2016} & 3500 &

& & & 0.026

\\

\bottomhline

\end{tabular}

\label{tab:4}

\belowtable{} % Table Footnotes

\end{table*}

To assess the impact of local albedo uncertainty on the determination and the robustness of potential temporal trends, we randomly perturbed the distributed bare-ice albedo values of every grid cell and scene, and for all 39 individual glaciers with the computed average uncertainty of local albedo of 0.026 (average pixel number of 3500). The re-evaluation of the long-term albedo trends significant at the 80\% level according to the MK test revealed that they were not affected by the random perturbation of the albedo values. Both a very similar area of the glaciers' bare-ice surfaces and distribution of trend magnitude was found in the perturbed datasets. However, for trends significant at the 95\% confidence level or higher a slightly smaller area (11\,km\textsuperscript{2}) was detected (c.f. Table 3). Within this area, the majority (77\%) of all pixels is affected by negative trends, which is highly similar as obtained by the original albedo datasets (cf. Table \ref{tab:3}). Moreover, trends in local bare-ice albedo remained robust even if assumed uncertainties were chosen substantially higher than just the value for snap-shot uncertainty.

\section{Results}

\subsection{Spatially distributed shortwave broadband albedo}

Figure \ref{fig:3} shows the spatio-temporal evolution of glacier-wide shortwave broadband albedo for Findelengletscher. The retrieval of meaningful albedo values is restricted by the quality of the surface reflectance data and, thus, the availability of realistic values in the individual bands needed for the narrow-to-broadband conversion. For Landsat TM/ETMETM+, a saturation problem over snow-covered areas exists, resulting in missing values for these regions (years 1999--2012 in Figure \ref{fig:3}). This problem is not present in the Landsat 8 data (years 2013--2016 in Figure \ref{fig:3}). The stripingMissing data in some of the Landsat ETMETM+ data, generated due to the scan line corrector (SLC) failure post May 2003, also occurs in our albedo retrievals (e.g 0899.09.2004 in Figure \ref{fig:3}) and produces regions with missing data. We tested the impact of the SLC failure by simulating missing data for three scenes with an intact SLC for Findelengletscher. SLC failure resulted in slightly higher mean bare-ice albedo values (1.2 to 2.2\%, e.g. 12.08.2000 SLC-on mean bare-ice

albedo 0.204 versus SLC-off mean bare-ice albedo 0.209 indicating a difference of 2.2\%), which is a negligible impact. Although, we applied a cloud removal algorithm, our results are still impacted by cloud shadows that are harder to detect without manual effort (e.g. 18.08.2002 in Figure \ref{fig:3}). However, the bare-ice area is almost always well represented and inferred albedo is realistic, hence allowing for a monitoring through time.

[revised manuscript text omitted]

\centering

\begin{tabular}{lccccc}%{\textwidth, column = c}

\tophline

Class     & Albedo trend &         \multicolumn{2}{c}{Confidence level 80\%} & \multicolumn{2}{c}{Confidence level 95\%}  \\

```latex
(\#)      & (albedo change decade\textsuperscript{-1}) & (\%) &   (km\textsuperscript{2}) &        (\%)    &        (km\textsuperscript{2}) \\

\middlehline

1 &      < $-$0.05 &      8.8 &    10.1 &  28.9 &  3.9 \\

2 &      $-$0.05 to $-$0.03 &      11.2 &  12.9 &  28.2 &  3.8 \\

3 &      $-$0.03 to $-$0.01 &      21.5 &  24.6 &  25.6 &  3.5 \\

4 &      $-$0.01 to 0.01 &       26.0 &  29.8 &  1.8 &    0.2 \\

5 &      0.01 to 0.03 &   17.9 &  20.5 &  2.7 &    0.4 \\

6 &      0.03 to 0.05 &   8.4 &   9.7 &    5.8 &    0.8 \\

7 &      > 0.05 &          6.1 &    7.0 &    6.9 &    0.9 \\

\middlehline

Total &  &        100 &    114.5 & 100 &    13.5 \\

1 &      < $ $0.05 &      7.4 &    9.9 &    31.9 &  5.1 \\

2 &      $ $0.05 to $ $0.03 &      11.6 &  15.4 &  30.4 &  4.9 \\

3 &      $ $0.03 to $ $0.01 &      22.6 &  30.1 &  21.1 &  3.4 \\

4 &      $ $0.01 to 0.01 &       26.7 &  35.6 &  1.8 &    0.3 \\

5 &      0.01 to 0.03 &   18.0 &  24.0 &  1.8 &    0.3 \\

6 &      0.03 to 0.05 &   7.9 &   10.5 &  5.3 &    0.8 \\

7 &      > 0.05 &          5.8 &    7.7 &    7.7 &    1.2 \\

\middlehline

Total &  &        100 &    133 &    100 &    16 \\

\bottomhline

\end{tabular}

\label{tab:3}

\belowtable{} % Table Footnotes

\end{table*}

\begin{figure*}[h]

%\includegraphics[width=12cm]{figures/NEW/Trend_Signi_50_plot_new_samescales_NEW}

\includegraphics[width=12cm]{figures/NEW/Figure_5}
```

\caption{Confidence levels of bare-ice albedo trends over the study period 1999 to 2016 according to the MK test. See Figure \ref{fig:1} for the location of the different panels in the Swiss Alps.}

\label{fig:5}

\end{figure*}

\begin{figure*}[h]

%\includegraphics[width=12cm]{figures/NEW/Trend_50_plot_new_7classes_numbers}

\includegraphics[width=12cm]{figures/NEW/Figure_6}

\caption{Classified albedo trends per decade for all grid cells with trends significant at the 80\% confidence level or higher according to the MK test. Averages for areas with bare-ice albedo trends significant at the 95\% confidence level are given for selected large glaciers. See Figure \ref{fig:1} for the location of the different panels in the Swiss Alps.}

\label{fig:6}

\end{figure*}

For most of the bare-ice area, the derived trends in albedo were only significant at low levels. Compared to the glaciers' overall ablation area only relatively few grid cells with trends significant at the 95\% confidence level or higher (dark blue areas in Figure \ref{fig:5}) are present. The cells with significant trends at high confidence levels are usually situated at the termini or along the lower margins of the glaciers and trends are mostly negative (cf. Table \ref{tab:3}, Figures \ref{fig:5}--\ref{fig:7}). The darkening can be attributed to different causes. At the glacier termini, an accumulation of fine debris due to the deposition of allochthonous material and/or melt-out of englacial debris is most likely. These materials, together with the presence of organic material, usually dark and humic substances, decrease local albedo values considerably and foster the growth of algae and bacteria \citep{Hodson2010, Yallop2012, Takeuchi2013, Stibal2017}. However, many of these effects and interactions are still unclear. Along the glacier margins an increase in debris cover due to small collapses or input of morainic material and, hence, a deposition of rather thick debris on the bare-ice is possible. Moreover, the appearance of debris-rich basal ice alongside the lower glacier margins due to the general glacier recession poses a further cause of local darkening \citep{Hubbard1995, Hubbard2009}. Along the central area of the glacier tongue, particularly in the vicinity of medial moraines (e.g. in the case of Gornergletscher, Figure \ref{fig:7}), a strongly negative albedo trend indicates an expanding medial moraine, changing the local area from clean to (partly) debris-covered ice. In contrast, we also find significant positive albedo trends for some locations on the glacier tongues (see Figure \ref{fig:7}). These might be explained by the effect of glacier flow changing the position of the medial moraine, hence leading to a transition from debris-covered to clean ice with a higher albedo for certain grid cells. Lateral shifts of the position of medial moraines are possible for retreating glaciers \citep{Anderson2000}.

The investigation of the lithology surrounding the 39 individual glaciers and their overall albedo trend observed for the study period (Table \ref{tab:1}) revealed that glaciers predominantly surrounded by

less abrasive rocks (calcareous phyllites, limestones and marly shales, CERCHAR Abrasivity Index (CAI) 0--2 after \citet{Kasling2010}) exhibited a stronger negative albedo change of $-$0.05 per decade compared to glaciers that are located in an area of very to extremely abrasive rocks ($-$0.03 albedo change per decade; amphibolites, basic rocks, gneiss, granites, mica shists and syenites, CAI 2--6 after \citep{Kasling2010}).

\begin{figure}[h]

%\includegraphics[width=8.3cm]{figures/NEW/Trend_Signi_50_ge95_plot_new_samescales_ALETSCH_NEW2-01}

\includegraphics[width=8.3cm]{figures/NEW/Figure_7}

\caption{(a) Close-up of bare-ice albedo trends per decade significant at the 95\% confidence level or higher for the tongue of Aletsch, and (b) time-series of bare-ice albedo between 1999 and 2016 for ten randomly selected points on the terminus (crosses in (a)) including a linear fit (dashed purple, r = $-$0.6).}

\label{fig:7}

\end{figure}

\section{Discussion}

\subsection{Uncertainty analysis}

Our results are subject to uncertainties arising from errors in the input data, the albedo retrieval approach, the general data processing and the availability of data. An assessment of the uncertainties stemming from the input data, such as saturation or stripping issues in Landsat TM and ETM data, as well as the performance of the narrow-to-broadband formula (Equation \ref{eq:1}) is beyond the scope of this study. Moreover, it became clear, that the most recent Landsat 8 data provide better results for snow-covered areas compared to Landsat TM and ETM (cf. Figure \ref{fig:3}). However, as this study only focuses on bare-ice areas, this input data quality issue is not influencing the analysis of temporal albedo evolution.

The evaluation of bare-ice versus snow-covered grid cells might result in some misclassified cells. Clouds that were not detected by the cloud removal algorithm and shadows due to clouds and/or steep topography may further influence/falsify calculated bare-ice albedos of individual grid cells. However, manual checks revealed a low frequency of such cases. Uncertainty due to mixed pixels, specifically pixels along the margins of a glacier, can influence the temporal albedo trend observed in these areas. We minimized this effect by using glacier outlines updated to 2016 in order to exclude grid cells from the analysis that become ice-free towards the end of the study period.

To account for the uncertainty introduced by the temporal availability of remote sensing data we performed a comprehensive uncertainty analysis based on ten end-of-summer Landsat 8 scenes acquired between 2013 and 2016 (Table \ref{tab:4}). The analysis was performed for one glacier,

Findelen, as most scenes were available for this glacier due to the overlapping coverage of this glacier by two different Landsat scenes (path/row 194/28 and 195/28). For the same grid cell and multiple satellite scenes acquired during the same year (1--5 weeks apart at maximum) we found an average variability in inferred albedo of 0.026 over all four investigated years (2013--2016) (Table \ref{tab:4}). Assuming that bare-ice albedo remains constant over this short time period in reality, this value provides a direct uncertainty estimate for local satellite-retrieved albedo.

\begin{table*}[h]

\caption{Overview of scenes used in the uncertainty analysis. Px refers to the number of pixels that were used to derive uncertainty. Mean ($\alpha$\textsubscript{mean}), minimum ($\alpha$\textsubscript{min}) and maximum ($\alpha$\textsubscript{max}) albedo, as well as the mean ($\sigma$\textsubscript{mean}) standard deviation of point-based bare-ice albedo for each individual scene pair or triple per year are given.}

\centering

\begin{tabular}{lcccccc}%{\textwidth, column = c}

\tophline

Year     & Day &          Px       & $\alpha$\textsubscript{mean} & $\alpha$\textsubscript{min} & $\alpha$\textsubscript{max} & $\sigma$\textsubscript{mean} \\

\middlehline

\multirow{2}{*}{2013} & 09 Sept. & \multirow{2}{*}{1190} & 0.204 & 0.052 & 0.370 & \multirow{2}{*}{0.040} \\

                      & 25 Sept. &                       & 0.233 & 0.051 & 0.361 &                        \\

\middlehline

\multirow{3}{*}{2014} & 26 Aug.  & \multirow{3}{*}{3869} & 0.213 & 0.051 & 0.382 & \multirow{3}{*}{0.024} \\

                      & 01 Sept. &                       & 0.224 & 0.052 & 0.383 &                        \\

                      & 27 Sept. &                       & 0.255 & 0.054 & 0.396 &                        \\

\middlehline

\multirow{3}{*}{2015} & 07 Aug.  & \multirow{3}{*}{3446} & 0.174 & 0.052 & 0.403 & \multirow{3}{*}{0.031} \\

[Figure]

& 30 Aug. &
& 0.178 &
0.051 & 0.356 &
\\

& 08 Sept. &
& 0.236 &
0.053 & 0.358 &
\\

\middlehline

\multirow{2}{*}{2016} & 25 Aug.  & \multirow{2}{*}{5495} & 0.152 & 0.051 & 0.315 & \multirow{2}{*}{0.008} \\

& 01 Sept. &
& 0.156 &
0.051 & 0.295 &
\\

\middlehline

\multicolumn{2}{c}{mean for 2013--2016} & 3500 & & & & 0.026 & \\

\bottomhline

\end{tabular}

\label{tab:4}

\belowtable{} % Table Footnotes

\end{table*}

To assess the impact of local albedo uncertainty on the determination and the robustness of potential temporal trends, we randomly perturbed the distributed bare-ice albedo values of every grid cell and scene, and for all 39 individual glaciers with the computed average uncertainty of local albedo of 0.026 (average pixel number of 3500). The re-evaluation of the long-term albedo trends significant at the 80\% level according to the MK test revealed that they were not affected by the random perturbation of the albedo values. Both a very similar area of the glaciers' bare-ice surfaces and distribution of trend magnitude was found in the perturbed datasets. However, for trends significant at the 95\% confidence level or higher a slightly smaller area (11\,km\textsuperscript{2}) was detected (c.f. Table 3). Within this area, the majority (77\%) of all pixels is affected by negative trends, which is highly similar as obtained by the original albedo datasets (cf. Table \ref{tab:3}). Thus, the inferred trends in local bare-ice albedo are considered to be robust despite the uncertainty in the albedo retrieval.

\subsection{Snap-shot uncertainty and dependencies}

In contrast to the quasi-continuous measurement setup of an automatic weather station, which is however only representative for a limited spatial extent \citep{Ryan2017a}, airborne and spaceborne remote sensing datasets only represent a snap-shot in time. Hence, the temporal variability is only included to a certain degree and thus provokes a snap-shot uncertainty in surface albedo for evolution analyses. The meteorological conditions prior to the acquisition of the remote sensing imagery are highly important for the snap-shot uncertainty \citep{Fugazza2016}. \citet{Naegeli2017} highlighted this fact by cross-comparing albedo products from three different sensors with acquisition times within one week. If glacier-wide albedo is compared, a dataset acquired later in the ablation season is expected to show a larger bare-ice area characterised by low albedo values compared to a dataset acquired at the beginning of the melting period. However, this is only true, if meteorological characteristics between the individual acquisition dates are relatively constant. Snowfall or heavy rainfall events might significantly alter the ice surface conditions and the associated albedo values. While fresh snow increases the albedo strongly \citep[e.g.][]{Brock2004} and decreases the extent of the bare-ice area \citep{Naegeli2017}, rain can have a two-sided effect. A heavy precipitation event can lead to a short-term (between 1 to 4 days \citep{Azzoni2016}) increase in albedo due to decreasing surface roughness and/or wash-out of fine debris present on the ice surface (between 5 to 20\% according to \citet{Brock2004} and \citet{Azzoni2016}), whereas light rainfall can cause the presence of a thin waterfilm on the glacier ice surface that absorbs radiation much stronger than the underlying ice and thus result in a decreased albedo. Similarly, a long-lasting phase with high air temperatures or intense shortwave radiation input during mid-day can lead to a permanent or temporary waterfilm on the ice surface that reduces reflectivity and thus shortwave broadband albedo considerably \citep{Cutler1996, Jonsell2003, Paul2005}. Moreover, a remaining thin snow cover might cause slightly increased albedo values in the ablation area (still being in the typical range of glacier ice) that is difficult to be recognized with remote sensing data sets only \citep{Naegeli2017}.

The strong spatio-temporal variability of shortwave broadband albedo is dependent on various factors modulating the local glacier surface characteristics. Besides influencing the state of glacier surface albedo, these dependencies also impact on the changes in bare-ice albedo. For example, 
[revised manuscript text omitted]

%% The following commands are for the statements about the availability of data sets and/or software code corresponding to the manuscript.

%% It is strongly recommended to make use of these sections in case data sets and/or software code have been part of your research the article is based on.

%\codeavailability{TEXT} %% use this section when having only software code available

%

%

%\dataavailability{TEXT} %% use this section when having only data sets available

%

%

%\codedataavailability{TEXT} %% use this section when having data sets and software code available

%\appendix

%\section{}   %% Appendix A

%

%\subsection{}    %% Appendix A1, A2, etc.

%

%

%\noappendix      %% use this to mark the end of the appendix section

%% Regarding figures and tables in appendices, the following two options are possible depending on your general handling of figures and tables in the manuscript environment:

%% Option 1: If you sorted all figures and tables into the sections of the text, please also sort the appendix figures and appendix tables into the respective appendix sections.

%% They will be correctly named automatically.

%% Option 2: If you put all figures after the reference list, please insert appendix tables and figures after the normal tables and figures.

%% To rename them correctly to A1, A2, etc., please add the following commands in front of them:

%\appendixfigures  %% needs to be added in front of appendix figures

%

%\appendixtables   %% needs to be added in front of appendix tables

%% Please add \clearpage between each table and/or figure. Further guidelines on figures and tables can be found below.

%

%\authorcontribution{TEXT} %% optional section

%

%\competinginterests{TEXT} %% this section is mandatory even if you declare that no competing interests are present

%

%\disclaimer{TEXT} %% optional section

\begin{acknowledgements}

This study is funded by a grant of the Swiss University Conference and ETH board in frame of the KIP-5 project Swiss Earth Observatory Network (SEON). KN was further supported by an Early Postdoc.Mobility fellowship of the Swiss National Science Foundation (SNSF, grant P2FRP2\_174888). Landsat Surface Reflectance products were provided by the courtesy of the U.S. Geological Survey Earth Resources Observation and Science Center. The digital elevation model was obtained from the Federal Office of Topography swisstopo. We thank T. Irvine-Fynn for his constructive comments on the manuscript. Two anonymous reviewers are acknowledged for their constructive comments.

\end{acknowledgements}

---

## Author Response (AR2)

Dear Editor,

We hereby submit the revised manuscript with a new title "Change detection of bare-ice albedo in the Swiss Alps?" by Kathrin Naegeli, Matthias Huss and Martin Hoelzle to be re-considered for publication as an article in The Cryosphere. We have taken into account all comments by the two anonymous reviewers and responded to all concerns raised.

The main revisions concern:

- the change of the title
- more details about the methodology of the analysis regarding the surrounding geology
- consistently distinguish between local and regional albedo change
- inclusion of more information about local, positive albedo changes
- expansion of discussion of possible causes for local, positive albedo changes
- reformulation of the conclusions

Below we respond to all comments by the two anonymous referees. The responses (bold font style) are following the referees' comments (normal font style) directly. The corresponding revised sentences in the manuscript are given in quotation marks.

We would like to thank you in advance for your consideration of the article and we are looking forward to your reply.

Sincerely,

Kathrin Naegeli and co-authors
* * *
**Comments by anonymous Referee #1**

Reviewer comment: I ackowledge that the the Authors answered to my previous review of their paper. Most of my concerns were addressed. I still have some perplexities regarding the comparison with the lithology of rocks surrounding the glaciers. In section 3 (Methods) there is no explaination on how they performed the comparison between rocks and albedo trends. The CAI index (CERCHAR Abrasivity Index) is presented only in the results, there is no mention on how it was obtained and why. The relation between rock types and darkening trends is very interesting, and should be developed with more details in my opinion, since it could be an explaination of the variability in albedo trends.

**Answer: We agree on the fact that this analysis was not well introduced. We thus added the following statements:**

*Related statement in the Study site and data section:*

"To contextualise our results, the lithology surrounding the individual glaciers based on the lithological-petrographic map of Switzerland (GK500) provided by Swiss Geotechnical Commission (SGTK) was used; the map is at 1:500000 scale, and shows the subsurface strata subdivided into 25 groups according to their formation, their mineralogical composition, their particle size and their crystallinity. Based on *Käsling and Thuro*, [2010, see their Table 2] these groups were divided into less abrasive rocks (calcareous phyllites, limestones and marly shales, CERCHAR Abrasivity Index (CAI) 0–2) and very to extremely abrasive rocks (amphibolites, basic rocks, gneiss, granites, mica shists and syenites, CAI 2–6). Thereupon each individual glacier was assigned to one of the sub-groups (CAI 0-2 or CAI 2-6)."

*Related statement in the Discussion section:*

"Apart from the meteorological conditions that strongly influence bare-ice surfaces, the surrounding lithology of a glacier determines (at least partially) the availability of fine debris material that can be transported by wind and water, and be deposited on the glacier ice, reducing its albedo considerably (Di Mauro et al., 2015, 2017; Azzoni et al., 2016). Thus, easily erodible rock-types provide more loose material that might be transported by wind and water on to the glacier surface and, hence, impact the bare-ice albedo. This is supported by our analysis of the surrounding lithology and the albedo change of each individual glacier. However, no relation between the albedo of the surrounding geology and the magnitude of the ice albedo change was evident."
* * *
Reviewer comment: In line 3 pg 18 I read: "No relation between the albedo of the surrounding geology and the magnitude of the ice albedo change was evident however"
So, I suppose that surrounding rocks were classified on the basis of their albedo (?) and then the albedo was compared (linear regression? PCA?) with the magnitude of ice albedo change (averaged for the whole glacier?). More details should be provided on this further comparison. I did not find anything in the methods.

**Answer: To clarify these questions, we would like to note that we did not investigate the albedo of the surrounding rocks in detail. In fact, we classified the 25 rock types from the lithological-petrographic map based on the CERCHAR Abrasivity Index (CAI) into two groups (see also answer and clarification in the suggested text above) and assigned each glacier to one of the groups. We then analysed the albedo change per decade per glacier in regard to these two groups (less and very/extremely abrasive). As stated in the manuscript, no relation between the albedo of the surrounding geology (i.e. brighter and darker rocks) and the albedo change on the ice was found. Therefore, this analysis only supports the fact that easily erodible rock-types provide more loose material that might be transported by wind and water on to the glacier surface and hence impact the bare-ice albedo compared to very to extremely abrasive rock-types.**

Comments by anonymous Referee #2

Reviewer comment: I thank the authors for considering my comments and submitting a revised version of their manuscript. Importantly, the authors considered my main concern about their conclusion by refining their glacier mask to exclude from the analysis some obvious areas of glaciers that did not qualify as bare-ice, namely medial moraines and areas where tributaries separated from the main glacier. I also appreciate that the authors tidied up the numerous oversights in dates of the images being used. It will certainly facilitate the consideration of this work in the future. Despite this modification, I however regret I do not find in this revision a large enough improvement to address fully my overarching concerns.

**Answer: We regret that we did not fulfil the reviewer's expectations in our revision of the manuscript. Thanks to the comments below, we further improved the manuscript and addressed the concerns raised.**
* * *
Reviewer comment: Despite this adjustment in the methodology and revised results, I still find that the areas of significant change in albedo remain largely indicative of step-change in surface conditions, change in flow and possibly compounded with imperfect co-registration that I find confusingly presented as a regional subtle decrease in bare-ice albedo.

**Answer: We would like to point out that our results do not show any darkening of bare-ice at an ablation-area or regional scale and that this is clearly stated in the manuscript. While the uncertainty that might be introduced with imperfect co-registration is mentioned in the section "uncertainty assessment", the mentioned step-change in surface conditions or change in flow is not hindering the detection of an albedo change over time, but is rather a possible cause of the observed changes.**

*Related statement in the Method section:*

"These data are geo-referenced with ≤ 12m radial root-mean-square error and intercalibrated across the different Landsat sensors (Young et al., 2017)."

*Related statement in the Discussion section:*

"In the frame of this study, we were unable to detect a spatially wide-spread, regional trend in bare-ice glacier albedo at a significant confidence level. However, for certain regions of the glaciers, such as the lowermost glacier tongues or along the lower margins, significant negative trends were found. Hence, a clear darkening was observed at the local scale for a limited number of grid cells rather than for entire ablation areas."
* * *
Reviewer comment: I don't think that the pattern of significant trends presented in Figure 5 and 6 when scrutinized with the corresponding images in the context of glacier flow and demise, fully and unambiguously support the conclusions being drawn or the way there are presented.

**Answer: We fully understand the reviewer's concerns and have invested more effort to better isolate the significant conclusions given by our analysis. We also pay more attention to local effects, and have added a detailed discussion of these issues, and their impact on our results. Please see the revised conclusions at the very end.**

*Related statements in the Results section:*

"The darkening can be attributed to different causes. At the glacier termini, an accumulation of fine debris due to the deposition of allochthonous material and/or melt-out of englacial debris is most likely. These materials, together with the presence of organic material, usually dark and humic substances, decrease local albedo values considerably and foster the growth of algae and bacteria (Hodson et al., 2010; Yallop et al., 2012; Takeuchi, 2013; Stibal et al., 2017)."

"Along the glacier margins an increase in debris cover due to small collapses or input of morainic material and, hence, a deposition of rather thick debris on the bare-ice is possible. Moreover, the appearance of debris-rich basal ice alongside the lower glacier margins due to the general glacier recession poses a further cause of local darkening (Hubbard and Sharp, 1995; Hubbard et al., 2009)."

"In contrast, we also find significant positive albedo trends for some locations on the glacier tongues (see Figure 7). These might be explained by the effect of glacier flow changing the position of the medial moraine, hence leading to a transition from debris-covered to

clean ice with a higher albedo for certain grid cells. Lateral shifts of the position of medial moraines are possible for retreating glaciers (Anderson, 2000)."

*Related statements in the Discussion section:*

"Thus, the occurrence of grid cells with positive albedo changes is not surprising, but hard to explicitly link to one specific cause such as the dynamics of medial moraines. The latter might favour local positive albedo changes over time. Localized microtopographic effects, i.e. changes in slope and aspect or modulations in the surface crust (e.g. growth of larger, brighter ice crystals) and the development of cryoconite holes (in contrast to a thin dispersed debris layer) can also strongly impact the evolution of bare-ice albedo."
* * *
Reviewer comment: Figure 7 although restrictive in space remains revealing of the obvious departure between what appears discussed as a general trend and what pattern of change the analysis truly reveals.

**Answer: To account for this comment, we distinguish between local, ablation-area and regional trends more clearly to align our discussion and conclusions more tightly with the obtained results, which indeed show no general trends (neither at ablation-area nor regional scale), but significant albedo changes at a local grid cell scale.**

*See related statements in the Discussion section above.*

*Related statements in the Conclusions section:*

"While we did not find a darkening of bare-ice glacier areas at the regional scale or averaged for the ablation areas of individual glaciers, significant albedo trends (95% confidence level or higher) were revealed at the local scale. These individual grid cells or small areas were mainly located at the glacier termini or along the lower glacier margins in case of negative albedo trends, and along the central flowline further up-glacier in case of positive albedo trends."
* * *
Reviewer comment: To some extent, I find the authors attach a lot of importance to the overall negative trend without fully discarding that the trend may be an effect of the relative share of bare-ice becoming largely debris-covered in a context of glacier recession, and despite some amendments to the glacier mask. Although this phenomenon is mentioned, I don't think its full effect on the general conclusion is fairly represented.

**Answer: Again, we would like to point out that our results do not show any darkening of bare-ice at an ablation-area or regional scale. Due to accumulation of impurities mainly along the lower margins of the glaciers, a significant negative albedo change could be detected locally, however. This accumulation is linked to the increased availability of loose debris alongside the glaciers (lateral and end moraines) due to glacier recession or debris input from other source such as avalanches, wind transport or melt-out of englacial debris. We discuss the possible causes of this albedo change at a local scale at several places in the manuscript.**

*Related statements in the Results section:*

"At the glacier termini, an accumulation of fine debris due to the deposition of allochthonous material and/or melt-out of englacial debris is most likely."

"Along the glacier margins an increase in debris cover due to small collapses or input of morainic material and, hence, a deposition of rather thick debris on the bare-ice is possible."

*Related statements in the Discussion section:*

"In the frame of this study, we were unable to detect a spatially wide-spread, regional trend in bare-ice glacier albedo. However, for certain regions of the glaciers, such as the lowermost glacier tongues or along the lower margins, significant negative trends were found. Hence, a clear darkening was observed at the local scale for a limited number of grid cells rather than for entire ablation areas."

"(…), the surrounding lithology of a glacier determines (at least partially) the availability of fine debris material that can be transported by wind and water, and be deposited on the glacier ice, reducing its albedo considerably (Di Mauro et al., 2015, 2017; Azzoni et al., 2016). Thus, easily erodible rock-types provide more loose material that might be transported by wind and water on to the glacier surface and hence impact the bare-ice albedo."
* * *
Reviewer comment: To some extent, the pattern of significant changes should equally invite to discuss areas of positive trend, which are however ignored.

**Answer: We agree on the fact that the amount of grid cells exhibiting positive albedo trends was only mentioned in the results but not included in the discussion. We therefore added a respective statement to paragraph 5.3 "Possible causes and dependencies of bare-ice darkening" in addition to other places where the positive changes in albedo are mentioned.**

*Related statements in the Results section:*

"For some grid cells, about 15% or 2 km$^2$, also positive albedo trends significant at the 95% confidence level were detected however."

"In contrast, we also find significant positive albedo trends for some locations on the glacier tongues (see Figure 7). These might be explained by the effect of glacier flow changing the position of the medial moraine, hence leading to a transition from debris-covered to clean ice with a higher albedo for certain grid cells. Lateral shifts of the position of medial moraines are possible for retreating glaciers (Anderson, 2000)."

*Related statements in the Discussion section:*

"However, in the context of this study it is important to note that lateral shifts and growth and/or loss in volume of medial moraines might strongly impact the albedo evolution of some parts of the glaciers. Areas covered by thick debris were excluded from all analyses, but some mixed grid cells alongside medial moraines might still impact the results locally. Thus, the occurrence of grid cells with positive albedo changes is not surprising, but hard to explicitly link to one specific cause such as the dynamics of medial moraines. The latter might favour local positive albedo changes over time. Localized microtopographic effects, i.e. changes in slope and aspect or modulations in the surface crust (e.g. growth of larger, brighter ice crystals) and the development of cryoconite holes (in contrast to a thin dispersed debris layer) can also strongly impact the evolution of bare-ice albedo."
* * *
Reviewer comment: Since no trend is detected overall, the question of where and how much the albedo has risen to compensate the decrease in ablation areas could also be expected.

**Answer: As shown in figures 5 and 6 in the manuscript there is an overall trend for the investigated bare-ice areas of the 39 glaciers. However, this general trend is not significant (significant being defined with 95% confidence level or higher) and thus not discussed in detail. Albedo changes with a significant trend level only occur at local scale, both in negative and positive direction. We agree that the areas with significant positive albedo trends were not mentioned and discussed clearly so far and thus worked on this issue. Please see various answers to comments above and below.**
* * *
Reviewer comment: This revision therefore does not fundamentally change my perception of a disconnect between the overall conclusions and what I believe can be interpreted from the results. The fact that the title remains a provocative question is also, to me, revealing of an analysis that is finally not truly conclusive as I suggested in my earlier comments.

**Answer: The original title formulated as a question was intended to point out that there is no clear, simple, general conclusion about darkening of bare-ice in the Swiss Alps. However, as we do not want to mislead future readers by the title, we agree to adjust it and make the following suggestion:**

"Change detection of bare-ice albedo in the Swiss Alps"
* * *
Reviewer comment: The fact that the study to assess change at the scale of each glacier considers variable bare-ice area that depend on the size of the remaining accumulation area for each year remains problematic. The albedo of ice is not equal everywhere and one could argue that albedo of bare ice would tend to decrease towards the terminus. In this study, the weight of such effect is unequal through each year and to me remain a methodological issue that the author did not address

in a way that I find suitable.

Answer: We agree with the reviewer that the methodology of this analysis is questionable. We thus investigated the effect of using variable, yearly outlines for one individual glacier (Findelen, greatest data availability) on the mean bare-ice albedo per year . The analysis showed, that the impact is negligible, i.e. less than 3% (or 0.0005) difference in mean bare-ice albedo, if yearly outlines are used compared to using the 2016 outline. For example, for the year 2005 we find a mean bare-ice albedo of 0.0196 when using the 2005 outline compared to 0.0191 when using the 2016 outline. The trend in ablation-area albedo remains insignificant. Thus, we will keep the results as they are stated in the manuscript. However, in response to the reviewer's comment we adjusted the wording to clarify the applied methodology and address the issue raised.

*Related title and statements in the Results section:*

"4.2 Regional and ablation-area trend in bare-ice albedo"

"We averaged mean albedo over the entire bare-ice area for each year and glacier to obtain 39 individual time-series for the study period 1999 to 2016. As the outlines from year 2016 are consistently used over time, constant, minimal extents per glacier are evaluated. In addition, overall, yearly averages were determined based on the individual time-series of the 39 glaciers (Figure 4a)."
* * *
Reviewer comment: Another example of obvious issues I can see is found in the map of albedo of Findelengletscher for 2016 in Figure 3. The potential effects of cloud misclassification remains visible close to the terminus. In this regard, I find that my earlier comment about the cloud masking approach is not convincingly addressed.

Answer: Unfortunately, Figure 3 was somewhat misleading as it included albedo values < 0.05, which were completely excluded in our analysis, however (see statement in the Methods section). We updated the figure accordingly and the stated issue for year 2016 is fixed.

*Related statement in the Methods section:*

"Unrealistic albedo values, i.e. over 1 or below 0.05, are set to no data."

[Figure]

Revised Figure 3
* * *
Reviewer comment: I also stand by my earlier comment that the claim of albedo products being of "very high accuracy" and the reference of average deviation being less than 0.001 misrepresentative or ambiguous despite the author's response. I note that the authors stressed and introduced sources of uncertainties far more that in the earlier version. While addressing some of my specific comments, I don't find that this brought much of the expected modulation to their conclusion.

Answer: We agree on the fact that the statement of average deviation being less than 0.001 could be perceived as somewhat misleading: This number is attributed to the retrieval approach only (compared to a more sophisticated method to retrieve albedo) but does not include other uncertainties (e.g. input data, general data processing or environmental factors). We thus modified the statement accordingly.

*Related statements in the Methods section:*

"As shown by Naegeli et al. (2017) this albedo retrieval approach can be applied also to the most recent mission Landsat 8 and is suitable for mountain glaciers. It provides albedo products that have a high accuracy and only deviate marginally (< 0.01) from a more sophisticated albedo retrieval approach if using the same baseline dataset. Uncertainties in the albedo product not stemming from the retrieval approach but caused by the input data or the general data processing, such as saturation problems over snow covered areas or missing topographic correction on the radiometry, are elaborated in Section 3.5."

Answer: With regard to the overall conclusions, we understand the reviewer's comment and, thus, have revised the conclusions accordingly.

"Based on 15 Landsat scenes over a 17-year study period, we assessed the spatio-temporal evolution of bare-ice glacier surface albedo for 39 glaciers in the western and southern Swiss Alps. Our results indicate that the considered spatial scale (local versus regional) is crucial for the investigation of albedo trends and the detection of a potential darkening effect that is often referred to in recent literature (Takeuchi, 2001; Oerlemans et al., 2009; Dumont et al., 2014; Wang et al., 2014; Mernild et al., 2015; Tedesco et al., 2016). While we did not find a darkening of bare-ice glacier areas at the regional scale or averaged for the ablation areas of individual glaciers, significant albedo trends (95% confidence level or higher) were, however, revealed at the local scale. These individual grid cells or small areas were mainly located at the glacier termini or along the lower glacier margins in case of negative albedo trends (84% of all significant trends), and along the central flowline further up-glacier in case of positive albedo trends (16%).

The presented study is subject to various uncertainties stemming from the input data itself, its processing and availability, the albedo retrieval approach or environmental factors. However, unfortunately most of them are hard to numeralise. Nevertheless, our uncertainty assessment revealed highly similar trend patterns, thus indicating the robustness of the inferred albedo trends. We would like to emphasize the importance of the snap-shot uncertainty — limited availability of end-of-summer scenes demand recognition. Specifically, the meteorological conditions preceding the acquisition of the satellite data can influence bare-ice albedo, e.g. summer snow fall events, and  should be taken into account.

Although, only snap-shots of glacier surface albedo are available, the almost two-decade long time-series indicate significant trends for about 13.5 km$^2$ (corresponding to about 12% of the average end-of-summer bare-ice surface in the study area) at the local scale. Thereof almost 8 km$^2$ exhibit clear negative trends of $\leq$ −0.03 per decade. In contrast, only about 2 km$^2$ of all grid cells with significant albedo trends show positive ($\geq$ +0.03 per decade) and about 4 km$^2$ show weak changes in bare-ice albedo (> −0.03 and < +0.03 per decade). For the areas with negative albedo trends over the last two decades, the ice-albedo feedback enhanced melt rates which are expected to be enforced in the near future. Even though the darkening of glacier ice has been found to occur over only a limited area of the investigated glaciers, the projected enlargement of bare-ice areas characterised by low albedo coupled with the predicted prolongation of the melt season will most likely strongly impact on the glacier surface energy balance and substantially enhance glacier mass loss."
* * *
%\title{Change detection of bare-ice albedo in the Swiss Alps }

\title{Change detection of bare-ice albedo in the Swiss Alps }

% \Author[affil]{given_name}{surname}

\Author[1,2]{Kathrin}{Naegeli}

\Author[3,4]{Matthias}{Huss}

\Author[3]{Martin}{Hoelzle}

\affil[1]{Institut of Geography and Oeschger Center for Climate Change Research, University of Bern, Bern, Switzerland}

\affil[2]{Centre for Glaciology, Department of Geography and Earth Sciences, Aberystwyth University, Wales, UK}

\affil[3]{Department of Geosciences, University of Fribourg, Fribourg, Switzerland}

\affil[4]{Laboratory of Hydraulics, Hydrology and Glaciology (VAW), ETH Zurich, Zurich}

%% The [] brackets identify the author with the corresponding affiliation. 1, 2, 3, etc. should be inserted.

\runningtitle{Change detection of bare-ice albedo in the Swiss Alps}

\runningauthor{Naegeli et al.}

\correspondence{Kathrin Naegeli (kathrin.naegeli@giub.unibe.ch)}

\received{}

\pubdiscuss{} %% only important for two-stage journals

\revised{}

\accepted{}

\published{}

%% These dates will be inserted by Copernicus Publications during the typesetting process.

\firstpage{1}

\maketitle

\begin{abstract}

[revised manuscript text omitted]

\centering

\begin{tabular}{lcccccc}%{\textwidth, column = c}

\tophline

Landsat mission &        Date &  \multicolumn{2}{c}{Bare-ice area} &  Clouds & Shadows \\

(sensor) &        (dd.mm.yyyy) &(km\textsuperscript{2}) &        (\%) &  (\%) & (\%)  \\

\middlehline

Landsat 7 (ETM+)       &        11.09.1999       &        119.7 &        27.1 &        0.1 &        4.5       \\

Landsat 7 (ETM+)       &        12.08.2000       &        89.7 &        20.3 &        0.5 &        1.4       \\

Landsat 7 (ETM+)       &        18.08.2002       &        61.0 &        13.8 &        1.0 &        3.5       \\

Landsat 5 (TM)  &        13.08.2003        &        182.5 &        41.3 &        0.0 &        0.4       \\

Landsat 7 (ETM+)    &    08.09.2004    &    132.0    &    29.9    &    0.0    &    2.5    \\

Landsat 7 (ETM+)    &    10.08.2005    &    72.6    &    16.4    &    5.9    &    1.0    \\

Landsat 5 (TM) &    22.09.2006    &    95.8    &    21.7    &    0.0    &    6.0    \\

Landsat 7 (ETM+)    &    18.08.2008    &    24.6    &    5.6    &    0.1    &    0.8    \\

Landsat 5 (TM) &    30.09.2009    &    185.2    &    41.9    &    0.0    &    7.2    \\

Landsat 7 (ETM+)    &    12.09.2011    &    146.8    &    33.2    &    0.4    &    2.8    \\

Landsat 7 (ETM+)    &    14.09.2012    &    22.1    &    5.0    &    0.8    &    3.7    \\

Landsat 8 (OLI) &    09.09.2013*    &    \multirow{2}{*}{91.1}    &    \multirow{2}{*}{20.6}    &    14.0    &    4.5    \\

Landsat 8 (OLI) &    25.09.2013*    &    &    &    0.5    &    6.7    \\

Landsat 8 (OLI) &    28.09.2014    &    153.5    &    34.7    &    1.0    &    8.3    \\

Landsat 8 (OLI) &    30.08.2015    &    214.9    &    48.6    &    1.4    &    2.1    \\

Landsat 8 (OLI) &    01.09.2016    &    189.4    &    42.8    &    2.3    &    3.1    \\

\bottomhline

\end{tabular}

\label{tab:2}

\belowtable{*For each individual glacier only one of these two scenes in 2013 is taken (based on minimal cloud and/or snow coverage).} % Table Footnotes

\end{table*}

\subsection{Albedo retrieval}

\label{subsec:albedoretrieval}

We applied the narrow-to-broadband conversion by \citet{Liang2001} to obtain shortwave broadband albedo $\alpha$\textsubscript{short} from the surface reflectance data. The conversion is based on five of the seven individual bands, and is formulated as follows:

\begin{equation}

    \alpha_{short} = 0.356\alpha _{1} + 0.130\alpha _{3} + 0.373\alpha _{4} + 0.085\alpha _{5} + 0.072\alpha _{7} - 0.0018

\label{eq:1}

\end{equation}

where $\alpha$\textsubscript{i} represents the narrowband ground reflectance of TM/ETM+ in band i. For Landsat OLI, the band numbers were adjusted accordingly. This conversion was developed based on a large empirical data set and the band configurations of Landsat TM/ETM+. As shown by \citet{Naegeli2017} this albedo retrieval approach can be applied also to the most recent mission Landsat 8 and is suitable for mountain glaciers. It provides albedo products that have a high accuracy and only deviate marginally (< 0.01) from a more sophisticated albedo retrieval approach if using the same baseline dataset. Uncertainties in the albedo product not stemming from the retrieval approach but caused by the input data or the general data processing, such as saturation problems over snow covered areas or missing topographic correction on the radiometry, are elaborated in Section 3.5. Unrealistic albedo values, i.e. over 1 or below 0.05, are set to no data.

\subsection{Surface type evaluation}

The delineation of bare-ice area versus snow-covered surfaces is based on a multi-step classification scheme of the surface albedo values (Figure \ref{fig:2}). The classification is thus based on a physical parameter specific for both snow and ice. In a first step, two threshold values for \textit{certainly snow} ($\alpha$ > 0.55) and \textit{certainly ice} ($\alpha$ < 0.25) are defined (\textit{primary surface type evaluation}, Figure \ref{fig:2}) based on recommendations in the literature \citep{Cuffey2010}. This results in a critical albedo range (0.25\,<\,$\alpha$\,<\,0.55), where an unambiguous assignment of the surface type, i.e. snow or ice, is not possible without considering other parameters. Within this range of albedo values, outliers are suppressed by adjusting all albedo values ($\alpha$\textsubscript{corr}) by multiplying with a constant value (SLA\textsubscript{const}). We, therefore, take advantage of a digital elevation model available for all glaciers to evaluate the average albedo in elevation bands of 20 m within this critical albedo range. The transition between ice and snow is typically characterized by a distinct change in albedo \citep[e.g.][]{Hall1987, Zeng1983,Winther1993}. We thus derive an estimate of the mean snowline altitude (SLA) for each glacier and scene based on the greatest slope of the albedo-elevation profile. The albedo for this altitude is considered to be the site- and scene-specific albedo threshold discriminating snow and ice and is henceforth termed $\alpha$\textsubscript{crit}.

\begin{figure}[h!]

%\includegraphics[width=8.3cm]{figures/NEW/SurfaceTypeEvaluation_FlowChart_4-01}

\includegraphics[width=8.3cm]{figures/NEW/Figure_2}

[revised manuscript text omitted]

\centering

\begin{tabular}{lcccccc}%{\textwidth, column = c}

\tophline

Year     & Day &           Px        & $\alpha$\textsubscript{mean} & $\alpha$\textsubscript{min} & $\alpha$\textsubscript{max} & $\sigma$\textsubscript{mean} \\

\middlehline

\multirow{2}{*}{2013} & 09 Sept. & \multirow{2}{*}{1190} & 0.204 & 0.052 & 0.370 & \multirow{2}{*}{0.040} \\

                                                                    & 25 Sept. &
                                                                    & 0.233 &
0.051 & 0.361 &
              \\

\middlehline

\multirow{3}{*}{2014} & 27 Aug.  & \multirow{3}{*}{3869} & 0.213 & 0.051 & 0.382 & \multirow{3}{*}{0.024} \\

                                                                    & 12 Sept. &
                                                                    & 0.224 &
0.052 & 0.383 &
              \\

                                                                    & 28 Sept. &
                                                                    & 0.255 &
0.054 & 0.396 &
              \\

\middlehline

\multirow{3}{*}{2015} & 07 Aug.  & \multirow{3}{*}{3446} & 0.174 & 0.052 & 0.403 & \multirow{3}{*}{0.031} \\

                                                                    & 30 Aug.  &
                                                                    & 0.178 &
0.051 & 0.356 &
              \\

& 08 Sept. &
& 0.236 &
0.053 & 0.358 &
            \\

\middlehline

\multirow{2}{*}{2016} & 25 Aug.  & \multirow{2}{*}{5495} & 0.152 & 0.051 & 0.315 &
\multirow{2}{*}{0.008} \\

& 01 Sept. &
& 0.156 &
0.051 & 0.295 &
            \\

\middlehline

\multicolumn{2}{c}{mean for 2013--2016} & 3500                                     &
            &                 &                 & 0.026
                                    \\

\bottomhline

\end{tabular}

\label{tab:4}

\belowtable{} % Table Footnotes

\end{table*}

To assess the impact of local albedo uncertainty on the determination and the robustness of potential temporal trends, we randomly perturbed the distributed bare-ice albedo values of every grid cell and scene, and for all 39 individual glaciers with the computed average uncertainty of local albedo of 0.026 (average pixel number of 3500). The re-evaluation of the long-term albedo trends significant at the 80\% level according to the MK test revealed that they were not affected by the random perturbation of the albedo values. Both a very similar area of the glaciers' bare-ice surfaces and distribution of trend magnitude was found in the perturbed datasets. However, for trends significant at the 95\% confidence level or higher a slightly smaller area (11\,km\textsuperscript{2}) was detected (c.f. Table 3). Within this area, the majority (77\%) of all pixels is affected by negative trends, which is highly similar as obtained by the original albedo datasets (cf. Table \ref{tab:3}). Moreover, trends in local bare-ice albedo remained robust even if assumed uncertainties were chosen substantially higher than just the value for snap-shot uncertainty.

\section{Results}

\subsection{Spatially distributed shortwave broadband albedo}

\label{subsec:spatiallydistributed}

Figure \ref{fig:3} shows the spatio-temporal evolution of glacier-wide shortwave broadband albedo for Findelengletscher. The retrieval of meaningful albedo values is restricted by the quality of the surface reflectance data and, thus, the availability of realistic values in the individual bands needed for the narrow-to-broadband conversion. For Landsat TM/ETM+, a saturation problem over snow-covered areas exists, resulting in missing values for these regions (years 1999--2012 in Figure \ref{fig:3}). This problem is not present in the Landsat 8 data (years 2013--2016 in Figure \ref{fig:3}). Missing data in some of the Landsat ETM+ data, generated due to the scan line corrector (SLC) failure post May 2003, also occurs in our albedo retrievals (e.g. 08.09.2004 in Figure \ref{fig:3. We tested the impact of the SLC failure by simulating missing data for three scenes with an intact SLC for Findelengletscher. SLC failure resulted in slightly higher mean bare-ice albedo values (1.2 to 2.2\%, e.g. 12.08.2000 SLC-on mean bare-ice albedo 0.204 versus SLC-off mean bare-ice albedo 0.209 indicating a difference of 2.2\%), which is a negligible impact. Although, we applied a cloud removal algorithm, our results are still impacted by cloud shadows that are harder to detect without manual effort (e.g. 18.08.2002 in Figure \ref{fig:3}). However, the bare-ice area is almost always well represented and inferred albedo is realistic, hence allowing for a monitoring through time.

[revised manuscript text omitted]

\centering

\begin{tabular}{lccccc}%{\textwidth, column = c}

\tophline

Class    & Albedo trend &         \multicolumn{2}{c}{Confidence level 80\%} & \multicolumn{2}{c}{Confidence level 95\%}  \\

(\#)     & (albedo change decade\textsuperscript{-1}) & (\%) &   (km\textsuperscript{2}) &         (\%) &      (km\textsuperscript{2}) \\

\middlehline

1 &      < $-$0.05 &       8.8 &    10.1 &  28.9 &  3.9 \\

2 &      $-$0.05 to $-$0.03 &     11.2 &  12.9 &  28.2 &  3.8 \\

3 &      $-$0.03 to $-$0.01 &     21.5 &  24.6 &  25.6 &  3.5 \\

4 &      $-$0.01 to 0.01 &        26.0 &  29.8 &  1.8 &    0.2 \\

5 &      0.01 to 0.03 &   17.9 &  20.5 &  2.7 &    0.4 \\

6 &      0.03 to 0.05 &   8.4 &   9.7 &    5.8 &    0.8 \\

7 &      > 0.05 &         6.1 &    7.0 &    6.9 &    0.9 \\

\middlehline

Total &  &       100 &   114.5 & 100 &   13.5 \\

\bottomhline
\end{tabular}
\label{tab:3}
\belowtable{} % Table Footnotes
\end{table*}

\begin{figure*}[h]

%\includegraphics[width=12cm]{figures/NEW/Trend_Signi_50_plot_new_samescales_NEW}

\includegraphics[width=12cm]{figures/NEW/Figure_5}
```

\caption{Confidence levels of bare-ice albedo trends over the study period 1999 to 2016 according to the MK test. See Figure \ref{fig:1} for the location of the different panels in the Swiss Alps.}

\label{fig:5}

\end{figure*}

\begin{figure*}[h]

%\includegraphics[width=12cm]{figures/NEW/Trend_50_plot_new_7classes_numbers}

\includegraphics[width=12cm]{figures/NEW/Figure_6}

\caption{Classified albedo trends per decade for all grid cells with trends significant at the 80\% confidence level or higher according to the MK test. Averages for areas with bare-ice albedo trends significant at the 95\% confidence level are given for selected large glaciers. See Figure \ref{fig:1} for the location of the different panels in the Swiss Alps.}

\label{fig:6}

\end{figure*}

For most of the bare-ice area, the derived trends in albedo were only significant at low levels. Compared to the glaciers' overall ablation area only relatively few grid cells with trends significant at the 95\% confidence level or higher (dark blue areas in Figure \ref{fig:5}) are present. The cells with significant trends at high confidence levels are usually situated at the termini or along the lower margins of the glaciers and trends are mostly negative (cf. Table \ref{tab:3}, Figures \ref{fig:5}--\ref{fig:7}). The darkening can be attributed to different causes. At the glacier termini, an accumulation of fine debris due to the deposition of allochthonous material and/or melt-out of englacial debris is most likely. These materials, together with the presence of organic material, usually dark and humic substances, decrease local albedo values considerably and foster the growth of algae and bacteria \citep{Hodson2010, Yallop2012, Takeuchi2013, Stibal2017}. However, many of these effects and interactions are still unclear. Along the glacier margins an increase in debris cover due to small collapses or input of morainic material and, hence, a deposition of rather thick debris on the bare-ice is possible. Moreover, the appearance of debris-rich basal ice alongside the lower glacier margins due to the general glacier recession poses a further cause of local darkening \citep{Hubbard1995, Hubbard2009}. Along the central area of the glacier tongue, particularly in the vicinity of medial moraines (e.g. in the case of Gornergletscher, Figure \ref{fig:7}), a strongly negative albedo trend indicates an expanding medial moraine, changing the local area from clean to (partly) debris-covered ice. In contrast, we also find significant positive albedo trends for some locations on the glacier tongues (see Figure \ref{fig:7}). These might be explained by the effect of glacier flow changing the position of the medial moraine, hence leading to a transition from debris-covered to clean ice with a higher albedo for certain grid cells. Lateral shifts of the position of medial moraines are possible for retreating glaciers \citep{Anderson2000}.

The investigation of the lithology surrounding the 39 individual glaciers and their overall albedo trend observed for the study period (Table \ref{tab:1}) revealed that glaciers predominantly surrounded by

less abrasive rocks (calcareous phyllites, limestones and marly shales, CERCHAR Abrasivity Index (CAI) 0--2 after \citet{Kasling2010}) exhibited a stronger negative albedo change of $-$0.05 per decade compared to glaciers that are located in an area of very to extremely abrasive rocks  ($-$0.03 albedo change per decade; amphibolites, basic rocks, gneiss, granites, mica shists and syenites, CAI 2--6 after \citep{Kasling2010}).

\begin{figure}[h]

%\includegraphics[width=8.3cm]{figures/NEW/Trend_Signi_50_ge95_plot_new_samescales_ALETSCH_NEW2-01}

\includegraphics[width=8.3cm]{figures/NEW/Figure_7}

[revised manuscript text omitted]

%% The following commands are for the statements about the availability of data sets and/or software code corresponding to the manuscript.

%% It is strongly recommended to make use of these sections in case data sets and/or software code have been part of your research the article is based on.

%\codeavailability{TEXT} %% use this section when having only software code available

%

%

%\dataavailability{TEXT} %% use this section when having only data sets available

%

%

%\codedataavailability{TEXT} %% use this section when having data sets and software code available

%\appendix

%\section{}    %% Appendix A

%

%\subsection{}    %% Appendix A1, A2, etc.

%

%

%\noappendix     %% use this to mark the end of the appendix section

%% Regarding figures and tables in appendices, the following two options are possible depending on your general handling of figures and tables in the manuscript environment:

%% Option 1: If you sorted all figures and tables into the sections of the text, please also sort the appendix figures and appendix tables into the respective appendix sections.

%% They will be correctly named automatically.

%% Option 2: If you put all figures after the reference list, please insert appendix tables and figures after the normal tables and figures.

%% To rename them correctly to A1, A2, etc., please add the following commands in front of them:

%\appendixfigures  %% needs to be added in front of appendix figures

%

%\appendixtables   %% needs to be added in front of appendix tables

%% Please add \clearpage between each table and/or figure. Further guidelines on figures and tables can be found below.

%

%\authorcontribution{TEXT} %% optional section

%

%\competinginterests{TEXT} %% this section is mandatory even if you declare that no competing interests are present

%

%\disclaimer{TEXT} %% optional section

\begin{acknowledgements}

This study is funded by a grant of the Swiss University Conference and ETH board in frame of the KIP-5 project Swiss Earth Observatory Network (SEON). KN was further supported by an Early Postdoc.Mobility fellowship of the Swiss National Science Foundation (SNSF, grant P2FRP2\_174888). Landsat Surface Reflectance products were provided by the courtesy of the U.S. Geological Survey Earth Resources Observation and Science Center. The digital elevation model was obtained from the Federal Office of Topography swisstopo. We thank T. Irvine-Fynn for his constructive comments on the manuscript. Two anonymous reviewers are acknowledged for their constructive comments.

\end{acknowledgements}